# CONVERGENCE OF MUON WITH NEWTON–SCHULZ

**Gyu Yeol Kim**
Seoul National University
gyuyeolkim@snu.ac.kr

**Min-hwan Oh**
Seoul National University
minoh@snu.ac.kr

## ABSTRACT

We analyze MUON as originally proposed—using the momentum orthogonalization with a few NEWTON–SCHULZ steps. The prior theoretical results replace this key step in MUON with an exact SVD-based polar factor. We prove that MUON with NEWTON–SCHULZ converges to a stationary point at the same rate as the SVD-polar idealization, up to a constant factor for a given number $q$ of NEWTON–SCHULZ steps. We further analyze this constant factor and prove that it converges to 1 doubly exponentially in $q$ and improves with the degree of the polynomial used in NEWTON–SCHULZ for approximating the orthogonalization direction. We also prove that MUON improves the rank dependence compared to its vector-based counterpart, SGD with momentum. Our results explain why MUON with a few low-degree NEWTON–SCHULZ steps matches exact-polar (SVD) behavior at a much faster wall-clock time and explain how much momentum matrix orthogonalization via NEWTON–SCHULZ benefits over the vector-based optimizer. Overall, our theory justifies the practical NEWTON–SCHULZ design of MUON, narrowing its practice–theory gap.

## 1 INTRODUCTION

Modern deep neural networks comprise billions of parameters and demand highly efficient training procedures. A persistent challenge is that most widely used optimizers—such as stochastic gradient descent (SGD) (Robbins & Monro, 1951) and adaptive methods such as Adam (Kingma & Ba, 2015)—operate on *vectorized* parameters, thereby discarding the native *matrix* structure present in linear layers and attention projections. Optimizers that explicitly respect matrix structure can, in principle, yield search directions that are better aligned with the underlying geometry while remaining computationally efficient at scale.

MUON (Jordan et al., 2024) is an optimizer designed for matrix-structured parameters. At each iteration, instead of following the raw momentum, MUON *orthogonalizes* the momentum matrix and then uses this orthogonalized direction to update the weights. In practice, this orthogonalization is not computed via an exact singular value decomposition (SVD)—which is accurate but expensive—but is approximated efficiently by a small, fixed number of NEWTON–SCHULZ steps. Empirical studies (Jordan et al., 2024; Liu et al., 2025a) have reported strong performance at scale with this SVD-free implementation, making MUON an attractive alternative to vector-based optimizers. Despite recent attempts to analyze the convergence of MUON (Shen et al., 2025; Li & Hong, 2025; Sato et al., 2025; Pethick et al., 2025a;b), theory still lags behind practice. Existing analyses typically study an *idealized* variant that replaces the NEWTON–SCHULZ step—central to practical MUON—with an exact polar step computed by SVD for analytical convenience. This leaves open whether the *actual* SVD-free orthogonalization used in practice—i.e., a finite number of NEWTON–SCHULZ steps—admits principled nonconvex convergence guarantees, and how the NEWTON–SCHULZ approximation impacts rank dependence and efficiency. Therefore, the following research questions remain open:

**Research questions.**

- Does MUON with NEWTON–SCHULZ admit nonconvex convergence guarantees, and how do its rates compare to the exact SVD–polar idealization?

- How does the NEWTON–SCHULZ steps $q$ and NEWTON–SCHULZ polynomial degree $\kappa$ control the accuracy–compute trade-off? In particular, how large is the gap caused by using NEWTON–SCHULZ, and how does that gap become negligible?

- Can we show that MUON converges faster than the vector-counterpart, SGD with momentum? What geometric mechanisms and rank dependence drive this gap?

To address these questions, we analyze MUON as originally proposed (Jordan et al., 2024): MUON with momentum orthogonalization computed via NEWTON–SCHULZ. For nonconvex objectives, under standard smoothness assumptions, we prove the convergence of MUON to a stationary point, measured by the nuclear norm of the gradient. We establish that the convergence rate in the number of iterations matches the idealized (but not used in practice) SVD-based exact polar variant, up to a constant factor that depends on the polar approximation error $\varepsilon_q$ (defined in Definition 3) for the fixed number of NEWTON–SCHULZ steps $q$. Moreover, we show that the polar approximation error $\varepsilon_q$ shrinks doubly exponentially as $q$ grows and decays with larger $\kappa$, which is the degree of the polynomial used in NEWTON–SCHULZ steps. Recursive updates using this polynomial allow the optimizer to find the approximated orthogonalization direction of the momentum matrix. Hence, with only a few NEWTON–SCHULZ steps, the convergence rate of MUON quickly tends to the convergence rate of the exact polar step via SVD. Consequently, our results imply that, because a few NEWTON–SCHULZ steps are far cheaper per iteration than SVD, the practical MUON implementation with NEWTON–SCHULZ attains substantially faster wall-clock convergence.

Our main contributions are summarized as follows:

- **The first convergence result of MUON with NEWTON–SCHULZ.** To our knowledge, we present the first nonconvex convergence guarantees for MUON with a finite number of NEWTON–SCHULZ steps (Theorem 1), as originally proposed and practically used. It is important to note that even for convex optimization, the convergence of MUON with NEWTON–SCHULZ has not been shown previously. The key distinction from the existing analyses of MUON is that we do not replace NEWTON–SCHULZ in the original MUON with the exact polar computed by SVD.

- **Analysis of polar approximation error and wall-clock convergence.** We prove that the polar approximation error $\varepsilon_q$ due to using NEWTON–SCHULZ instead of SVD, in the MUON optimizer, decays doubly exponentially with the number of NEWTON–SCHULZ steps $q$ and decays with the degree $\kappa$ of the polynomial required to approximate the orthogonalization direction of the momentum matrix (Definition 2). Thus, even with a few steps of NEWTON–SCHULZ, the convergence of MUON with NEWTON–SCHULZ becomes arbitrarily close to that of the SVD-variant in the number of iterations (Theorems 2 and 4). Hence, given that per-iteration computation is much more efficient for NEWTON–SCHULZ steps compared to SVD, the overall convergence in wall-clock time is much faster for MUON with NEWTON–SCHULZ.

- **Sharper rank dependence in MUON with NEWTON–SCHULZ.** To prove the comparative advantage of MUON against the vector-based counterpart, we demonstrate for the first time that MUON with NEWTON–SCHULZ sharpens the rank dependence of the momentum matrix in the leading term, compared to its vector-based counterpart, SGD with momentum. (see Table 1 and Theorem 3).

## 2 RELATED WORK

**MUON and momentum orthogonalization.** Jordan et al. (2024) introduced MUON, which orthogonalizes a momentum matrix via a few NEWTON–SCHULZ steps (SVD-free), and reported strong empirical results at LLM scale (Liu et al., 2025a). Earlier work orthogonalized gradients by SVD before applying momentum (Orthogonal-SGDM; Tuddenham et al. (2022)), whereas MUON applies momentum before orthogonalization and replaces SVD with NEWTON–SCHULZ, resulting in faster computation with only matrix multiplications. More details about MUON are described in 3.3.

**Second-order preconditioners vs. MUON.** Matrix-aware optimizers such as SHAMPOO (Gupta et al., 2018) SOAP (Vyas et al., 2024) and their variants (An et al., 2025) are *second-order preconditioners*: they maintain layerwise curvature (Kronecker-factored second moments) and periodically

apply inverse-root preconditioning. By contrast, *Muon is not a second-order method*; it neither estimates nor inverts curvature but orthogonalizes the momentum matrix via a few Newton–Schulz iterations (SVD-free, using only matrix multiplications). These methods target different mechanisms—curvature preconditioning vs. projection/normalization—and can be complementary rather than directly comparable.

**Practical efficiency of MUON.** Large-scale training reports for MUON (Liu et al., 2025a; Shah et al., 2025; Tveit et al., 2025) and communication/memory-aware variants (Ahn & Dion, 2025; Liu et al., 2025b) motivate a theory that is SVD-free and GPU-aligned. Our analysis of NEWTON–SCHULZ, a key step in the MUON optimizer, adopts precisely that stance. The analysis provided in this work can be adapted to other variants of MUON.

**Convergence analysis of MUON.** Several recent analyses examine MUON but typically either idealize the orthogonalization by assuming an exact SVD polar step (Shen et al., 2025), work under Frobenius-smoothness with dimension-driven constants (Li & Hong, 2025), focus on stability/variant phenomena (Sato et al., 2025), or offer complementary lenses (steepest descent under norms, trust-region views, implicit constraints, or LMO/Frank-Wolfe formulations) without addressing NEWTON–SCHULZ step accuracy in nonconvex rates (Bernstein & Newhouse, 2024; Kovalev, 2025; Chen et al., 2025; Riabinin et al., 2025; Pethick et al., 2025a;b).

**Key distinctions compared to the existing analyses of MUON.** Prior work either assumes exact SVD polar steps, measures progress in a geometry that obscures rank benefits, or does not quantify the approximation from NEWTON–SCHULZ in MUON. Our work is the first to analyze how two key parameters—the number of NEWTON–SCHULZ steps and the degree of the polynomial used for finding the orthogonalization direction of the momentum matrix approximately—affect the convergence rate of MUON. The results indicate a nonconvex convergence rate to a stationary point, an explicit and rapidly vanishing constant factor derived from NEWTON–SCHULZ instead of SVD, and sharper rank dependence than SGD with momentum under the same metric. Overall, our work explains why MUON converges quickly (particularly in wall-clock time) as well as why only a few steps of NEWTON–SCHULZ suffice in practice.

## 3 PRELIMINARIES

### 3.1 NOTATIONS

For matrix $X \in \mathbb{R}^{m \times n}$, $X^\top$ is its transpose. For $X = (x_{ij}) \in \mathbb{R}^{n \times n}$, $\mathrm{tr}(X) := \sum_{i=1}^{n} x_{ii}$. We write $\|X\|_*$, $\|X\|_{\mathrm{op}}$, and $\|X\|_F$ for the nuclear, spectral (operator), and Frobenius norms, respectively, and $\langle X, Y \rangle_F := \mathrm{tr}(X^\top Y)$. For a thin SVD $X = U\Sigma V^\top$, the *polar factor* is $\mathrm{Polar}(X) := UV^\top$, a partial isometry with $\|\mathrm{Polar}(X)\|_{\mathrm{op}} \leq 1$ and $\langle X, \mathrm{Polar}(X) \rangle_F = \|X\|_*$ (See Appendix A.1). For two sequences $\{a_n\}_{n=1}^{\infty}$ and $\{b_n\}_{n=1}^{\infty}$, $a_n = \mathcal{O}(b_n)$ implies that there exists a constant $C > 0$ such that $a_n \leq C b_n$ holds for all $n \geq 1$. We use $\mathbb{E}[\cdot]$ for expectations over all algorithmic randomness.

### 3.2 PROBLEM SETTING: NONCONVEX OPTIMIZATION

We consider the stochastic optimization of a matrix-valued parameter: $W \in \mathbb{R}^{m \times n}$

$$\min_{W \in \mathbb{R}^{m \times n}} f(W) = \mathbb{E}_\xi[f(W; \xi)],$$

where the objective $f$ is nonconvex with $f^* := \inf_W f(W) > -\infty$. We denote by $r := \min\{m, n\}$ the maximal possible rank of the matrix. At iteration $t$, a mini-batch $\xi_t = (\xi_{t,1}, \ldots, \xi_{t,B})$ is drawn with $\{\xi_{t,i}\}_i$ i.i.d., and $\{\xi_t\}_t$ are independent across $t = 1, \ldots, T$. At $t = 0$, the model parameter $W_0 \in \mathbb{R}^{m \times n}$ is initialized, and we define the initial sub-optimality as $D := f(W_0) - f^*$. In this paper, the following assumptions are made for the convergence analysis:

**Assumption 1** (Lipschitz smoothness). *The objective function $f : \mathbb{R}^{m \times n} \to \mathbb{R}$ is continuously differentiable and $L$-Lipschitz smooth, i.e., for all $X, Y \in \mathbb{R}^{m \times n}$,*

$$\|\nabla f(X) - \nabla f(Y)\|_* \leq L \|X - Y\|_{\mathrm{op}}.$$

*We use smoothness with respect to the operator norm (and the nuclear norm as its dual).*

**Assumption 2** (Bounded variance). *We assume $\nabla f(W; \xi)$ is an unbiased stochastic estimator of the true gradient $\nabla f(W)$ and has bounded variance for all $W$ and a single sample $\xi$, i.e.*

$$\mathbb{E}[\nabla f(W; \xi)] = \nabla f(W), \qquad \mathbb{E}\big[\|\nabla f(W; \xi) - \nabla f(W)\|_F^2\big] \leq \sigma^2.$$

*For a mini-batch of size $B$, the variance is at most $\sigma^2/B$.*

Assumptions 1 and 2 are standard for analyzing first-order methods in stochastic optimization (Boyd & Vandenberghe, 2004; Nemirovski et al., 2009; Candes & Recht, 2012; Shapiro et al., 2021; Reddi et al., 2018; Zou et al., 2019). $L$-smoothness is typically defined using a norm and its dual, and the specific operator-nuclear geometry chosen here is a natural fit when working with matrix parameters and polar or orthogonal updates (Nesterov, 2013; Beck, 2017; Jaggi, 2013). The bounded-variance assumption, where the mini-batch variance is $\sigma^2/B$, is a foundational concept for algorithms like SGD in both convex and nonconvex scenarios (Bottou et al., 2018; Ghadimi & Lan, 2013).

In this paper, the metric for the convergence rate is defined as follows:

**Definition 1** ($\epsilon$-stationary point). *We call $W \in \mathbb{R}^{m \times n}$ an $\epsilon$-stationary point (in the nuclear norm) if $\mathbb{E}[\|\nabla f(W)\|_*] \leq \epsilon$. Equivalently, we say an algorithm attains $\epsilon$-stationarity in $T$ steps if*

$$\frac{1}{T} \sum_{t=1}^{T} \mathbb{E}[\|\nabla f(W_{t-1})\|_*] \leq \epsilon.$$

Note that when working with functions that have matrix inputs, using the nuclear norm to find a stationary point provides a more restrictive and precise condition than using the standard Frobenius norm. In other words, if a point satisfies the stationarity condition for the nuclear norm, it is guaranteed to also satisfy the condition for the Frobenius norm.

### 3.3 MUON ALGORITHM AND NEWTON–SCHULZ ORTHOGONALIZATION

---

**Algorithm 1** MUON (with the illustration of NEWTON–SCHULZ orthogonalization)

---

**Require:** Learning rate $\eta > 0$, momentum $\beta \in [0,1)$, batch size $B$, NEWTON–SCHULZ steps $q \in \mathbb{N}$, NEWTON–SCHULZ polynomial $p_\kappa$ (degree $\kappa$), total iteration $T$.

1: **Initialize:** $M_0 \leftarrow 0$, $W_0 \in \mathbb{R}^{m \times n}$
2: **for** $t = 1$ to $T$ **do**
3:      $G_t \leftarrow \frac{1}{B} \sum_{i=1}^{B} \nabla f(W_{t-1}; \xi_{t,i})$            ▷ Compute (batch) gradients
4:      $M_t \leftarrow \beta M_{t-1} + G_t$
5:      $X_{t,0} \leftarrow M_t/\alpha_t$ with $\alpha_t = \max\{1, \|M_t\|_F\}$       ▷ Pre-NEWTON–SCHULZ scaling
6:      **for** $j = 1$ to $q$ **do**
7:          $X_{t,j} \leftarrow p_\kappa(X_{t,j-1} X_{t,j-1}^\top) X_{t,j-1}$      ▷ NEWTON–SCHULZ steps (Lines 6-9)
8:      **end for**
9:      $O_t \leftarrow X_{t,q}$
10:     $W_t \leftarrow W_{t-1} - \eta O_t$                 ▷ Update parameters
11: **end for**

---

For clarity of exposition, we present pseudocode for MUON with an explicit illustration of the NEWTON–SCHULZ steps (Lines 5–9) in Algorithm 1. Note that this is not a new algorithm; it is the original method of Jordan et al. (2024), here written in a more general mini-batch form with a step-by-step illustration of NEWTON–SCHULZ. Rather than orthogonalizing via SVD, MUON approximates the orthogonalization direction using only matrix multiplications; the key mechanism enabling this is the NEWTON–SCHULZ-based orthogonalization.

The key advantages of using NEWTON–SCHULZ for orthogonalization are as follows:

- NEWTON–SCHULZ makes MUON inversion-free and SVD-free. SVD is computationally expensive and makes each iteration costly. In contrast, the NEWTON–SCHULZ approach relies solely on matrix multiplications, yielding substantially better per-iteration efficiency—especially for large parameter matrices.

- NEWTON–SCHULZ is an iterative method that allows for precise control over the degree of orthogonality by adjusting the number of iterations. The number of NEWTON–SCHULZ steps provides a direct trade-off between computational cost and the degree of orthogonality, offering valuable flexibility.

## 3.4 Newton–Schulz polynomial

In the MUON algorithm, at every iteration, a scaled momentum matrix $X$ is orthogonalized via NEWTON–SCHULZ steps. First, the matrix $XX^\top$ is formed and is then passed to a polynomial function $p_\kappa$ with degree $\kappa$. Recursive updates by this polynomial make the matrix $X$ nearly orthogonal, i.e., $XX^\top = I$. We first define this function $p_\kappa$ in Definition 2 and state the properties of this polynomial used in NEWTON–SCHULZ.

**Definition 2** (NEWTON–SCHULZ polynomial). *For degree $\kappa \in \mathbb{N}$, the* NEWTON–SCHULZ *polynomial is the Taylor truncation of $1/\sqrt{\lambda}$ at $\lambda = 1$, i.e.,*

$$p^{(s)}(1) = \frac{d^s}{d\lambda^s} \lambda^{-1/2} \Bigg|_{\lambda=1}$$

*for $s = 1, \ldots, \kappa$. The explicit form of the* NEWTON–SCHULZ *polynomial for degree $\kappa$ is*

$$p_\kappa(\lambda) = \sum_{s=0}^{\kappa} c_s (1-\lambda)^s, \qquad c_s = \frac{(2s)!}{4^s(s!)^2} > 0.$$

*Equivalently, with reparametrization $u = 1 - \lambda \in [0,1]$, $p_\kappa(1-u) = \sum_{s=0}^{\kappa} c_s u^s$.*

**Proposition 1** (Properties of $p_\kappa$). *For $\lambda \in [0,1]$:*

- **Positivity.** $p_\kappa(\lambda) > 0$ and $p_\kappa(\lambda) \geq 1$ with equality iff $\lambda = 1$.

- **Monotonicity of $\tau$.** Let $\tau(\lambda) := \lambda[p_\kappa(\lambda)]^2$, then we have $\tau$ non-decreasing on $[0,1]$ and $\tau(1) = 1$.

*Consequently, for any symmetric $A \succeq 0$ with spectrum in $[0,1]$, the* NEWTON–SCHULZ *update $A \mapsto p_\kappa(A)Ap_\kappa(A)$ satisfies $\|p_\kappa(A)Ap_\kappa(A)\|_{\mathrm{op}} \leq 1$:* NEWTON–SCHULZ *steps preserve the unit spectral ball (see Appendix A.3). Moreover, the property of the function $\tau$ is used when proving how fast does one step of* NEWTON–SCHULZ *make the momentum orthogonal (Lemma 2).*

In order to quantify the degree of orthogonality of the output matrix $X_{t,q}$ after $q$ steps of NEWTON–SCHULZ, and to measure the approximation error derived from NEWTON–SCHULZ compared to the exact-polar method via SVD under operator-norm, we define the following:

**Definition 3** (Orthogonality residual and polar approximation error). *For fixed $t$, let $\Pi_t$ be the orthogonal projector onto $\mathrm{range}(M_t)$. With $\{X_{t,j}\}_{j=0}^{q}$ from Algorithm 1, define the orthogonality residual $\delta_{t,j}$ and the polar approximation error $\varepsilon_{t,q}$ by*

$$\delta_{t,j} := \|\Pi_t - X_{t,j}X_{t,j}^\top\|_{\mathrm{op}} \in [0,1), \qquad \varepsilon_{t,q} := \|X_{t,q} - \mathrm{Polar}(M_t)\|_{\mathrm{op}}.$$

*Define $\varepsilon_q := \sup_t \varepsilon_{t,q}$ and $\delta_0 := \sup_t \delta_{t,0}$.*

*Remark 1.* In MUON, there is a scaling step for the momentum matrix before applying it to the recursive update by the NEWTON–SCHULZ polynomial (Line 5 in Algorithm 1). This scaling ensures $\|X_{t,0}\|_{\mathrm{op}} \leq 1$, which is required to apply the NEWTON–SCHULZ polynomial properties described in Proposition 1. In parallel, the initial residual is strictly less than 1, i.e., $\delta_{t,0} \in [0,1)$ for every iteration $t$ (see Appendix D).

## 4 Main Results

We begin by stating our two main theorems: (i) a nonconvex convergence rate for MUON with a finite number of NEWTON–SCHULZ steps (Theorem 1); and (ii) an explicit bound on the multiplicative constant induced by NEWTON–SCHULZ, together with its (doubly) exponential decay in $q$ (Theorem 2). We then provide brief proof sketches for both theorems in Sections 4.3 and 4.4. For comparisons, Section 4.5 also presents convergence rates for the idealized MUON with an exact SVD-based polar step and for SGD with momentum—the vector-based baseline—stated under the same nuclear-norm stationarity metric.

### 4.1 Convergence of MUON (with Newton–Schulz)

**Theorem 1** (Convergence of MUON with NEWTON–SCHULZ). *Suppose Assumptions 1 and 2 hold, and run MUON (Algorithm 1) with initialization $W_0 \in \mathbb{R}^{m \times n}$. Choose the stepsize and momentum*

*as $\eta = \sqrt{\frac{(1-\beta)D}{TL}}$ and $\beta = 1 - \min\left\{\frac{\sqrt{LDB}}{\sigma\sqrt{rT}}, 1\right\}$ where $r = \min\{m, n\}$. Then there exists a factor $\chi_q > 0$, depending only on the number $q$ of* NEWTON–SCHULZ *steps, such that*

$$\frac{1}{T}\sum_{t=1}^{T}\mathbb{E}\big[\|\nabla f(W_{t-1})\|_*\big] \leq \chi_q \cdot \mathcal{O}\left(\left(\frac{r\sigma^2 LD}{BT}\right)^{1/4} + \sqrt{\frac{LD}{T}} + \frac{\sigma r}{\sqrt{BT}}\right)$$

*Consequently,* MUON *with* NEWTON–SCHULZ *attains $\epsilon$-stationarity with an iteration complexity of $T = \mathcal{O}\left(\max\left\{\frac{\chi_q^2 LD}{\epsilon^2}, \frac{\chi_q^2 r^2\sigma^2}{B\epsilon^2}, \frac{\chi_q^4 r\sigma^2 LD}{B\epsilon^4}\right\}\right)$ iterations.*

**Discussions of Theorem 1.** Theorem 1 guarantees that MUON with NEWTON–SCHULZ converges to an $\epsilon$-stationary point. To the best of our knowledge, this convergence guarantee is the first result for MUON with NEWTON–SCHULZ. Moreover, as shown later by comparison with the SVD-based polar variant, the iteration complexity of MUON with NEWTON–SCHULZ matches the exact-polar rate up to a multiplicative factor $\chi_q$ that depends on the polar-approximation error $\varepsilon_q$ (e.g., Table 1). Crucially, we show later in Theorem 2 that $\chi_q \to 1$ converges at an exponential rate in the number of NEWTON–SCHULZ steps $q$, so the convergence gap (in the number of iterations) to the ideal SVD-polar rate can be made arbitrarily small. Since each NEWTON–SCHULZ step is substantially cheaper than an SVD, these results provide the first theoretical explanation for the superior practical performance observed for the original (SVD-free) MUON.

## 4.2 Decay Rate of $\varepsilon_q$ and Convergence Rate of $\chi_q \to 1$

We now quantify how fast $\chi_q$ approaches 1. For a given number $q$ of NEWTON–SCHULZ steps, we can show that the polar-approximation error $\varepsilon_q$ decays *doubly exponentially* in $q$ (with faster decay for larger $\kappa$). Since $\chi_q$ is controlled by $\varepsilon_q$, it follows that $\chi_q \to 1$ is at the same rate. The following theorem formalizes this result.

**Theorem 2** (Upper-bounds on $\varepsilon_q$ and $\chi_q$). *For the* NEWTON–SCHULZ *polynomial with degree $\kappa$ and for any $t$, $\delta_{t,q} \leq \delta_{t,0}^{(\kappa+1)^q}$. Hence, the bound of the polar approximation error $\varepsilon_q$ and the factor $\chi_q$ occurred by* NEWTON–SCHULZ *is*

$$\varepsilon_q \leq 1 - \sqrt{1 - \delta_0^{(\kappa+1)^q}} \leq \delta_0^{(\kappa+1)^q}, \qquad \chi_q = \frac{1}{1-\varepsilon_q} \leq \frac{1}{\sqrt{1-\delta_0^{(\kappa+1)^q}}},$$

*where $\delta_0 := \sup_t \delta_{t,0} < 1$.*

**Discussion of Theorem 2.** The theorem shows that $\varepsilon_q$ is bounded by $\delta_0^{(\kappa+1)^q}$. Hence, $\varepsilon_q$ vanishes *doubly exponentially* in $q$ (and improves with larger $\kappa$). Therefore, $\chi_q \to 1$ at the same doubly exponential rate. Together with Theorem 1, this result implies that the iteration-complexity gap between NEWTON–SCHULZ and the idealized SVD-polar update becomes negligible after only a few NEWTON–SCHULZ steps.

**Practical implication.** A finite number of NEWTON–SCHULZ steps yields iteration complexity essentially indistinguishable from exact SVD updates (up to the factor $\chi_q \to 1$ doubly exponentially fast), while dramatically reducing per-iteration cost by using only matrix multiplications.

## 4.3 Proof Sketch for Theorem 1.

We briefly outline the main ideas. First, introduce the scaled momentum $N_t = (1-\beta)M_t$ (faithfully following the original update rule), so the EMA becomes $N_t \leftarrow \beta N_{t-1} + (1-\beta)G_t$. Next, apply the *descent lemma* (Lemma 6) to the update $W_t \leftarrow W_{t-1} - \eta O_t$, which yields a term of the form $\langle \nabla f(W_{t-1}), O_t\rangle_F$. Decompose this inner product as

$$\langle \nabla f(W_{t-1}), O_t\rangle_F = \langle N_t, O_t\rangle_F + \langle \nabla f(W_{t-1}) - N_t, O_t\rangle_F,$$

thereby isolating the momentum mismatch. Prior analyses typically stop here and average over iterations. In contrast, we further split $\langle N_t, O_t\rangle_F$ as

$$\langle N_t, O_t\rangle_F = \langle N_t, P_t\rangle_F + \langle N_t, O_t - P_t\rangle_F,$$

Table 1: Comparison of convergence rates.

| Method | Convergence rate |
|---|---|
| SGD with momentum (Theorem 3) | $\mathcal{O}\left(\left(\frac{r^2\sigma^2 LD}{BT}\right)^{1/4} + \sqrt{\frac{rLD}{T}}\right)$ |
| MUON with SVD (Theorem 4) | $\mathcal{O}\left(\left(\frac{r\sigma^2 LD}{BT}\right)^{1/4} + \sqrt{\frac{LD}{T}} + \frac{\sigma r}{\sqrt{BT}}\right)$ |
| MUON (Theorem 1) | $\chi_q \cdot \mathcal{O}\left(\left(\frac{r\sigma^2 LD}{BT}\right)^{1/4} + \sqrt{\frac{LD}{T}} + \frac{\sigma r}{\sqrt{BT}}\right)$ |

$D = f(W_0) - f^*$, $r = \min\{m, n\}$, iterations $T$, batch size $B$, Lipschitz constant $L$, variance bound $\sigma^2$

The factor $\chi_q$ converges to 1 at exponential rate in step $q$. (Theorem 2): $\chi_q \leq [1 - \delta_0^{(\kappa+1)^q}]^{-1/2}$, where $q$ is the number of NEWTON–SCHULZ step and $\kappa$ is the degree of the NEWTON–SCHULZ polynomial (Def. 2).

separating the exact polar factor $P_t$ from the NEWTON–SCHULZ orthogonalizer (the output matrix of the NEWTON–SCHULZ routine). As we define the *polar approximation error* $\varepsilon_q$ as the discrepancy between the exact polar factor $P_t = \text{Polar}(N_t) = \text{Polar}(M_t)$ and the actual step $O_t$ produced by $q$ steps of NEWTON–SCHULZ, we can control this part with respect to $\varepsilon_q$. This yields a one-step descent inequality for MUON that explicitly includes the NEWTON–SCHULZ-induced error $\varepsilon_q$. Averaging this inequality over $t = 1, \ldots, T$ and choosing $\eta$ and $\beta$ as specified in Theorem 1 produces the stated convergence rate. Full details appear in Appendix B.

## 4.4 PROOF SKETCH FOR THEOREM 2

We outline how Theorem 2 follows; full proofs appear in Appendix D. Recall that in Theorem 1, the convergence rate contains the polar-approximation error $\varepsilon_q$ and the multiplicative factor $\chi_q$ depending on $\varepsilon_q$. To bound the resulting multiplicative factor $\chi_q$, we relate $\varepsilon_q$ to an *orthogonality residual* that quantifies how close the NEWTON–SCHULZ iterate is to having orthonormal columns. Concretely, letting $X_{t,q}$ be the output matrix after $q$ NEWTON–SCHULZ steps applied to the (scaled) momentum at iteration $t$, define $\delta_{t,q} := \|\Pi_t - X_{t,q}X_{t,q}^\top\|_{\text{op}}$.

The next lemma provides the spectral link between the residual and the polar-approximation error:

**Lemma 1** (Orthogonality residual vs. Polar approximation error). *Let $\lambda_{\min}^+$ be the smallest positive eigenvalue of $X_{t,q}X_{t,q}^\top$ restricted to $\text{range}(M_t)$ (set $\lambda_{\min}^+ = 1$ if $\text{rank}(M_t) = 0$). Then*

$$\delta_{t,q} = 1 - \lambda_{\min}^+, \qquad \varepsilon_{t,q} = 1 - \sqrt{\lambda_{\min}^+} = 1 - \sqrt{1 - \delta_{t,q}}.$$

Next, we describe how a single NEWTON–SCHULZ step transforms the residual through the degree-$\kappa$ polynomial $p_\kappa$:

**Lemma 2** (Residual update). *For NEWTON–SCHULZ polynomial $p_\kappa$, the orthogonality residual $\delta_{t,j}$ is updated by NEWTON–SCHULZ per step as*

$$\delta_{t,j+1} = \phi(\delta_{t,j}),$$

*where $\phi(u) := 1 - (1 - u)\left[p_\kappa(1 - u)\right]^2 = 1 - \tau(1 - u)$.*

To prove Lemma 2, we introduce $\tau(\lambda) := \lambda\left[p_\kappa(\lambda)\right]^2$, note that $\tau$ is non-decreasing on $[0, 1]$ and satisfies $\tau(1) \leq 1$ (Proposition 1), and then translate this monotonicity to the residual map $\phi$.

Finally, we obtain a contraction bound and its multi-step consequence:

**Lemma 3** (Residual decay by NEWTON–SCHULZ polynomial). *For NEWTON–SCHULZ polynomial $p_\kappa$, $\phi(u) \leq u^{\kappa+1}$ on $[0, 1]$ where $\phi$ is a function defined in Lemma 2. Hence, for every $t$ and all $j \geq 0$,*

$$\delta_{t,j+1} \leq \delta_{t,j}^{\kappa+1}, \qquad \delta_{t,q} \leq \delta_{t,0}^{(\kappa+1)^q}.$$

## 4.5 COMPARISONS WITH SGD WITH MOMENTUM AND MUON WITH SVD

To enable a fair comparison with the original MUON with NEWTON–SCHULZ method, we also establish convergence guarantees for two baselines: (i) *SGD with momentum* and (ii) the *idealized*

MUON *with an exact polar step computed by SVD*. All results are derived under the same smoothness and bounded-variance assumptions, and progress is measured by the nuclear norm of the gradient, so the rates are directly comparable.

The convergence bound in Theorem 3 for *SGD with momentum* (Theorem 3) is, to our knowledge, new and may be of independent interest. It provides a clean reference point for vector-based updates under the same stationarity metric.

**Theorem 3** (Convergence of SGD with momentum). *Suppose Assumptions 1 and 2 hold, and run SGD with momentum. Choosing* $\eta = \min\left\{\frac{1-\beta}{L}, \frac{(1-\beta)^2}{4L}\right\}$ *and* $\beta = 1 - \min\{\frac{\sqrt{LDB}}{\sigma\sqrt{T}}, 1\}$*, the following holds*

$$\frac{1}{T}\sum_{t=1}^{T}\mathbb{E}\|\nabla f(W_{t-1})\|_* \leq \mathcal{O}\left(\left(\frac{r^2\sigma^2 LD}{BT}\right)^{1/4} + \sqrt{\frac{rLD}{T}}\right),$$

*Consequently,* SGD *with momentum attains* $\epsilon$*-stationarity with an iteration complexity of* $\mathcal{O}\left(\max\left\{\frac{rLD}{\epsilon^2}, \frac{r^2\sigma^2 LD}{B\epsilon^4}\right\}\right)$.

Note that the rate for MUON *with SVD* (Theorem 4) is obtained as a special case of MUON with NEWTON–SCHULZ analysis by setting the polar-approximation error to zero (i.e., replacing NEWTON–SCHULZ with an exact polar step). This "oracle" baseline clarifies the gap that NEWTON–SCHULZ needs to close in practice and is useful for interpreting the effect of a finite number of NEWTON–SCHULZ steps.

**Theorem 4** (MUON with SVD). *Under Assumptions 1 and 2, setting* $\varepsilon_q = 0$ *in Theorem 1 yields*

$$\frac{1}{T}\sum_{t=1}^{T}\mathbb{E}\|\nabla f(W_{t-1})\|_* \leq \mathcal{O}\left(\left(\frac{r\sigma^2 LD}{BT}\right)^{1/4} + \sqrt{\frac{LD}{T}} + \frac{\sigma r}{\sqrt{BT}}\right),$$

*Consequently, the idealized* MUON *with* SVD *attains* $\epsilon$*-stationarity with iteration complexity of* $\mathcal{O}\left(\max\left\{\frac{LD}{\epsilon^2}, \frac{r^2\sigma^2}{B\epsilon^2}, \frac{r\sigma^2 LD}{B\epsilon^4}\right\}\right)$

**Discussion of Theorems 3 and 4.** Relative to SGD with momentum, the SVD-based polar variant of MUON sharpens the rank dependence in the leading $\mathcal{O}(1/T^{1/4})$ term from $r^{1/2}$ to $r^{1/4}$ under the same nuclear-norm stationarity metric. Geometrically, the polar step aligns the update with the leading singular structure of the gradient (via spectral–nuclear duality), converting a Frobenius-aligned descent direction into one that is optimally aligned for the nuclear norm, thereby sharpening the $r$ dependence.

Turning to the original MUON with NEWTON–SCHULZ, Theorem 1 shows that its iteration complexity matches that of the SVD-based polar variant (Theorem 4) up to a multiplicative factor $\chi_q$. By Theorem 2, $\chi_q \to 1$ *doubly exponentially* fast in the number of NEWTON–SCHULZ steps $q$ (and improves with the polynomial degree $\kappa$). Consequently, even a small $q$ yields rates that are essentially indistinguishable from the ideal SVD-based baseline in terms of iteration complexity. Since each NEWTON–SCHULZ step uses only matrix multiplications (and avoids SVD), the per-iteration cost is much lower, which explains the superior wall-clock performance of the original SVD-free MUON observed in practice. Moreover, because MUON with a few NEWTON–SCHULZ steps converges rapidly to the SVD-based variant, the sharper rank dependence established for the SVD-based MUON—which provides a comparative advantage over SGD with momentum—effectively carries over to MUON with NEWTON–SCHULZ, while preserving its computational efficiency.

## 5 NUMERICAL EXPERIMENTS

**Setup.** We conduct a numerical experiment with the CIFAR-10 (50k/10k) dataset and a CNN model, specifically CIFARNET, which has approximately 2M parameters. We compare optimizers: SGD with momentum (baseline), idealized MUON with SVD, and MUON with NEWTON–SCHULZ ($q \in \{1, 2, 3\}$). For the NEWTON–SCHULZ step sweep, we use the NEWTON–SCHULZ polynomial $p_\kappa$ with degree $\kappa = 2$. We run 50 epochs, and the batch size is $B = 512$. Results are plotted in Fig. 1.

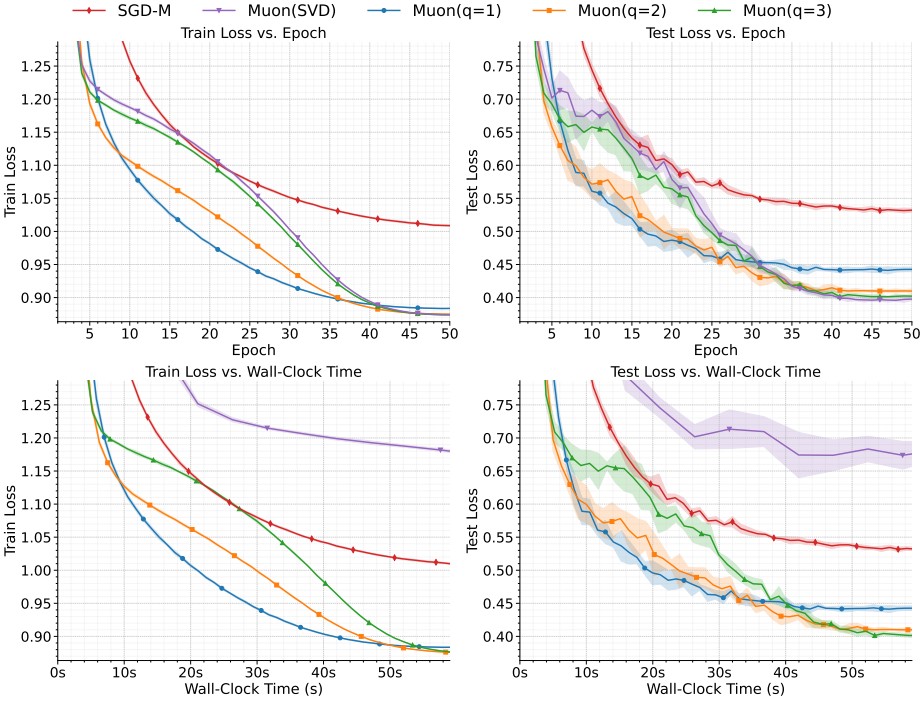

Figure 1: **NEWTON–SCHULZ steps** ($q$) **ablation.** MUON with NEWTON–SCHULZ for $q \in \{1, 2, 3\}$ vs. MUON (SVD) and SGD with momentum (SGD-M, baseline).

Performance is assessed by plotting the training loss (left column) and test loss (right column) over epochs (top row) as well as the cumulative wall-clock time (bottom row). Results represent the average of five runs with different random seeds, including standard deviations. More detailed numerical settings are described in Appendix F.1.

**Ablation on NEWTON–SCHULZ step $q$.** As $q$ increases, the learning dynamics per epoch steadily improve: MUON with $q = 1$ already outperforms SGD-M, and $q \in \{2, 3\}$ nearly coincides with the SVD-based MUON in both train loss and test loss, in line with Theorems 1 and 2, which state that the NEWTON–SCHULZ variant matches the SVD iteration complexity up to a factor $\chi_q \to 1$ that decays doubly exponentially in $q$. At the same time, the bottom row of Fig. 1 shows that MUON with $q = 2$ or 3 reaches a given test loss substantially faster in wall-clock time than the SVD variant, reflecting the lower per-iteration cost of the NEWTON–SCHULZ update.

We additionally performed numerical experiments on various datasets using models at different scales (with the number of parameters indicated in parentheses): a multilayer perceptron (MLP) with 0.5M parameters on MNIST; ResNet-18 (11.2M) on CIFAR-100; WideResNet-28-10 (36.6M) on Tiny-ImageNet; NanoGPT (124M, Transformer) on FineWeb; and a GPT-2–based model (1.3B, Transformer) on FineWeb. All additional experimental results are presented in Appendix G.

**Ablation on NEWTON–SCHULZ polynomial degree $\kappa$.** We perform a controlled ablation that varies the *degree* of the NEWTON–SCHULZ polynomial $\kappa$ while fixing the number of NEWTON–SCHULZ steps to $q = 3$ for all variants. Increasing the degree $\kappa \in \{1, \ldots, 5\}$ improves optimization (the loss drops faster at a fixed epoch) but lengthens each step, yielding a clear accuracy–time trade-off (See Appendix H.1). This mirrors the theory that the residual contracts as $\delta_{j+1} \leq \delta_j^{\kappa+1}$ (Lemma 3), while computation scales with polynomial evaluations.

**Rank dependence.** We vary the monitored layer's effective rank $r \in \{16, 32, 64, 128, 216\}$ and plot the epoch-averaged $\|\nabla f(W)\|_*$ on a log–log scale. SGD-M shows a positive slope of approximately 0.3 (grows with $r$), whereas MUON and its variants are nearly flat. These observations are precisely in line with Theorem 3 and Theorem 4, which state that orthogonalizing momentum reduces the rank

$r$ dependence of the leading $\mathcal{O}(1/T^{1/4})$ term from $r^{1/2}$ to $r^{1/4}$. The more detailed experimental settings and results are deferred to Appendix H.2.

## 6 CONCLUSION

We analyzed practical MUON with finite NEWTON–SCHULZ steps and proved nonconvex convergence to $\epsilon$-stationarity with iteration complexity matching the SVD-polar idealization, up to a factor $\chi_q$ that shrinks doubly exponentially in $q$ (and improves with $\kappa$). Thus, a few NEWTON–SCHULZ steps recover the idealized rate while remaining SVD-free and GPU-friendly. We also provided baselines for SGD with momentum and the SVD-polar variant, showing that MUON weakens rank dependence under the same metric —closing the theory–practice gap and explaining its practical performance.

## USE OF LARGE LANGUAGE MODELS

Large Language Models (LLMs) are used solely as assistive tools for writing. Specifically, we employed an LLM to improve the clarity, grammar, and style of exposition. No part of the research ideation, algorithm design, theoretical analysis, or experimental results involved the use of LLMs. The authors take full responsibility for the content of the paper.

### ACKNOWLEDGMENTS

This work was supported by the National Research Foundation of Korea (NRF) grant and the Institute of Information & communications Technology Planning & Evaluation (IITP) grant both funded by the Korea government (MSIT) (No. RS-2022-NR071853, RS-2023-00222663, RS-2025-25463302) and by AI-Bio Research Grant through Seoul National University.

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

## A  APPENDIX

### A.1  BASIC FACTS FOR MATRIX NORMS

**Definition 4** (Schatten Norms). *For each $p \in [1, \infty]$, the Schatten-$p$ norm is defined as $\|X\|_{S_p} := (\sum_i \sigma_i(X)^p)^{1/p}$, where $\sigma_i(X)$ are the singular values of $X$.*

- *Nuclear/trace norm: $\|X\|_{S_1} = \|X\|_* = \sum_i \sigma_i(X)$ (Sum of singular values)*

- *spectral/operator norm: $\|X\|_{S_\infty} = \|X\|_{\mathrm{op}} = \max_i \sigma_i(X)$ (largest singular value)*

- *Frobenius norm: $\|X\|_{S_2} = \|X\|_F = \sqrt{\sum_i \sigma_i(X)^2}$*

*For any conjugate pairs $(p, q)$, i.e., $\frac{1}{p} + \frac{1}{q} = 1$, the norms $\|\cdot\|_{S_p}$ and $\|\cdot\|_{S_q}$ are dual to each other. In particular, the nuclear norm and the spectral norm are duals.*

**Lemma 4** (Hölder's inequality). *For any $A, B \in \mathbb{R}^{m \times n}$,*

$$|\langle A, B\rangle_F| \ \leq \ \|A\|_* \|B\|_{\mathrm{op}}.$$

*Proof.* Let $\mathrm{SVD}(A) = (U, \Sigma, V)$.

$$\langle A, B\rangle_F = \mathrm{tr}(A^\top B) = \mathrm{tr}((U\Sigma V^\top)^\top B) = \mathrm{tr}(V\Sigma U^\top B) = \mathrm{tr}(\Sigma U^\top BV)$$

Let's define a new matrix $C = U^\top BV$. Since $U^\top$ and $V$ are orthogonal matrices, their operator norm is 1.

$$\|C\|_{\mathrm{op}} = \|U^\top BV\|_{\mathrm{op}} \leq \|U^\top\|_{\mathrm{op}} \|B\|_{\mathrm{op}} \|V\|_{\mathrm{op}} = 1 \cdot \|B\|_{\mathrm{op}} \cdot 1 = \|B\|_{\mathrm{op}}$$

Since $\Sigma$ is diagonal, $\mathrm{tr}(\Sigma C) = \sum_i \sigma_i(A) C_{ii}$ where $\sigma_i(A)$ is the $i$-th singular value of $A$.

$$|\langle A, B\rangle_F| = |\mathrm{tr}(\Sigma C)| = \left| \sum_i \sigma_i(A) C_{ii} \right| \leq \sum_i \sigma_i(A) |C_{ii}| \leq \sum_i \sigma_i(A) \|C\|_{\mathrm{op}} = \|A\|_* \|C\|_{\mathrm{op}}$$

Combining inequalities, we have $|\langle A, B\rangle_F| \leq \|A\|_* \|B\|_{\mathrm{op}}$. $\qquad\square$

**Lemma 5.** *For any $A, B \in \mathbb{R}^{m \times n}$ and any constant $\beta \in (0, 1)$,*

$$\|A + B\|_F^2 \ \leq \ \frac{1}{\beta} \|A\|_F^2 + \frac{1}{1 - \beta} \|B\|_F^2.$$

*Proof.* By Young's inequality (i.e., $\langle x, y\rangle_F \leq \frac{\epsilon}{2}\|x\|_F^2 + \frac{1}{2\epsilon}\|y\|_F^2$ with $\epsilon > 0$), for any positive constant $c > 0$,

$$\|A + B\|_F^2 = \|A\|_F^2 + \|B\|_F^2 + 2\langle A, B\rangle_F$$
$$\leq \|A\|^2 + \|B\|_F^2 + c\|A\|_F^2 + \frac{1}{c}\|B\|_F^2$$
$$= (1 + c)\|A\|_F^2 + \left(1 + \frac{1}{c}\right)\|B\|_F^2$$

If we choose $c = \frac{1 - \beta}{\beta}$ where $\beta \in (0, 1)$, we get

$$\|A + B\|_F^2 \leq \frac{1}{\beta}\|A\|_F^2 + \frac{1}{1 - \beta}\|B\|_F^2$$

$$\square$$

**Proposition 2.** *Let $X \in \mathbb{R}^{m \times n}$ with $r = \min\{m, n\}$. Denote that its singular values are $\{\sigma_i\}_{i=1}^r$. Then the following holds:*

*(i) $\|X\|_F = \sqrt{\mathrm{tr}(X^\top X)} = \sqrt{\sum_{ij} X_{i,j}^2}$*

*Proof.* Let $\mathrm{SVD}(X) = (U, \Sigma, V)$. Then

$$\mathrm{tr}(X^\top X) = \mathrm{tr}\left((U\Sigma V^\top)^\top(U\Sigma V^\top)\right) = \mathrm{tr}(V\Sigma^\top U^\top U\Sigma V^\top) = \mathrm{tr}(V\Sigma^\top \Sigma V^\top)$$
$$= \mathrm{tr}(V^\top V\Sigma^\top \Sigma) = \mathrm{tr}(\Sigma^\top \Sigma) = \sum_i \sigma_i(X)^2 = \|X\|_F^2$$

Since $(X^\top)_{ji} = X_{ij}$, we have

$$(X^\top X)_{jj} = \sum_{i=1}^m (X^\top)_{ji} X_{ij} = \sum_{i=1}^m X_{ij}^2$$

Hence, $\mathrm{tr}(X^\top X) = \sum_{j=1}^n (X^\top X)_{jj} = \sum_{j=1}^n \sum_{i=1}^m X_{ij}^2$ ☐

(ii) *If $P = \mathrm{Polar}(X)$, then $\langle X, P \rangle_F = \|X\|_*$.*

*Proof.* Let $\mathrm{SVD}(X) = (U, \Sigma, V)$. Then $P = \mathrm{Polar}(X) = UV^\top$.

$$\langle X, P \rangle_F = \langle U\Sigma V^\top, UV^\top \rangle_F = \mathrm{tr}((U\Sigma V^\top)^\top UV^\top) = \mathrm{tr}(V\Sigma U^\top UV^\top)$$
$$= \mathrm{tr}(V\Sigma V^\top) = \mathrm{tr}(V^\top V\Sigma) = \mathrm{tr}(\Sigma) = \sum_i \sigma_i(X) = \|X\|_*$$

☐

(iii) $\|X\|_* \leq \sqrt{r}\,\|X\|_F$.

*Proof.* By Cauchy-Schwarz inequality,

$$\|X\|_* = \sum_{i=1}^r \sigma_i(X) \leq \sqrt{r \sum_{i=1}^r \sigma_i(X)^2} = \sqrt{r}\|X\|_F$$

☐

(iv) $\|X\|_{\mathrm{op}} \leq \|X\|_F \leq \|X\|_*$.

*Proof.* Since singular values are always non-negative ($\sigma_i(X) \geq 0$),

$$\left(\max_i \sigma_i(X)\right)^2 \leq \sum_i \sigma_i(X)^2 \leq \left(\sum_i \sigma_i(X)\right)^2$$

Taking the square root, we get $\|X\|_{\mathrm{op}} \leq \|X\|_F \leq \|X\|_*$. ☐

(v) *The polar factor is scale-invariant:* $\mathrm{Polar}(cX) = \mathrm{Polar}(X)$ *for all $c > 0$.*

*Proof.* Let $\mathrm{SVD}(X) = (U, \Sigma, V)$. Then $P = \mathrm{Polar}(X) = UV^\top$. Also, $\mathrm{SVD}(cX) = (U, c\Sigma, V)$, so $\mathrm{Polar}(cX) = UV^\top$. Thus, $\mathrm{Polar}(cX) = \mathrm{Polar}(X)$ for all $c > 0$. ☐

## A.2 LEMMAS UNDER ASSUMPTIONS

### A.2.1 ASSUMPTION 1

**Definition 5** (*$L$-smoothness*). *A differentiable function $f : \mathbb{R}^{m \times n} \to \mathbb{R}$ is $L$-smooth if its gradient is $L$-Lipschitz continuous, i.e., for all $X, Y \in \mathbb{R}^{m \times n}$,*

$$\|\nabla f(X) - \nabla f(Y)\|_* \leq L\|X - Y\|_{\mathrm{op}}$$

*We use smoothness with respect to the operator norm (and the nuclear norm as its dual).*

**Lemma 6** (Descent Lemma). *Let $f : \mathbb{R}^{m \times n} \to \mathbb{R}$ be $L$-smooth. Then, for all $X, Y \in \mathbb{R}^{m \times n}$,*

$$f(Y) \leq f(X) + \langle \nabla f(X), Y - X \rangle_F + \frac{L}{2} \|Y - X\|_{\mathrm{op}}^2.$$

*Proof.* Define an auxiliary scalar function $g : [0, 1] \to \mathbb{R}$ by considering the value of $f$ along the line segment from $X$ to $Y$:

$$g(s) := f(X + s(Y - X))$$

By the fundamental theorem of calculus,

$$f(Y) - f(X) = g(1) - g(0) = \int_0^1 g'(s) \, ds$$

Using the chain rule, $g'(s) = \langle \nabla f(X + s(Y - X)), Y - X \rangle_F$. Substituting this back into the integral equation gives:

$$f(Y) - f(X) = \int_0^1 \langle \nabla f(X + s(Y - X)), Y - X \rangle_F ds$$

Add and subtract $\langle \nabla f(X), Y - X \rangle_F = \int_0^1 \langle \nabla f(X), Y - X \rangle_F ds$, we have:

$$f(Y) - f(X) = \langle \nabla f(X), Y - X \rangle_F + \int_0^1 \langle \nabla f(X + s(Y - X)) - \nabla f(X), Y - X \rangle_F ds$$

We now use the generalized Cauchy-Schwarz inequality or Hölder's inequality (Lemma 4), which states that for any matrices $A, B \in \mathbb{R}^{m \times n}$ and $|\langle A, B \rangle_F| \leq \|A\|_* \|B\|_{\mathrm{op}}$, the nuclear norm ($\| \cdot \|_*$) is the dual norm to the operator norm ($\| \cdot \|_{\mathrm{op}}$). Applying this to the integrand:

$$\langle \nabla f(X + s(Y - X)) - \nabla f(X), Y - X \rangle_F \leq \|\nabla f(X + s(Y - X)) - \nabla f(X)\|_* \|Y - X\|_{\mathrm{op}}$$

By the $L$-smoothness of $f$:

$$\|\nabla f(X + s(Y - X)) - \nabla f(X)\|_* \leq L\|s(Y - X)\|_{\mathrm{op}} = Ls\|Y - X\|_{\mathrm{op}}$$

Combining these inequalities, we obtain a bound on the integrand, yielding the final form:

$$f(Y) - f(X) \leq \langle \nabla f(X), Y - X \rangle_F + \int_0^1 Ls\|Y - X\|_{\mathrm{op}}^2 ds = \langle \nabla f(X), Y - X \rangle_F + \frac{L}{2}\|Y - X\|_{\mathrm{op}}^2$$

$\square$

**Lemma 7** (Gradient-gap inequality). *Under Assumption 1, for any $W$, we have*

$$\|\nabla f(W)\|_F^2 \leq 2L \left( f(W) - f^* \right)$$

*Proof.* In the descent lemma (Lemma 6), let $Y = X - \frac{1}{L}\nabla f(X)$. Then, we have

$$f\left(X - \frac{1}{L}\nabla f(X)\right) \leq f(X) + \left\langle \nabla f(X), X - \frac{1}{L}\nabla f(X) - X \right\rangle_F + \frac{L}{2} \left\| X - \frac{1}{L}\nabla f(X) - X \right\|_{\mathrm{op}}^2$$

$$= f(X) - \frac{1}{L}\|\nabla f(X)\|_F^2 + \frac{1}{2L}\|\nabla f(X)\|_{\mathrm{op}}^2$$

$$\leq f(X) - \frac{1}{L}\|\nabla f(X)\|_F^2 + \frac{1}{2L}\|\nabla f(X)\|_F^2$$

$$= f(X) - \frac{1}{2L}\|\nabla f(X)\|_F^2$$

where the last inequality is due to Proposition 2(iv). Since $f^* \leq f\left(X - \frac{1}{L}\nabla f(X)\right)$, we have

$$\|\nabla f(W)\|_F^2 \leq 2L \left( f(W) - f^* \right)$$

$\square$

### A.2.2 ASSUMPTION 2

**Lemma 8** (Unbiasedness and bounded variance). *If Assumption 2 holds, then*

$$\mathbb{E}\left[\|G_t - \nabla f(W_t)\|_F^2\right] \leq \frac{\sigma^2}{B} \quad \text{and} \quad \mathbb{E}[\|G_t - \nabla f(W_t)\|_F] \leq \frac{\sigma}{\sqrt{B}}.$$

*where $B$ is the batch size.*

*Proof.* Since $G_t = \frac{1}{B}\sum_{i=1}^{B} g_i$ with $g_i = \nabla f(W_t; \xi_{t,i})$,

$$\mathbb{E}\left[\|G_t - \nabla f(W_t)\|_F^2\right] = \mathbb{E}\left[\left\|\frac{1}{B}\sum_{i=1}^{B}(\nabla f(W_t; \xi_{t,i}) - \nabla f(W_t))\right\|_F^2\right]$$

$$= \frac{1}{B^2}\mathbb{E}\left[\left\|\sum_{i=1}^{B}(\nabla f(W_t; \xi_{t,i}) - \nabla f(W_t))\right\|_F^2\right]$$

$$= \frac{1}{B^2}\sum_{i=1}^{B}\mathbb{E}\left[\|\nabla f(W_t; \xi_{t,i}) - \nabla f(W_t)\|_F^2\right] \leq \frac{\sigma^2}{B}.$$

where the last equality is due to i.i.d. uniform sampling with $\mathbb{E}[\nabla f(W_t; \xi_{t,i})] = \nabla f(W_t)$ and independence. The last inequality is due to $\mathbb{E}[\|\nabla f(W; \xi_{t,i}) - \nabla f(W)\|_F^2] \leq \sigma^2$. According to Jensen's inequality, we have

$$\mathbb{E}[\|G_t - \nabla f(W_t)\|_F] \leq \sqrt{\mathbb{E}[\|G_t - \nabla f(W_t)\|_F^2]} = \frac{\sigma}{\sqrt{B}}.$$

$\square$

## A.3 NEWTON–SCHULZ POLYNOMIAL

NEWTON–SCHULZ steps orthogonalize a matrix $X$ via NEWTON–SCHULZ steps applied to $A = XX^\top$. Define NEWTON–SCHULZ polynomial $p_\kappa$ the degree is $\kappa$. The following are the properties of $p_\kappa$ along with their proofs.

**NEWTON–SCHULZ polynomial $p_\kappa$.**

For degree $\kappa \in \mathbb{N}$, the NEWTON–SCHULZ polynomial is the Taylor truncation of $1/\sqrt{\lambda}$ at $\lambda = 1$, i.e.,

$$p^{(s)}(1) = \frac{d^s}{d\lambda^s}\lambda^{-1/2}\bigg|_{\lambda=1}$$

for $s = 1, \ldots, \kappa$. The explicit form of the NEWTON–SCHULZ polynomial for degree $\kappa$ is

$$p_\kappa(\lambda) = \sum_{s=0}^{\kappa} c_s(1-\lambda)^s, \qquad c_s = \frac{(2s)!}{4^s(s!)^2} > 0.$$

Equivalently, with reparametrization $u = 1 - \lambda \in [0, 1]$ and $p_\kappa(1 - u) = \sum_{s=0}^{\kappa} c_s u^s$.

**Proposition 1 (Properties of $p_\kappa$).** For $\lambda \in [0, 1]$:

- **Positivity.** $p_\kappa(\lambda) > 0$ and $p_\kappa(\lambda) \geq 1$ with equality if and only if $\lambda = 1$.
- **Monotonicity of $\tau$.** Let $\tau(\lambda) := \lambda[p_\kappa(\lambda)]^2$; then we have $\tau$ non-decreasing on $[0, 1]$ and $\tau(1) = 1$.

Consequently, for any symmetric $A \succeq 0$ with spectrum in $[0, 1]$, the NEWTON–SCHULZ update $A \mapsto p_\kappa(A)Ap_\kappa(A)$ satisfies $\|p_\kappa(A)Ap_\kappa(A)\|_{\mathrm{op}} \leq 1$: NEWTON–SCHULZ steps preserve the unit spectral ball (see Appendix A.3).

*proof.*

**Positivity.** $p_\kappa(\lambda) > 0$ and $p_\kappa(\lambda) \geq 1$ with equality if and only if $\lambda = 1$.

*Proof.* Separating the first term (for $s = 0$) from the summation that defines $p_\kappa(\lambda)$:

$$p_\kappa(\lambda) = \sum_{s=0}^{\kappa} c_s(1-\lambda)^s = c_0(a-\lambda)^0 + \sum_{s=1}^{\kappa} c_s(1-\lambda)^s$$

The coefficient $c_0$ is $\frac{(2\cdot 0)!}{4^0(0!)^2} = 1$. The rest of the polynomial is the sum $\sum_{s=1}^{\kappa} c_s(1-\lambda)^s$. For any term in this sum and for any $\lambda \in [0, 1]$, the coefficients $c_s$ are strictly positive for all $s \geq 1$. Also, for any exponent $s \geq 1$, $(1-\lambda)^s$ is non-negative in the range $[0, 1]$. Each term $c_s(1-\lambda)^s$ is a product of a positive number and a non-negative number, which means each term is non-negative. Hence,

$$p_\kappa(\lambda) = 1 + \sum_{s=1}^{\kappa} c_s(1-\lambda)^s \geq 1 + 0 = 1$$

This proves that $p_\kappa(\lambda) \geq 1$ for all $\lambda \in [0, 1]$. It is also trivially true that $p_\kappa(\lambda) > 0$. $\qquad\square$

**Monotonicity of $\tau(\lambda) := \lambda[p_\kappa(\lambda)]^2$.**

*Proof.* First note $\tau(1) = 1 \cdot p_\kappa(1)^2 = 1$. For monotonicity, differentiate:

$$\tau'(\lambda) = p_\kappa(\lambda)^2 + 2\lambda p_\kappa(\lambda)p_\kappa'(\lambda) = p_\kappa(\lambda)\left(p_\kappa(\lambda) + 2\lambda p_\kappa'(\lambda)\right).$$

Set $u = 1 - \lambda$ and define $q(u) := p_\kappa(1 - u) = \sum_{s=0}^{\kappa} c_s u^s$. Then

$$p_\kappa(\lambda) = q(u), \qquad p_\kappa'(\lambda) = \frac{d}{d\lambda}q(1-\lambda) = -q'(u), \qquad \lambda = 1 - u.$$

Hence

$$\tau'(\lambda) = q(u)\underbrace{\left(q(u) - 2(1-u)q'(u)\right)}_{:=S(u)}.$$

Now, we compute $S(u)$ in closed form. Since

$$q(u) = \sum_{s=0}^{\kappa} c_s u^s, \qquad q'(u) = \sum_{s=1}^{\kappa} s c_s u^{s-1},$$

we obtain

$$S(u) = \sum_{s=0}^{\kappa} c_s u^s - 2(1-u) \sum_{s=1}^{\kappa} s c_s u^{s-1}.$$

Expanding and collecting coefficients of $u^t$ gives

$$S(u) = (c_0 - 2c_1) + \sum_{t=1}^{\kappa-1} \left( (2t+1)c_t - 2(t+1)c_{t+1} \right) u^t + (2\kappa+1)c_\kappa u^\kappa.$$

Using the ratio identity

$$\frac{c_{s+1}}{c_s} = \frac{(2s+2)!(s!)^2 4^s}{(2s)!((s+1)!)^2 4^{s+1}} = \frac{(2s+2)(2s+1)}{4(s+1)^2} = \frac{2s+1}{2(s+1)},$$

for $t = 0, \ldots, \kappa - 1$,

$$(2t+1)c_t - 2(t+1)c_{t+1} = (2t+1)c_t - 2(t+1)\left( \frac{2t+1}{2(t+1)} c_t \right) = 0,$$

and also $c_0 - 2c_1 = 1 - 2 \cdot \frac{1}{2} = 0$. Therefore, all coefficients up to degree $\kappa - 1$ vanish, and we are left with

$$S(u) = (2\kappa+1)c_\kappa u^\kappa \geq 0 \qquad \text{for } u \in [0,1],$$

with strict positivity for $u > 0$ when $\kappa \geq 1$. Since $q(u) > 0$ on $[0,1]$, we conclude

$$\tau'(\lambda) = q(u)S(u) \geq 0 \qquad \text{for all } \lambda \in [0,1],$$

i.e., $\tau$ is non-decreasing on $[0,1]$, and $\tau'(\lambda) > 0$ for $\lambda \in [0,1)$ when $\kappa \geq 1$. Together with $\tau(1) = 1$, this proves the proposition. $\qquad \square$

**NS step preserves the unit spectral ball.** Let $A \succeq 0$ be symmetric with spectrum $\sigma(A) \subset [0,1]$. By spectral calculus,

$$p_\kappa(A)Ap_\kappa(A) = U \operatorname{diag}\left( \lambda_i p_\kappa(\lambda_i)^2 \right) U^\top,$$

where $A = U \operatorname{diag}(\lambda_i)U^\top$. Thus

$$\|p_\kappa(A)Ap_\kappa(A)\|_{\mathrm{op}} = \max_i \left( \lambda_i p_\kappa(\lambda_i)^2 \right) = \max_i \tau(\lambda_i) \leq \tau(1) = 1,$$

because $\tau$ is non-decreasing on $[0,1]$. Hence, the NEWTON–SCHULZ update maps the unit spectral ball into itself.

## B    MUON WITH FINITE NEWTON–SCHULZ ITERATION

---

**Algorithm 1** MUON with NEWTON–SCHULZ Orthogonalization

---

**Require:** learning rate $\eta > 0$, momentum $\beta \in [0, 1)$, NEWTON–SCHULZ steps $q \in \mathbb{N}$, NEWTON–SCHULZ polynomial $p_\kappa$ with degree $\kappa$, batch size $B$, total iteration $T$.
1: **Initialize:** $M_0 \leftarrow 0$, $W_0 \in \mathbb{R}^{m \times n}$
2: **for** $t = 1$ to $T$ **do**
3:     $G_t \leftarrow \frac{1}{B} \sum_{i=1}^{B} \nabla f(W_{t-1}; \xi_{t,i})$
4:     $M_t \leftarrow \beta M_{t-1} + G_t$
5:     $X_{t,0} \leftarrow M_t / \alpha_t$ with $\alpha_t = \max\{1, \|M_t\|_F\}$          *(scaling ensures $\|X_{t,0}\|_{\mathrm{op}} \leq 1$)*
6:     **for** $j = 1$ to $q$ **do**
7:         $X_{t,j} \leftarrow p_\kappa(X_{t,j-1} X_{t,j-1}^\top) X_{t,j-1}$
8:     **end for**
9:     $O_t \leftarrow X_{t,q}$
10:    $W_t \leftarrow W_{t-1} - \eta O_t$
11: **end for**

---

*Proof of Theorem 1.* First, we introduce the *scaled* EMA momentum: $N_t := (1 - \beta)M_t$. Then, we get $N_t = \beta N_{t-1} + (1 - \beta)G_t$. Note that the polar factor is scale-invariant (Proposition 2(v)). Let $P_t$ be the polar factor of $M_t$, i.e., $P_t = \mathrm{Polar}(M_t)$. Hence, $P_t := \mathrm{Polar}(N_t) = \mathrm{Polar}(M_t)$.

By Assumption 1, we start from *descent lemma* (Lemma 6). Since $W_t = W_{t-1} - \eta O_t$, we have

$$f(W_t) \leq f(W_{t-1}) + \langle \nabla f(W_{t-1}), W_t - W_{t-1} \rangle_F + \frac{L}{2}\|W_t - W_{t-1}\|_{\mathrm{op}}^2$$

$$= f(W_{t-1}) - \eta \langle \nabla f(W_{t-1}), O_t \rangle_F + \frac{L}{2}\eta^2 \|O_t\|_{\mathrm{op}}^2$$

$$= f(W_{t-1}) - \eta \langle N_t, O_t \rangle_F + \eta \langle N_t - \nabla f(W_{t-1}), O_t \rangle_F + \frac{L}{2}\eta^2 \|O_t\|_{\mathrm{op}}^2$$

$$\leq f(W_{t-1}) - \eta \langle N_t, O_t \rangle_F + \eta \|N_t - \nabla f(W_{t-1})\|_* \|O_t\|_{\mathrm{op}} + \frac{L}{2}\eta^2 \|O_t\|_{\mathrm{op}}^2$$

$$= f(W_{t-1}) - \eta \langle N_t, P_t \rangle_F + \eta \langle N_t, P_t - O_t \rangle_F + \eta \|N_t - \nabla f(W_{t-1})\|_* \|O_t\|_{\mathrm{op}} + \frac{L}{2}\eta^2 \|O_t\|_{\mathrm{op}}^2$$

$$= f(W_{t-1}) - \eta \|N_t\|_* + \eta \langle N_t, P_t - O_t \rangle_F + \eta \|N_t - \nabla f(W_{t-1})\|_* \|O_t\|_{\mathrm{op}} + \frac{L}{2}\eta^2 \|O_t\|_{\mathrm{op}}^2$$

$$\leq f(W_{t-1}) - \eta \|N_t\|_* + \eta \|N_t\|_* \|P_t - O_t\|_{\mathrm{op}} + \eta \|N_t - \nabla f(W_{t-1})\|_* \|O_t\|_{\mathrm{op}} + \frac{L}{2}\eta^2 \|O_t\|_{\mathrm{op}}^2$$

where the second inequality and the third inequality are due to Hölder's inequality (Lemma 4), and the last equality is due to $\langle N_t, P_t \rangle_F = \|N_t\|_*$ (Proposition 2(ii)).

Now, we define the *polar approximation error*, which is the error between the exact polar factor $P_t = \mathrm{Polar}(N_t) = \mathrm{Polar}(M_t)$ and the actual step $O_t$ generated by $q$-step NEWTON–SCHULZ steps, i.e., $O_t = $ NEWTON–SCHULZ$(M_t, q)$, measured in the operator norm.

$$\varepsilon_{t,q} := \|O_t - P_t\|_{\mathrm{op}} \quad \text{and} \quad \varepsilon_q := \sup_t \varepsilon_{t,q}$$

Since $\|P_t\|_{\mathrm{op}} \leq 1$, $\|O_t\|_{\mathrm{op}} \leq 1 + \varepsilon_{t,q} \leq 1 + \varepsilon_q$, we have

$$f(W_t) \leq f(W_{t-1}) - \eta(1 - \|P_t - O_t\|_{\mathrm{op}})\|N_t\|_* + \eta \|N_t - \nabla f(W_{t-1})\|_* \|O_t\|_{\mathrm{op}} + \frac{L}{2}\eta^2 \|O_t\|_{\mathrm{op}}^2$$

$$\leq f(W_{t-1}) - \eta(1 - \varepsilon_{t,q})\|N_t\|_* + \eta(1 + \varepsilon_{t,q})\|N_t - \nabla f(W_{t-1})\|_* + \frac{L}{2}\eta^2(1 + \varepsilon_{t,q})^2$$

$$\leq f(W_{t-1}) - \eta(1 - \varepsilon_{t,q})\|\nabla f(W_{t-1})\|_* + \eta(1 - \varepsilon_{t,q})\|\nabla f(W_{t-1}) - N_t\|_*$$
$$\quad + \eta(1 + \varepsilon_{t,q})\|N_t - \nabla f(W_{t-1})\|_* + \frac{L}{2}\eta^2(1 + \varepsilon_{t,q})^2$$

$$= f(W_{t-1}) - \eta(1 - \varepsilon_{t,q})\|\nabla f(W_{t-1})\|_* + 2\eta \|N_t - \nabla f(W_{t-1})\|_* + \frac{L}{2}\eta^2(1 + \varepsilon_{t,q})^2$$

where the last inequality is due to $-\|A\|_* \leq -\|B\|_* + \|A - B\|_*$ and the last equality holds because $(1 - \varepsilon_{t,q}) + (1 + \varepsilon_{t,q}) = 2$. Since $\varepsilon_{t,q} \leq \varepsilon_q$, we arrive at the clean one-step inequality as

$$f(W_t) \leq f(W_{t-1}) - \eta(1 - \varepsilon_q)\|\nabla f(W_{t-1})\|_* + 2\eta \|\nabla f(W_{t-1}) - N_t\|_* + \frac{L}{2}\eta^2(1 + \varepsilon_q)^2. \tag{1}$$

Rearranging and taking the expectation yields

$$\mathbb{E}\|\nabla f(W_{t-1})\|_* \le \frac{\mathbb{E}[f(W_{t-1}) - f(W_t)]}{\eta(1 - \varepsilon_q)} + \frac{2}{1 - \varepsilon_q}\mathbb{E}\|\nabla f(W_{t-1}) - N_t\|_* + \frac{L\eta(1 + \varepsilon_q)^2}{2(1 - \varepsilon_q)}. \tag{2}$$

**Bounding $\mathbb{E}[\|\nabla f(W_{t-1}) - N_t\|_*]$.**

In order to bound $\mathbb{E}[\|\nabla f(W_{t-1}) - N_t\|_*]$, we introduce the true scaled momentum $\bar{N}_t$ defined by the true (full-batch) gradient $\nabla f(W_t)$ instead of $G_t$ for each step $t$:

- $\bar{N}_t = \beta\bar{N}_{t-1} + (1 - \beta)\nabla f(W_{t-1})$ for $t > 0$ and $\bar{N}_0 = 0$.

- Note that $N_t = \beta N_{t-1} + (1 - \beta)G_t$ for $t > 0$ and $N_0 = 0$.

Then we can decompose $\mathbb{E}[\|\nabla f(W_{t-1}) - N_t\|_*]$ as

$$\mathbb{E}[\|\nabla f(W_{t-1}) - N_t\|_*] \le \underbrace{\mathbb{E}[\|\nabla f(W_{t-1}) - \bar{N}_t\|_*]}_{\text{Term (A)}} + \underbrace{\mathbb{E}[\|\bar{N}_t - N_t\|_*]}_{\text{Term (B)}} \tag{3}$$

**Bounding the Term (A).** Let $D_t = \|\nabla f(W_{t-1}) - \bar{N}_t\|_*$. From the recursion,

$$\nabla f(W_{t-1}) - \bar{N}_t = \beta(\nabla f(W_{t-1}) - \bar{N}_{t-1})$$

Hence, we have

$$\begin{aligned} D_t &= \beta\|\nabla f(W_{t-1}) - \bar{N}_{t-1}\|_* \\ &\le \beta\|\nabla f(W_{t-2}) - \bar{N}_{t-1}\| + \beta\|\nabla f(W_{t-1}) - \nabla f(W_{t-2})\|_* \\ &= \beta D_{t-1} + \beta\|\nabla f(W_{t-1}) - \nabla f(W_{t-2})\|_*. \end{aligned}$$

Applying Assumption 1 and using the fact that $\|O_{t-1}\|_{\text{op}} \le 1 + \varepsilon_{t-1,q} \le 1 + \varepsilon_q$, we have

$$\|\nabla f(W_{t-1}) - \nabla f(W_{t-2})\|_* \le L\|W_{t-1} - W_{t-2}\|_{\text{op}} = L\eta\|O_{t-1}\|_{\text{op}} \le L\eta(1 + \varepsilon_q)$$

Hence, we have the following recursion,

$$D_t \le \beta D_{t-1} + \beta L\eta(1 + \varepsilon_q)$$

Since $\bar{N}_0 = 0$, we have

$$D_1 = \|\nabla f(W_0) - \bar{N}_1\|_* = \|\nabla f(W_0) - (\beta\bar{N}_0 + (1 - \beta)\nabla f(W_0))\|_* = \beta\|\nabla f(W_0)\|_*$$

By unrolling the recursion and taking the expectation, we get

$$\mathbb{E}[D_t] \le \beta^t\mathbb{E}[\|\nabla f(W_0)\|_*] + \sum_{i=1}^{t}\beta^i L\eta(1 + \varepsilon_q) \le \beta^t\|\nabla f(W_0)\|_* + \frac{\beta L\eta(1 + \varepsilon_q)}{1 - \beta} \tag{4}$$

**Bounding the Term (B).** When we unroll the EMA recursion of both $N_t$ and $\bar{N}_t$,

$$N_t = \beta N_{t-1} + (1 - \beta)G_t = \beta^t N_0 + (1 - \beta)\sum_{i=1}^{t}\beta^{t-i}G_i$$

$$\bar{N}_t = \beta\bar{N}_{t-1} + (1 - \beta)\nabla f(W_t) = \beta^t\bar{N}_0 + (1 - \beta)\sum_{i=1}^{t}\beta^{t-i}\nabla f(W_i)$$

Then, we compute the expectation of the Frobenius norm bias caused by $M_t$ and $\bar{M}_t$,

$$\mathbb{E}[\|N_t - \bar{N}_t\|_F] \le (1 - \beta)\mathbb{E}\left[\left\|\sum_{i=1}^{t}\beta^{t-i}(G_i - \nabla f(W_i))\right\|_F\right] \tag{5}$$

Applying Jensen's inequality and the linearity of expectation gives

$$\begin{aligned} (1 - \beta)\mathbb{E}\left[\left\|\sum_{i=1}^{t}\beta^{t-i}(G_i - \nabla f(W_i))\right\|_F\right] &\le \sqrt{(1 - \beta)^2\mathbb{E}\left[\left\|\sum_{i=1}^{t}\beta^{t-i}(G_i - \nabla f(W_i))\right\|_F^2\right]} \\ &= (1 - \beta)\sqrt{\mathbb{E}\left[\sum_{i=1}^{t}\beta^{2(t-i)}\|G_i - \nabla f(W_i))\|_F^2\right]} \\ &= (1 - \beta)\sqrt{\sum_{i=1}^{t}\beta^{2(t-i)}\mathbb{E}[\|G_i - \nabla f(W_i)\|_F^2]} \end{aligned}$$

By Lemma 8, Eq.(5) becomes

$$\mathbb{E}[\|N_t - \bar{N}_t\|_F] \le (1 - \beta)\sqrt{\sum_{i=1}^{t} \beta^{2(t-i)}\mathbb{E}[\|G_i - \nabla f(W_i)\|_F^2]}$$

$$\le (1 - \beta)\sqrt{\frac{1 - \beta^{2t}}{1 - \beta^2}\frac{\sigma^2}{B}} \le \sqrt{\frac{1 - \beta}{1 + \beta}}\frac{\sigma}{\sqrt{B}}$$

Using the fact that $\|X\|_* \le \sqrt{r}\|X\|_F$ for $X \in \mathbb{R}^{m \times n}$ with $r = \min\{m, n\}$ (Proposition 2(iii)), we have

$$\mathbb{E}[\|\bar{N}_t - N_t\|_*] \le \sqrt{r}\mathbb{E}[\|\bar{N}_t - N_t\|_F] \le \sqrt{\frac{1 - \beta}{1 + \beta}}\frac{\sqrt{r}\sigma}{\sqrt{B}} \tag{6}$$

Plugging Eq.(4) and Eq.(6) into Eq.(3), we obtain

$$\mathbb{E}[\|\nabla f(W_{t-1}) - N_t\|_*] \le \frac{\beta L\eta(1 + \varepsilon_q)}{1 - \beta} + \beta^t\|\nabla f(W_0)\|_* + \sqrt{\frac{1 - \beta}{1 + \beta}}\frac{\sqrt{r}\sigma}{\sqrt{B}} \tag{7}$$

**Averaging and tuning.** Plugging Eq.(7) into Eq.(2), we get

$$\mathbb{E}[\|\nabla f(W_{t-1})\|_*] \le \frac{\mathbb{E}[f(W_{t-1}) - f(W_t)]}{\eta(1 - \varepsilon_q)}$$

$$+ \frac{2}{1 - \varepsilon_q}\left(\frac{\beta L\eta(1 + \varepsilon_q)}{1 - \beta} + \beta^t\|\nabla f(W_0)\|_* + \sqrt{\frac{1 - \beta}{1 + \beta}}\frac{\sqrt{r}\sigma}{\sqrt{B}}\right) + \frac{L\eta(1 + \varepsilon_q)^2}{2(1 - \varepsilon_q)}$$

Averaging over $t = 1, \ldots T$, we obtain

$$\frac{1}{T}\sum_{t=1}^{T}\mathbb{E}[\|\nabla f(W_{t-1})\|_*] \le \frac{D}{T\eta(1 - \varepsilon_q)} + \frac{2}{1 - \varepsilon_q}\left(\frac{\beta L\eta(1 + \varepsilon_q)}{1 - \beta} + \sqrt{\frac{1 - \beta}{1 + \beta}}\frac{\sqrt{r}\sigma}{\sqrt{B}}\right)$$

$$+ \frac{2}{1 - \varepsilon_q}\left(\frac{1}{T}\sum_{t=1}^{T}\beta^t\|\nabla f(W_0)\|_*\right) + \frac{L\eta(1 + \varepsilon_q)^2}{2(1 - \varepsilon_q)}$$

Using Proposition 2(iii) and Lemma 7, $\|\nabla f(W_0)\|_* \le \sqrt{r}\|\nabla f(W_0)\|_F \le \sqrt{2rLD}$, where $D = f(W_0) - f^*$. Then, we have

$$\frac{1}{T}\sum_{t=1}^{T}\mathbb{E}[\|\nabla f(W_{t-1})\|_*] \le \frac{D}{T\eta(1 - \varepsilon_q)} + \frac{2}{1 - \varepsilon_q}\left(\frac{\beta L\eta(1 + \varepsilon_q)}{1 - \beta} + \sqrt{\frac{1 - \beta}{1 + \beta}}\frac{\sqrt{r}\sigma}{\sqrt{B}}\right)$$

$$+ \frac{2\beta\sqrt{2rLD}}{(1 - \varepsilon_q)T(1 - \beta)} + \frac{L\eta(1 + \varepsilon_q)^2}{2(1 - \varepsilon_q)}$$

where $D = f(W_0) - f^*$. By setting $\eta = \sqrt{\frac{(1 - \beta)D}{TL}}$, we obtain

$$\frac{1}{T}\sum_{t=1}^{T}\mathbb{E}[\|\nabla f(W_{t-1})\|_*]$$

$$\le \frac{1}{1 - \varepsilon_q}\left(\frac{D}{T\eta} + \frac{L\eta}{2}(1 + \varepsilon_q)^2 + \frac{2\beta L\eta(1 + \varepsilon_q)}{1 - \beta} + \frac{2\sigma\sqrt{r}}{\sqrt{B}}\sqrt{\frac{1 - \beta}{1 + \beta}} + \frac{2\beta\sqrt{2rLD}}{T(1 - \beta)}\right)$$

$$= \frac{1}{1 - \varepsilon_q}\left(\sqrt{\frac{LD}{T}}\left(\frac{1 + 2\beta(1 + \varepsilon_q)}{\sqrt{1 - \beta}} + \frac{(1 + \varepsilon_q)^2\sqrt{1 - \beta}}{2}\right) + \frac{2\sigma\sqrt{r}}{\sqrt{B}}\sqrt{\frac{1 - \beta}{1 + \beta}} + \frac{2\beta\sqrt{2rLD}}{T(1 - \beta)}\right)$$

Setting $\beta = 1 - \min\left\{\frac{\sqrt{LDB}}{\sigma\sqrt{rT}}, 1\right\}$, we get

$$\frac{1}{T}\sum_{t=1}^{T}\mathbb{E}[\|\nabla f(W_{t-1})\|_*]$$

$$\le \frac{1}{1 - \varepsilon_q}\left(\frac{(1 + \varepsilon_q)^2\sqrt{1 - \beta}}{2}\sqrt{\frac{LD}{T}} + \frac{1 + 2\beta(1 + \varepsilon_q)}{\sqrt{1 - \beta}}\sqrt{\frac{LD}{T}} + \frac{2\sigma\sqrt{r}}{\sqrt{B}}\sqrt{\frac{1 - \beta}{1 + \beta}} + \frac{2\beta\sqrt{2rLD}}{T(1 - \beta)}\right)$$

$$= \frac{1}{1 - \varepsilon_q}\left(\frac{(1 + \varepsilon_q)^2\sqrt{1 - \beta}}{2}\sqrt{\frac{LD}{T}} + \left(1 + 2\beta(1 + \varepsilon_q) + \frac{2}{\sqrt{1 + \beta}}\right)\left(\frac{r\sigma^2 LD}{BT}\right)^{1/4} + 2\sqrt{2}\beta\frac{\sigma r}{\sqrt{BT}}\right)$$

$$= \frac{1}{1 - \varepsilon_q}\cdot\mathcal{O}\left(\sqrt{\frac{LD}{T}} + \frac{\sigma r}{\sqrt{BT}} + \left(\frac{r\sigma^2 LD}{BT}\right)^{1/4}\right) \tag{8}$$

**From NEWTON–SCHULZ residual to factor $\chi_q$.** By the NEWTON–SCHULZ residual–error link (Lemma 1) and the residual contraction (Lemma 3) as assembled in Corollary 5, we have

$$\varepsilon_q = \sup_t \varepsilon_{t,q} = \sup_t \left(1 - \sqrt{1 - \delta_{t,q}}\right) \le 1 - \sqrt{1 - \delta_0^{(\kappa+1)^q}}.$$

Hence, we bound the factor $\chi_q$ as

$$\chi_q = (1 - \varepsilon_q)^{-1} \le \left[1 - \delta_0^{(\kappa+1)^q}\right]^{-1/2}$$

with $\delta_0 = \sup_t \delta_{t,0} < 1$ (Remark 1). Finally, we conclude from Eq.(8) that

$$\frac{1}{T} \sum_{t=1}^{T} \mathbb{E}\|\nabla f(W_{t-1})\|_* \le \chi_q \cdot \mathcal{O}\left(\sqrt{\frac{LD}{T}} + \frac{\sigma r}{\sqrt{BT}} + \left(\frac{r\sigma^2 LD}{BT}\right)^{1/4}\right), \qquad \chi_q = \frac{1}{1 - \varepsilon_q},$$

with the constant factor bounded by

$$\chi_q \le \left[1 - \delta_0^{(\kappa+1)^q}\right]^{-1/2}.$$

Finally, we can find an $\epsilon$-nuclear norm stationary point of $f$ with a complexity of

$$\mathcal{O}\left(\max\left\{\frac{\chi_q^2 LD}{\epsilon^2}, \frac{\chi_q^2 r^2 \sigma^2}{B\epsilon^2}, \frac{\chi_q^4 r\sigma^2 LD}{B\epsilon^4}\right\}\right)$$

$\square$

# C  MUON WITH SVD AND SGD WITH MOMENTUM

## C.1  THEOREM 4 (MUON WITH SVD)

---

**Algorithm 2** MUON with SVD

---

**Require:** learning rate $\eta > 0$, momentum $\beta \in [0, 1)$, NEWTON–SCHULZ steps $q \in \mathbb{N}$, batch size $B$, Total iteration $T$.

1: **Initialize:** $M_0 \leftarrow 0$, $W_0 \in \mathbb{R}^{m \times n}$
2: **for** $t = 1$ to $T$ **do**
3: $\quad G_t \leftarrow \frac{1}{B} \sum_{i=1}^{B} \nabla f(W_{t-1}; \xi_{t,i})$
4: $\quad M_t \leftarrow \beta M_{t-1} + G_t$
5: $\quad O_t \leftarrow \text{Polar}(M_t)$ $\qquad\qquad$ (SVD$(M_t) = (U_t, \Sigma_t, V_t)$, then $\text{Polar}(M_t) = U_t V_t^\top$)
6: $\quad W_t \leftarrow W_{t-1} - \eta O_t$
7: **end for**
8: **return** $W_T$

---

*Proof of Theorem 4.* First, we introduce the *scaled* EMA momentum: $N_t := (1 - \beta)M_t$. Then, we get $N_t = \beta N_{t-1} + (1 - \beta)G_t$. Note that the polar factor is scale-invariant (Proposition 2(v)). Let $P_t$ be the polar factor of $M_t$, i.e., $P_t = \text{Polar}(M_t)$. Hence, $P_t := \text{Polar}(N_t) = \text{Polar}(M_t)$.

By Assumption 1, we start from *descent lemma* (Lemma 6). Since $W_t = W_{t-1} - \eta O_t$, we have

$$f(W_t) \leq f(W_{t-1}) + \langle \nabla f(W_{t-1}), W_t - W_{t-1} \rangle_F + \frac{L}{2} \|W_t - W_{t-1}\|_{\text{op}}^2$$

$$= f(W_{t-1}) - \eta \langle \nabla f(W_{t-1}), O_t \rangle_F + \frac{L}{2} \eta^2 \|O_t\|_{\text{op}}^2$$

$$= f(W_{t-1}) - \eta \langle \nabla f(W_{t-1}), P_t \rangle_F + \frac{L}{2} \eta^2 \|P_t\|_{\text{op}}^2$$

where the last equality is due to $O_t = P_t = \text{Polar}(M_t)$. Let SVD$(M_t) = (U_t, \Sigma_t, V_t)$. Then $P_t = \text{Polar}(M_t) = U_t V_t^\top$. Since $P_t = \text{Polar}(M_t)$ is a partial isometry, $\|P_t\|_{\text{op}} \leq 1$ (and $\|P_t\|_{\text{op}} = 1$ if rank$(M_t) > 0$). Then, we have

$$f(W_t) \leq f(W_{t-1}) - \eta \langle \nabla f(W_{t-1}), P_t \rangle_F + \frac{L}{2} \eta^2$$

$$= f(W_{t-1}) - \eta \langle N_t, P_t \rangle_F + \eta \langle N_t - \nabla f(W_{t-1}), P_t \rangle_F + \frac{L\eta^2}{2}$$

$$= f(W_{t-1}) - \eta \|N_t\|_* + \eta \langle N_t - \nabla f(W_{t-1}), P_t \rangle_F + \frac{L\eta^2}{2}$$

where the last equality is due to $\langle N_t, P_t \rangle_F = \|N_t\|_*$ (Proposition 2(ii)). By Hölder's inequality (Lemma 4) and the triangle inequality, we have

$$f(W_t) \leq f(W_{t-1}) - \eta \|N_t\|_* + \eta \|N_t - \nabla f(W_{t-1})\|_* \|P_t\|_{\text{op}} + \frac{L\eta^2}{2}$$

$$\leq f(W_{t-1}) - \eta \|\nabla f(W_{t-1})\|_* + \eta \|\nabla f(W_{t-1}) - N_t\|_* + \eta \|N_t - \nabla f(W_{t-1})\|_* + \frac{L\eta^2}{2}$$

$$= f(W_{t-1}) - \eta \|\nabla f(W_{t-1})\|_* + 2\eta \|\nabla f(W_{t-1}) - N_t\|_* + \frac{L\eta^2}{2}.$$

Rearranging and taking expectation gives

$$\mathbb{E}[\|\nabla f(W_{t-1})\|_*] \leq \frac{\mathbb{E}[f(W_{t-1}) - f(W_t)]}{\eta} + 2\mathbb{E}[\|\nabla f(W_{t-1}) - N_t\|_*] + \frac{L\eta}{2} \tag{9}$$

**Bounding $\mathbb{E}[\|\nabla f(W_{t-1}) - N_t\|_*]$.**

In order to bound $\mathbb{E}[\|\nabla f(W_{t-1}) - N_t\|_*]$, we introduce the true scaled momentum $\bar{N}_t$ defined by the true (full-batch) gradient $\nabla f(W_t)$ instead of $G_t$ for each step $t$:

- $\bar{N}_t = \beta\bar{N}_{t-1} + (1-\beta)\nabla f(W_{t-1})$ for $t > 0$ and $\bar{N}_0 = 0$.

- Note that $N_t = \beta N_{t-1} + (1-\beta)G_t$ for $t > 0$ and $N_0 = 0$.

Then we can decompose $\mathbb{E}[\|\nabla f(W_{t-1}) - N_t\|_*]$ as

$$\mathbb{E}[\|\nabla f(W_{t-1}) - N_t\|_*] \leq \underbrace{\mathbb{E}[\|\nabla f(W_{t-1}) - \bar{N}_t\|_*]}_{\text{Term (A)}} + \underbrace{\mathbb{E}[\|\bar{N}_t - N_t\|_*]}_{\text{Term (B)}} \quad (10)$$

**Bounding the Term (A).** Let $D_t = \|\nabla f(W_{t-1}) - \bar{N}_t\|_*$. From the recursion,

$$\nabla f(W_{t-1}) - \bar{N}_t = \beta(\nabla f(W_{t-1}) - \bar{N}_{t-1})$$

Hence, we have

$$\begin{aligned} D_t &= \beta\|\nabla f(W_{t-1}) - \bar{N}_{t-1}\|_* \\ &\leq \beta\|\nabla f(W_{t-2}) - \bar{N}_{t-1}\| + \beta\|\nabla f(W_{t-1}) - \nabla f(W_{t-2})\|_* \\ &= \beta D_{t-1} + \beta\|\nabla f(W_{t-1}) - \nabla f(W_{t-2})\|_*. \end{aligned}$$

Applying Assumption 1 and using the fact that $\|O_{t-1}\|_{\text{op}} = \|P_{t-1}\|_{\text{op}} \leq 1$, we have

$$\|\nabla f(W_{t-1}) - \nabla f(W_{t-2})\|_* \leq L\|W_{t-1} - W_{t-2}\|_{\text{op}} = L\eta\|O_{t-1}\|_{\text{op}} \leq L\eta$$

Hence, we have the following recursion,

$$D_t \leq \beta D_{t-1} + \beta L\eta$$

Since $\bar{N}_0 = 0$, we have

$$D_1 = \|\nabla f(W_0) - \bar{N}_1\|_* = \|\nabla f(W_0) - (\beta\bar{N}_0 + (1-\beta)\nabla f(W_0))\|_* = \beta\|\nabla f(W_0)\|_*$$

By unrolling the recursion and taking the expectation, we get

$$\mathbb{E}[D_t] \leq \beta^t\mathbb{E}[\|\nabla f(W_0)\|_*] + \sum_{i=1}^t \beta^i L\eta \leq \beta^t\|\nabla f(W_0)\|_* + \frac{\beta L\eta}{1-\beta} \quad (11)$$

**Bounding the Term (B).** When we unroll the EMA recursion of both $N_t$ and $\bar{N}_t$,

$$N_t = \beta N_{t-1} + (1-\beta)G_t = \beta^t N_0 + (1-\beta)\sum_{i=1}^t \beta^{t-i}G_i$$

$$\bar{N}_t = \beta\bar{N}_{t-1} + (1-\beta)\nabla f(W_t) = \beta^t\bar{N}_0 + (1-\beta)\sum_{i=1}^t \beta^{t-i}\nabla f(W_i)$$

Then, we compute the expectation of the Frobenius norm bias caused by $M_t$ and $\bar{M}_t$,

$$\mathbb{E}[\|N_t - \bar{N}_t\|_F] \leq (1-\beta)\mathbb{E}\left[\left\|\sum_{i=1}^t \beta^{t-i}(G_i - \nabla f(W_i))\right\|_F\right] \quad (12)$$

Applying Jensen's inequality and the linearity of expectation gives

$$\begin{aligned} (1-\beta)\mathbb{E}\left[\left\|\sum_{i=1}^t \beta^{t-i}(G_i - \nabla f(W_i))\right\|_F\right] &\leq \sqrt{(1-\beta)^2\mathbb{E}\left[\left\|\sum_{i=1}^t \beta^{t-i}(G_i - \nabla f(W_i))\right\|_F^2\right]} \\ &= (1-\beta)\sqrt{\mathbb{E}\left[\sum_{i=1}^t \beta^{2(t-i)}\|G_i - \nabla f(W_i))\|_F^2\right]} \\ &= (1-\beta)\sqrt{\sum_{i=1}^t \beta^{2(t-i)}\mathbb{E}[\|G_i - \nabla f(W_i)\|_F^2]} \end{aligned}$$

By Lemma 8, Eq.(12) becomes

$$\mathbb{E}[\|N_t - \bar{N}_t\|_F] \le (1 - \beta)\sqrt{\sum_{i=1}^{t} \beta^{2(t-i)}\mathbb{E}[\|G_i - \nabla f(W_i)\|_F^2]}$$

$$\le (1 - \beta)\sqrt{\frac{1 - \beta^{2t}}{1 - \beta^2}\frac{\sigma^2}{B}} \le \sqrt{\frac{1 - \beta}{1 + \beta}}\frac{\sigma}{\sqrt{B}}$$

Using the fact that $\|X\|_* \le \sqrt{r}\|X\|_F$ for $X \in \mathbb{R}^{m \times n}$ with $r = \min\{m, n\}$ (Proposition 2(iii)), we have

$$\mathbb{E}[\|\bar{N}_t - N_t\|_*] \le \sqrt{r}\mathbb{E}[\|\bar{N}_t - N_t\|_F] \le \sqrt{\frac{1 - \beta}{1 + \beta}}\frac{\sqrt{r}\sigma}{\sqrt{B}} \tag{13}$$

Plugging Eq.(11) and Eq.(13) into Eq.(10), we obtain

$$\mathbb{E}[\|\nabla f(W_{t-1}) - N_t\|_*] \le \frac{\beta L\eta}{1 - \beta} + \beta^t\|\nabla f(W_0)\|_* + \sqrt{\frac{1 - \beta}{1 + \beta}}\frac{\sqrt{r}\sigma}{\sqrt{B}} \tag{14}$$

**Averaging and tuning.** Plugging Eq.(14) into Eq.(9), we get

$$\mathbb{E}[\|\nabla f(W_{t-1})\|_*] \le \frac{\mathbb{E}[f(W_{t-1}) - f(W_t)]}{\eta} + 2\left(\frac{\beta L\eta}{1 - \beta} + \beta^t\|\nabla f(W_0)\|_* + \sqrt{\frac{1 - \beta}{1 + \beta}}\frac{\sqrt{r}\sigma}{\sqrt{B}}\right) + \frac{L\eta}{2}$$

Averaging over $t = 1, \dots T$, we obtain

$$\frac{1}{T}\sum_{t=1}^{T}\mathbb{E}[\|\nabla f(W_{t-1})\|_*] \le \frac{D}{T\eta} + 2\left(\frac{\beta L\eta}{1 - \beta} + \sqrt{\frac{1 - \beta}{1 + \beta}}\frac{\sqrt{r}\sigma}{\sqrt{B}}\right) + \frac{2}{T}\sum_{t=1}^{T}\beta^t\|\nabla f(W_0)\|_* + \frac{L\eta}{2}$$

Using Proposition 2(iii) and Lemma 7, $\|\nabla f(W_0)\|_* \le \sqrt{r}\|\nabla f(W_0)\|_F \le \sqrt{2rLD}$, where $D = f(W_0) - f^*$. Then, we have

$$\frac{1}{T}\sum_{t=1}^{T}\mathbb{E}[\|\nabla f(W_{t-1})\|_*] \le \frac{D}{T\eta} + 2\left(\frac{\beta L\eta}{1 - \beta} + \sqrt{\frac{1 - \beta}{1 + \beta}}\frac{\sqrt{r}\sigma}{\sqrt{B}}\right) + \frac{2\beta\sqrt{2rLD}}{T(1 - \beta)} + \frac{L\eta}{2}$$

where $D = f(W_0) - f^*$. By setting $\eta = \sqrt{\frac{(1 - \beta)D}{TL}}$, we obtain

$$\frac{1}{T}\sum_{t=1}^{T}\mathbb{E}[\|\nabla f(W_{t-1})\|_*] \le \frac{D}{T\eta} + \frac{L\eta}{2} + \frac{2\beta L\eta}{1 - \beta} + \frac{2\sigma\sqrt{r}}{\sqrt{B}}\sqrt{\frac{1 - \beta}{1 + \beta}} + \frac{2\beta\sqrt{2rLD}}{T(1 - \beta)}$$

$$= \sqrt{\frac{LD}{T}}\left(\frac{1 + 2\beta}{\sqrt{1 - \beta}} + \frac{\sqrt{1 - \beta}}{2}\right) + \frac{2\sigma\sqrt{r}}{\sqrt{B}}\sqrt{\frac{1 - \beta}{1 + \beta}} + \frac{2\beta\sqrt{2rLD}}{T(1 - \beta)}$$

Setting $\beta = 1 - \min\left\{\frac{\sqrt{LDB}}{\sigma\sqrt{rT}}, 1\right\}$, we get

$$\frac{1}{T}\sum_{t=1}^{T}\mathbb{E}[\|\nabla f(W_{t-1})\|_*] \le \frac{\sqrt{1 - \beta}}{2}\sqrt{\frac{LD}{T}} + \frac{1 + 2\beta}{\sqrt{1 - \beta}}\sqrt{\frac{LD}{T}} + \frac{2\sigma\sqrt{r}}{\sqrt{B}}\sqrt{\frac{1 - \beta}{1 + \beta}} + \frac{2\beta\sqrt{2rLD}}{T(1 - \beta)}$$

$$= \frac{\sqrt{1 - \beta}}{2}\sqrt{\frac{LD}{T}} + \left(1 + 2\beta + \frac{2}{\sqrt{1 + \beta}}\right)\left(\frac{r\sigma^2 LD}{BT}\right)^{1/4} + 2\sqrt{2}\beta\frac{\sigma r}{\sqrt{BT}}$$

$$= \mathcal{O}\left(\sqrt{\frac{LD}{T}} + \frac{\sigma r}{\sqrt{BT}} + \left(\frac{r\sigma^2 LD}{BT}\right)^{1/4}\right) \tag{15}$$

Thus, we can find an $\epsilon$-nuclear norm stationary point of $f$ with a complexity of

$$\mathcal{O}\left(\max\left\{\frac{LD}{\epsilon^2}, \frac{r^2\sigma^2}{B\epsilon^2}, \frac{r\sigma^2 LD}{B\epsilon^4}\right\}\right)$$

$\square$

---

**Algorithm 3** SGD with momentum (SGD-M)

---

**Require:** learning rate $\eta > 0$, momentum $\beta \in [0, 1)$, batch size $B$
1: **Initialize:** $M_0 \leftarrow 0$, $W_0 \in \mathbb{R}^{m \times n}$
2: **for** $t = 1$ to $T$ **do**
3:    $G_t \leftarrow \frac{1}{B} \sum_{i=1}^{B} \nabla f(W_t; \xi_{t,i})$
4:    $M_t \leftarrow \beta M_{t-1} + G_t$
5:    $W_t \leftarrow W_{t-1} - \eta M_t$
6: **end for**
7: **return** $W_T$

---

### C.2 THEOREM 3 (SGD WITH MOMENTUM)

*Proof of Theorem 3.* First, we introduce the *scaled* EMA momentum: $N_t := (1 - \beta)M_t$. Then, we get $N_t = \beta N_{t-1} + (1 - \beta)G_t$ with $N_0 = 0$ and the scaled learning rate $\tilde{\eta} := \frac{\eta}{1-\beta}$, yielding $W_t = W_{t-1} - \tilde{\eta}N_t$.

By Assumption 1, we start from *the descent lemma* (Lemma 6). Since $W_t = W_{t-1} - \tilde{\eta}N_t$, we have

$$f(W_t) \leq f(W_{t-1}) + \langle \nabla f(W_{t-1}), W_t - W_{t-1} \rangle_F + \frac{L}{2}\|W_t - W_{t-1}\|_{\text{op}}^2$$

$$= f(W_{t-1}) - \tilde{\eta}\langle \nabla f(W_{t-1}), N_t \rangle_F + \frac{L}{2}\tilde{\eta}^2\|N_t\|_{\text{op}}^2$$

By using $\langle a, b \rangle_F = \frac{1}{2}\left(\|a\|_F^2 + \|b\|_F^2 - \|a - b\|_F^2\right)$, we obtain

$$f(W_t) \leq f(W_{t-1}) - \frac{\tilde{\eta}}{2}\|\nabla f(W_{t-1})\|_F^2 - \frac{\tilde{\eta}}{2}\|N_t\|_F^2 + \frac{\tilde{\eta}}{2}\|N_t - \nabla f(W_{t-1})\|_F^2 + \frac{L}{2}\tilde{\eta}^2\|N_t\|_{\text{op}}^2$$

$$\leq f(W_{t-1}) - \frac{\tilde{\eta}}{2}\|\nabla f(W_{t-1})\|_F^2 - \frac{\tilde{\eta}}{2}\|N_t\|_F^2 + \frac{\tilde{\eta}}{2}\|N_t - \nabla f(W_{t-1})\|_F^2 + \frac{L}{2}\tilde{\eta}^2\|N_t\|_F^2$$

$$= f(W_{t-1}) - \frac{\tilde{\eta}}{2}\|\nabla f(W_{t-1})\|_F^2 + \frac{\tilde{\eta}}{2}\|N_t - \nabla f(W_{t-1})\|_F^2 - \frac{\tilde{\eta}(1 - L\tilde{\eta})}{2}\|N_t\|_F^2$$

$$\leq f(W_{t-1}) - \frac{\tilde{\eta}}{2}\|\nabla f(W_{t-1})\|_F^2 + \frac{\tilde{\eta}}{2}\|N_t - \nabla f(W_{t-1})\|_F^2$$

where the second inequality is due to $\|N_t\|_{\text{op}} \leq \|N_t\|_F$ (Proposition 2(iv)). The last inequality holds by choosing $\tilde{\eta} \leq 1/L$ so that we can drop the last term.

Rearranging and taking expectation gives

$$\mathbb{E}\left[\|\nabla f(W_{t-1})\|_F^2\right] \leq \frac{2\mathbb{E}[f(W_{t-1}) - f(W_t)]}{\tilde{\eta}} + \mathbb{E}[\|N_t - \nabla f(W_{t-1})\|_F^2]$$

Averaging over $t = 1$ to $T$, we obtain

$$\frac{1}{T}\sum_{t=1}^{T}\mathbb{E}\left[\|\nabla f(W_{t-1})\|_F^2\right] \leq \frac{2D}{T\tilde{\eta}} + \frac{1}{T}\sum_{t=1}^{T}\mathbb{E}[\|N_t - \nabla f(W_{t-1})\|_F^2] \tag{16}$$

where $D = f(W_0) - f^*$. Denote

$$S_A := \frac{1}{T}\sum_{t=1}^{T}\mathbb{E}[\|\nabla f(W_{t-1})\|_F^2] \leq \frac{2D}{T\tilde{\eta}} + S_B, \qquad S_B := \frac{1}{T}\sum_{t=1}^{T}\mathbb{E}[\|N_t - \nabla f(W_{t-1})\|_F^2] \tag{17}$$

**Bounding** $\frac{1}{T}\sum_{t=1}^{T}\mathbb{E}[\|N_t - \nabla f(W_{t-1})\|_F^2]$**.**

We introduce the true scaled momentum $\bar{N}_t$ defined by the true (full-batch) gradient $\nabla f(W_t)$ instead of $G_t$ for each step $t$:

- $\bar{N}_t = \beta\bar{N}_{t-1} + (1 - \beta)\nabla f(W_t)$ for $t > 0$ and $\bar{N}_0 = 0$.

- Note that $N_t = \beta N_{t-1} + (1-\beta)G_t$ for $t > 0$ and $N_0 = 0$.

Then, we can decompose $\frac{1}{T}\sum_{t=1}^{T}\mathbb{E}[\|\nabla f(W_{t-1}) - N_t\|_F^2]$ using $\|a+b\|_F^2 \le 2\|a\|_F^2 + 2\|b\|_F^2$,

$$\frac{1}{T}\sum_{t=1}^{T}\mathbb{E}[\|\nabla f(W_{t-1}) - N_t\|_F^2] \le 2\underbrace{\frac{1}{T}\sum_{t=0}^{T-1}\mathbb{E}[\|\nabla f(W_{t-1}) - \bar{N}_t\|_F^2]}_{\text{Term (A)}} + 2\underbrace{\frac{1}{T}\sum_{t=1}^{T}\mathbb{E}[\|\bar{N}_t - N_t\|_F^2]}_{\text{Term (B)}}$$
(18)

**Bounding the Term (A).** Let $e_t := \nabla f(W_{t-1}) - \bar{N}_t$. Using the recursion for $\bar{N}_t$, we have

$$\begin{aligned}
e_t &= \nabla f(W_{t-1}) - \left(\beta\bar{N}_{t-1} + (1-\beta)\nabla f(W_t)\right) \\
&= \beta\left(\nabla f(W_{t-1}) - \bar{N}_{t-1}\right) + (1-\beta)\left(\nabla f(W_{t-1}) - \nabla f(W_t)\right) \\
&= \beta e_{t-1} + \beta\left(\nabla f(W_{t-1}) - \nabla f(W_{t-2})\right) + (1-\beta)\left(\nabla f(W_{t-1}) - \nabla f(W_t)\right)
\end{aligned}$$

Apply the variation of Young's inequality (Lemma 5) with $c = \frac{1-\beta}{\beta}$:

$$\begin{aligned}
\|e_t\|_F^2 &= \|\beta e_{t-1} + \beta\left(\nabla f(W_{t-1}) - \nabla f(W_{t-2})\right) + (1-\beta)\left(\nabla f(W_{t-1}) - \nabla f(W_t)\right)\|_F^2 \\
&= (1+c)\beta^2\|e_{t-1}\|_F^2 + (1+1/c)\|\beta\left(\nabla f(W_{t-1}) - \nabla f(W_{t-2})\right) + (1-\beta)\left(\nabla f(W_{t-1}) - \nabla f(W_t)\right)\|_F^2
\end{aligned}$$

Since $1 + c = \frac{1}{\beta}$ and $1 + 1/c = 1/(1-\beta)$, and $\|x + y\|^2 \le 2\|x\|^2 + 2\|y\|^2$, we get

$$\|e_t\|_F^2 \le \beta\|e_{t-1}\|_F^2 + \frac{2\beta^2}{1-\beta}\|\nabla f(W_{t-1}) - \nabla f(W_{t-2})\|_F^2 + 2(1-\beta)\|\nabla f(W_{t-1}) - \nabla f(W_t)\|_F^2.$$

By Assumption 1 and the norm monotonicity $\|\cdot\|_{\text{op}} \le \|\cdot\|_F \le \|\cdot\|_*$ (Proposition 2(iv)), we have

$$\begin{aligned}
\|\nabla f(W_t) - \nabla f(W_{t-1})\|_F^2 &\le \|\nabla f(W_t) - \nabla f(W_{t-1})\|_*^2 \le L^2\|W_t - W_{t-1}\|_{\text{op}}^2 \\
&= L^2\tilde{\eta}^2\|N_t\|_{\text{op}}^2 \le L^2\tilde{\eta}^2\|N_t\|_F^2
\end{aligned}$$

and

$$\begin{aligned}
\|\nabla f(W_{t-1}) - \nabla f(W_{t-2})\|_F^2 &\le \|\nabla f(W_{t-1}) - \nabla f(W_{t-2})\|_*^2 \le L^2\|W_{t-1} - W_{t-2}\|_{\text{op}}^2 \\
&= L^2\tilde{\eta}^2\|N_{t-1}\|_{\text{op}}^2 \le L^2\tilde{\eta}^2\|N_{t-1}\|_F^2
\end{aligned}$$

Hence, we have the following recursion,

$$\|e_t\|_F^2 \le \beta\|e_{t-1}\|_F^2 + \frac{2\beta^2 L^2\tilde{\eta}^2}{1-\beta}\|N_{t-1}\|_F^2 + 2(1-\beta)L^2\tilde{\eta}\|N_t\|_F^2$$

Taking expectation and averaging over $t = 1$ to $T$, we obtain

$$\begin{aligned}
\frac{1}{T}\sum_{t=1}^{T}\mathbb{E}[\|\nabla f(W_{t-1}) - \bar{N}_t\|_F^2] \le{}& \beta\frac{1}{T}\sum_{t=1}^{T}\|\nabla f(W_{t-2}) - \bar{N}_{t-1}\|_F^2 \\
&+ \frac{2\beta^2 L^2\tilde{\eta}^2}{1-\beta}\frac{1}{T}\sum_{t=1}^{T}\mathbb{E}[\|N_{t-1}\|_F^2] + 2(1-\beta)L^2\tilde{\eta}^2\frac{1}{T}\sum_{t=1}^{T}\mathbb{E}[\|N_t\|_F^2]
\end{aligned}$$

Since $\frac{1}{T}\sum_{t=1}^{T}\mathbb{E}[\|\nabla f(W_{t-2}) - \bar{N}_{t-1}\|_F^2] \le \frac{1}{T}\mathbb{E}\|e_0\|_F^2 + \frac{1}{T}\sum_{t=1}^{T}\mathbb{E}[\|\nabla f(W_{t-1}) - \bar{N}_t\|_F^2]$ and $\frac{1}{T}\sum_{t=1}^{T}\mathbb{E}\|N_{t-1}\|_F^2 \le \frac{1}{T}\sum_{t=1}^{T}\mathbb{E}\|N_t\|_F^2$,

$$\frac{1}{T}\sum_{t=0}^{T-1}\mathbb{E}[\|\nabla f(W_{t-1}) - \bar{N}_t\|_F^2] \le \frac{\beta\mathbb{E}\|e_0\|_F^2}{T(1-\beta)} + \frac{2L^2\tilde{\eta}^2}{1-\beta}\left(\frac{\beta^2}{1-\beta} + 1 - \beta\right)\left(\frac{1}{T}\sum_{t=1}^{T}\mathbb{E}\|N_t\|_F^2\right).$$
(19)

**Bounding the Term (B).** When we unroll the EMA recursion of both $N_t$ and $\bar{N}_t$,

$$N_t = \beta N_{t-1} + (1-\beta)G_t = \beta^t N_0 + (1-\beta)\sum_{i=1}^{t}\beta^{t-i}G_i$$

$$\bar{N}_t = \beta \bar{N}_{t-1} + (1-\beta)\nabla f(W_t) = \beta^t \bar{N}_0 + (1-\beta)\sum_{i=1}^{t}\beta^{t-i}\nabla f(W_i)$$

Then, we compute the expectation of the Frobenius squared norm bias caused by $N_t$ and $\bar{N}_t$,

$$\mathbb{E}[\|N_t - \bar{N}_t\|_F^2] \leq \mathbb{E}\left[\left\|\beta^t(N_0 - \bar{N}_0) + (1-\beta)\sum_{i=1}^{t}\beta^{t-i}(G_i - \nabla f(W_i))\right\|_F^2\right]$$

$$= \beta^{2t}\mathbb{E}[\|N_0 - \bar{N}_0\|_F^2] + (1-\beta)^2 \mathbb{E}\left[\left\|\sum_{i=1}^{t}\beta^{t-i}(G_i - \nabla f(W_i))\right\|_F^2\right]$$

$$= \beta^{2t}\mathbb{E}[\|N_0 - \bar{N}_0\|_F^2] + (1-\beta)^2 \sum_{i=1}^{t}\beta^{2(t-i)}\mathbb{E}[\|G_i - \nabla f(W_i))\|_F^2]$$

where equalities are due to Assumption 2, which states $\mathbb{E}[G_t] = \nabla f(W_t)$ so that the cross-term vanishes. Note that $\mathbb{E}[\|N_0 - \bar{N}_0\|_F^2] = 0$. By Lemma 8, we get

$$\mathbb{E}[\|N_t - \bar{N}_t\|_F^2] \leq (1-\beta)^2 \sum_{i=1}^{t}\beta^{2(t-i)}\mathbb{E}[\|G_i - \nabla f(W_i))\|_F^2]$$

$$\leq (1-\beta)^2 \sum_{i=1}^{t}\beta^{2(t-i)}\frac{\sigma^2}{B} \leq \frac{1-\beta}{1+\beta}\frac{\sigma^2}{B}$$

By averaging over $t = 1,\ldots,T$, we have

$$\frac{1}{T}\sum_{t=1}^{T}\mathbb{E}[\|N_t - \bar{N}_t\|_F^2] \leq \frac{1}{T}\sum_{t=1}^{T}\left(\frac{1-\beta}{1+\beta}\frac{\sigma^2}{B}\right) \leq \frac{1-\beta}{1+\beta}\frac{\sigma^2}{B} \tag{20}$$

Plugging Eq.(19) and Eq.(20) to Eq.(18), we obtain

$$\frac{1}{T}\sum_{t=1}^{T}\mathbb{E}[\|\nabla f(W_{t-1}) - N_t\|_F^2]$$

$$\leq 2\left(\frac{1-\beta}{1+\beta}\frac{\sigma^2}{B}\right) + 2\left(\frac{\beta\mathbb{E}\|e_0\|_F^2}{T(1-\beta)} + \frac{2L^2\tilde{\eta}^2}{1-\beta}\left(\frac{\beta^2}{1-\beta}+1-\beta\right)\left(\frac{1}{T}\sum_{t=1}^{T}\mathbb{E}\|N_t\|_F^2\right)\cdot\right)$$

$$\leq 2\left(\frac{1-\beta}{1+\beta}\frac{\sigma^2}{B}\right) + \frac{2\beta\mathbb{E}\|e_0\|_F^2}{T(1-\beta)}$$

$$+ \frac{8L^2\tilde{\eta}^2}{1-\beta}\left(\frac{\beta^2}{1-\beta}+1-\beta\right)\left(\frac{1}{T}\sum_{t=1}^{T}\mathbb{E}[\|\nabla f(W_{t-1})\|_F^2] + \frac{1}{T}\sum_{t=1}^{T}\mathbb{E}[\|N_t - \nabla f(W_{t-1})\|_F^2]\right) \tag{21}$$

where the last inequality is due to $\|a+b\|_F^2 \leq 2\|a\|_F^2 + 2\|b\|_F^2$. Eq.(21) can be expressed as

$$S_B \leq 2\left(\frac{1-\beta}{1+\beta}\frac{\sigma^2}{B}\right) + \frac{2\beta\mathbb{E}\|e_0\|_F^2}{T(1-\beta)} + \frac{8L^2\tilde{\eta}^2}{1-\beta}\left(\frac{\beta^2}{1-\beta}+1-\beta\right)(S_A + S_B)$$

By Lemma 7 and Proposition 2(iv), we have $\mathbb{E}\|e_0\|_F^2 = \mathbb{E}\|\nabla f(W_0)\|_F^2 \leq \mathbb{E}\|\nabla f(W_0)\|_*^2 \leq 2LD$. Therefore,

$$S_B \leq 2\left(\frac{1-\beta}{1+\beta}\frac{\sigma^2}{B}\right) + \frac{4\beta LD}{T(1-\beta)} + \frac{8L^2\tilde{\eta}^2}{1-\beta}\left(\frac{\beta^2}{1-\beta}+1-\beta\right)(S_A + S_B)$$

Let $\theta := \frac{4L^2\tilde{\eta}^2}{1-\beta}\left(\frac{\beta^2}{1-\beta}+1-\beta\right) = \frac{4L^2\tilde{\eta}^2}{(1-\beta)^2}K$ and $K := \beta^2 + (1-\beta)^2$. Then, we have

$$S_B \leq 2\theta(S_A + S_B) + \frac{4\beta LD}{(1-\beta)T} + 2\left(\frac{1-\beta}{1+\beta}\frac{\sigma^2}{B}\right)$$

Hence

$$(1-2\theta)S_B \leq 2\theta S_A + \frac{4\beta LD}{(1-\beta)T} + 2\left(\frac{1-\beta}{1+\beta}\frac{\sigma^2}{B}\right). \tag{22}$$

Insert Eq.(22) into Eq.(17), we obtain

$$S_A \leq \frac{2D}{T\tilde{\eta}} + S_B \leq \frac{2D}{T\tilde{\eta}} + \frac{2\theta}{1-2\theta}S_A + \frac{1}{1-2\theta}\left(\frac{4\beta LD}{(1-\beta)T} + 2\left(\frac{1-\beta}{1+\beta}\frac{\sigma^2}{B}\right)\right)$$

Therefore, provided $\theta < \frac{1}{4}$,

$$S_A \leq \frac{2(1-2\theta)}{1-4\theta} \cdot \frac{D}{\tilde{\eta}T} + \frac{1}{1-4\theta}\left(\frac{4\beta LD}{(1-\beta)T} + 2\left(\frac{1-\beta}{1+\beta}\frac{\sigma^2}{B}\right)\right). \tag{23}$$

Finally, we have

$$\frac{1}{T}\sum_{t=1}^{T}\mathbb{E}[\|\nabla f(W_{t-1})\|_F^2] \leq \frac{2(1-2\theta)}{1-4\theta} \cdot \frac{(1-\beta)D}{\eta T} + \frac{1}{1-4\theta}\left(\frac{4\beta LD}{(1-\beta)T} + 2\left(\frac{1-\beta}{1+\beta}\frac{\sigma^2}{B}\right)\right). \tag{24}$$

**Tuning $\eta$ and $\beta$.** We need $\tilde{\eta} \leq 1/L$ and $\theta < \frac{1}{4}$. Since $K = \beta^2 + (1-\beta)^2 \in [1/2, 1]$, a convenient sufficient choice is

$$\eta \leq \min\left\{\frac{1-\beta}{L}, \frac{(1-\beta)^2}{4L\sqrt{K}}\right\}$$

Applying Jensen's inequality and using the fact that $\|X\|_* \leq \sqrt{r}\|X\|_F$ for $X \in \mathbb{R}^{m\times n}$ with $r = \min\{m, n\}$ (Proposition 2(iii)), we get

$$\frac{1}{T}\sum_{t=0}^{T-1}\mathbb{E}[\|\nabla f(W_t)\|_*] \leq \sqrt{r}\left(\frac{1}{T}\sum_{t=0}^{T-1}\mathbb{E}[\|\nabla f(W_t)\|_F]\right) \leq \sqrt{r}\sqrt{\frac{1}{T}\sum_{t=0}^{T-1}\mathbb{E}[\|\nabla f(W_t)\|_F^2]}$$

$$\leq \sqrt{r}\sqrt{\frac{2(1-2\theta)}{1-4\theta} \cdot \frac{(1-\beta)D}{\eta T} + \frac{1}{1-4\theta}\left(\frac{4\beta LD}{(1-\beta)T} + 2\left(\frac{1-\beta}{1+\beta}\frac{\sigma^2}{B}\right)\right)}$$

$$\leq \sqrt{\frac{2r(1-2\theta)}{1-4\theta} \cdot \frac{(1-\beta)D}{\eta T}} + \sqrt{\frac{r}{1-4\theta} \cdot \frac{4\beta LD}{(1-\beta)T}} + \sqrt{\frac{r}{1-4\theta} \cdot \frac{2(1-\beta)}{1+\beta} \cdot \frac{\sigma^2}{B}}$$

where the last inequality is due to $\sqrt{a+b} \leq \sqrt{a} + \sqrt{b}$. By setting $\eta = \min\left\{\frac{1-\beta}{L}, \frac{(1-\beta)^2}{4L\sqrt{K}}\right\}$, which guarantees $\tilde{\eta} \leq 1/L$ and $\theta < \frac{1}{4}$, we obtain,

$$\frac{1}{T}\sum_{t=0}^{T-1}\mathbb{E}[\|\nabla f(W_t)\|_*] \leq \underbrace{\mathcal{O}\left(\sqrt{\frac{rLD}{T}}\right)}_{\text{Deterministic Term}} + \underbrace{\mathcal{O}\left(\sqrt{\frac{r\beta LD}{(1-\beta)T}}\right) + \mathcal{O}\left(\sqrt{\frac{r\sigma^2(1-\beta)}{(1+\beta)B}}\right)}_{\text{Stochastic Term}}$$

By setting $\beta = 1 - \min\left\{\frac{\sqrt{LDB}}{\sigma\sqrt{T}}, 1\right\}$, we get

$$\frac{1}{T}\sum_{t=0}^{T-1}\mathbb{E}[\|\nabla f(W_t)\|_*] \leq \mathcal{O}\left(\sqrt{\frac{rLD}{T}}\right) + \mathcal{O}\left(\sqrt{\frac{r\beta LD}{(1-\beta)T}}\right) + \mathcal{O}\left(\sqrt{\frac{r\sigma^2(1-\beta)}{(1+\beta)B}}\right)$$

$$\leq \mathcal{O}\left(\sqrt{\frac{rLD}{T}} + \left(\frac{r^2\sigma^2 LD}{BT}\right)^{1/4}\right)$$

Thus, we can find an $\epsilon$-nuclear norm stationary point of $f$ with a complexity of

$$\mathcal{O}\left(\max\left\{\frac{rLD}{\epsilon^2}, \frac{r^2\sigma^2 LD}{B\epsilon^4}\right\}\right)$$

$\square$

## D NEWTON–SCHULZ LEMMAS: PROOFS

**Remark 1 (Initial residual below one).**

With $X_{t,0} = M_t/\alpha_t$ and $\alpha_t = \max\{1, \|M_t\|_F\}$, we have $\delta_{t,0} \in [0,1)$ for every $t$; moreover $\delta_{t,0} = 0$ when $M_t = 0$.

*Proof.* Let $M_t = U\Sigma V^\top$ (rank $r_t$). Then $X_{t,0}X_{t,0}^\top = U(\Sigma^2/\alpha_t^2)U^\top$, so on $\mathrm{range}(M_t)$ the eigenvalues are $\sigma_i^2/\alpha_t^2 \in (0,1]$. If $r_t = 0$, set $\delta_{t,0} = 0$. Otherwise, the minimal positive eigenvalue $\lambda_{\min}^+ = \min_{i \le r_t} \sigma_i^2/\alpha_t^2 > 0$, hence by Lemma 1, $\delta_{t,0} = 1 - \lambda_{\min}^+ < 1$. □

**Lemma 9.** *(Polar factor invariance under* NEWTON–SCHULZ*).*

*As* NEWTON–SCHULZ *iterates by the polynomial* $p_\kappa$*, i.e.,* $X_{t,0} = M_t/\alpha_t$*,* $X_{t,j+1} = p_\kappa(X_{t,j}X_{t,j}^\top)X_{t,j}$*, the polar factor is invariant:* $\mathrm{Polar}(M_t) = \mathrm{Polar}(X_{t,j})$ *for all* $j \ge 0$*.*

*Proof.* First, the polar factor is invariant under scalar multiplication. ($\because$ Let $\mathrm{SVD}(X) = (U, \Sigma, V)$. Then $\mathrm{Polar}(X) = UV^\top$. Now, let $Y = cX$ for $c > 0$. Then $\mathrm{SVD}(Y) = \mathrm{SVD}(cX) = (U, c\Sigma, V)$. Thus, $\mathrm{Polar}(Y) = UV^\top$. Hence, $P_t = \mathrm{Polar}(M_t) = \mathrm{Polar}(M_t/\alpha_t) = \mathrm{Polar}(X_{t,0})$ holds.

Now, to prove by induction on $j \ge 0$, we assume that $P_t = \mathrm{Polar}(X_{t,j})$. Note that one step of NEWTON–SCHULZ for polynomial $p$ is defined as

$$X_{t,j+1} = p(X_{t,j}X_{t,j}^\top)X_{t,j}$$

Let $\mathrm{SVD}(X_{t,j}) = (U, \Sigma, V)$ with $\Sigma \succ 0$ (on $\mathrm{range}(X_{t,j})$), so that $\mathrm{Polar}(X_{t,j}) = UV^\top$. Then

$$p(X_{t,j}X_{t,j}^\top)X_{t,j} = p(U\Sigma V^\top(U\Sigma V^\top)^\top)U\Sigma V^\top = p(U\Sigma^2 U^\top)U\Sigma V^\top = Up(\Sigma^2)\Sigma V^\top$$

Thus, $p(X_{t,j}X_{t,j}^\top)X_{t,j}$ has the same left/right singular vectors $U, V$ as $X_{t,j}$. Hence,

$$\mathrm{Polar}(X_{t,j+1}) = \mathrm{Polar}(p(X_{t,j}X_{t,j}^\top)X_{t,j}) = \mathrm{Polar}(X_{t,j}) = UV^\top$$

By induction, we have

$$P_t = \mathrm{Polar}(M_t) = \mathrm{Polar}(X_{t,0}) = \mathrm{Polar}(X_{t,1}) = \ldots = \mathrm{Polar}(X_{t,q})$$

which implies the polar factor invariance under NEWTON–SCHULZ updates. □

**Lemma 10.** *(Support invariance under* NEWTON–SCHULZ*).*

*As* NEWTON–SCHULZ *iterates by the polynomial* $p_\kappa$*, i.e.,* $X_{t,0} = M_t/\alpha_t$*,* $X_{t,j+1} = p_\kappa(X_{t,j}X_{t,j}^\top)X_{t,j}$*, the support is invariant:* $\mathrm{range}(M_t) = \mathrm{range}(X_{t,j})$ *for all* $j \ge 0$*, and* $p_\kappa(X_{t,j}X_{t,j}^\top)$ *is positive definite on* $\mathrm{range}(X_{t,j})$*.*

*Proof.* On $\mathrm{range}(X_{t,j})$ the spectrum of $A := X_{t,j}X_{t,j}^\top$ lies in $(0,1]$. $p_\kappa$ is strictly positive on $(0,1]$ (Proposition 1), so $p_\kappa(A)|_{\mathrm{range}(X_{t,j})}$ is invertible. Hence

$$\mathrm{range}(X_{t,j+1}) = \mathrm{range}(p_\kappa(A)X_{t,j}) = \mathrm{range}(X_{t,j}),$$

and by induction $\mathrm{range}(X_{t,j}) = \mathrm{range}(M_t)$ for all $j$. □

*Remark 2* (Support-aware details). As in the main text, all NEWTON–SCHULZ quantities are restricted to $\mathrm{range}(M_t)$ (no standing full-rank assumption). Lemma 10 preserves the column space; $P_t$, $X_{t,j}X_{t,j}^\top$, and $\Pi_t$ vanish on $\mathrm{range}(M_t)^\perp$.

**Lemma 1 (Orthogonality residual vs. Polar approximation error).**

Let $\lambda_{\min}^+$ be the smallest positive eigenvalue of $X_{t,q}X_{t,q}^\top$ restricted to $\mathrm{range}(M_t)$ (set $\lambda_{\min}^+ = 1$ if $\mathrm{rank}(M_t) = 0$). Then

$$\delta_{t,q} = 1 - \lambda_{\min}^+, \qquad \varepsilon_{t,q} = 1 - \sqrt{\lambda_{\min}^+} = 1 - \sqrt{1 - \delta_{t,q}}.$$

*Proof.* Let $X_{t,q} = U\Sigma V^\top$ and $\Pi_t = UU^\top$ with $\Sigma = \mathrm{diag}(\sigma_i)$. Then,

$$\Pi_t - X_{t,q}X_{t,q}^\top = U(I - \Sigma^2)U^\top$$

on $\mathrm{range}(M_t)$ and vanishes on $\mathrm{range}(M_t)^\perp$. Thus, the orthogonality residual $\delta_{t,q}$ is computed as

$$\delta_{t,q} = \|\Pi_t - X_{t,q}X_{t,q}^\top\|_{\mathrm{op}} = \max_i |1 - \sigma_i^2| = 1 - \lambda_{\min}^+ \tag{25}$$

where $\lambda_{\min}^+$ is the smallest positive eigenvalue of $X_{t,q}X_{t,q}^\top$ on $\mathrm{range}(M_t)$. By the polar factor invariance under NEWTON–SCHULZ updates (Lemma 9), we have

$$\varepsilon_{t,q} = \|X_{t,q} - \mathrm{Polar}(M_t)\|_{\mathrm{op}} = \|U\Sigma V^\top - UV^\top\|_{\mathrm{op}} = \|\Sigma - I\|_{\mathrm{op}} = 1 - \min_i \sigma_i = 1 - \sqrt{\lambda_{\min}^+} \tag{26}$$

Combining Eq.(25) and Eq.(26), we conclude

$$\varepsilon_{t,q} = 1 - \sqrt{\lambda_{\min}^+} = 1 - \sqrt{1 - \delta_{t,j}}$$

$\square$

**Lemma 2 (Residual update).**

For NEWTON–SCHULZ polynomial $p_\kappa$ the orthogonality residual $\delta_{t,j}$ is updated by NEWTON–SCHULZ per step as

$$\delta_{t,j+1} = \phi(\delta_{t,j}),$$

where $\phi(u) := 1 - (1 - u)\left[p_\kappa(1 - u)\right]^2$.

*Proof.* Recall that NEWTON–SCHULZ update is defined as

$$X_{t,j+1} = p(X_{t,j}X_{t,j}^\top)X_{t,j}, \quad j = 0, 1, \ldots, q - 1$$

where $\|X_{t,0}\|_{\mathrm{op}} \le 1$. Let $A_{t,j}$ be $X_{t,j}X_{t,j}^\top$. Then $A_{t,j}$ is symmetric, because

$$A_{t,j}^\top = (X_{t,j}X_{t,j}^\top)^\top = (X_{t,j}^\top)^\top X_{t,j}^\top = X_{t,j}X_{t,j}^\top = A_{t,j}$$

Applying NEWTON–SCHULZ update, we get

$$A_{t,j+1} = X_{t,j+1}X_{t,j+1}^\top = (p(A_{t,j})X_{t,j})(p(A_{t,j})X_{t,j})^\top$$
$$= p(A_{t,j})X_{t,j}X_{t,j}^\top p(A_{t,j})^\top = p(A_{t,j})A_{t,j}p(A_{t,j})^\top$$

Since $A_{t,j}$ is symmetric, any polynomial $p(A_{t,j})$ is also symmetric. Thus, $p(A_{t,j})^\top = p(A_{t,j})$. Therefore, we have

$$A_{t,j+1} = p(A_{t,j})A_{t,j}p(A_{t,j}) \tag{27}$$

Since $A_{t,j}$ is symmetric, $A_{t,j}$ is orthogonally diagonalizable. Let $\{\lambda_0, \lambda_1, \ldots\}$ be the eigenvalues of $A_{t,j}$. Hence, we can write its spectral decomposition as $A_{t,j} = U\Lambda U^\top = U\,\mathrm{diag}(\lambda_l)U^\top$. By the key property of matrix polynomials, $p(A_{t,j}) = Up(\Lambda)U^\top = U\,\mathrm{diag}(p(\lambda_l))U^\top$. Now, we substitute the spectral decompositions of $A_{t,j}$ and $p(A_{t,j})$ into Eq.(27),

$$A_{t,j+1} = (Up(\Lambda)U^\top)(U\Lambda U^\top)(Up(\Lambda)U^\top) = U(p(\Lambda)\Lambda p(\Lambda))U^\top$$

Let the new diagonal matrix be $\Lambda' := p(\Lambda)\Lambda p(\Lambda)$. Then, the diagonal entries of $\Lambda'$ are $\lambda_\ell[p(\lambda_\ell)]^2$. The expression for $A_{t,j+1} = U\Lambda'U^\top$, which is the spectral decomposition of $A_{t,j+1}$. This form explicitly shows that the eigenvalues of $A_{t,j+1}$ are the diagonal entries of $\Lambda'$, which are $\lambda_\ell[p(\lambda_\ell)]^2$.

We conclude that $A_{t,j+1} = p(A_{t,j})A_{t,j}p(A_{t,j}) = U\,\mathrm{diag}(\lambda_i[p(\lambda_\ell)]^2)\,U^\top$ on $\mathrm{range}(M_t)$ and both $\Pi_t$ and $A_{t,j}$ vanish on $\mathrm{range}(M_t)^\perp$.

**Claim:** $\|X_{t,j}\|_{\mathrm{op}} \le 1$ **for all $j \ge 0$.**

For $j = 0$, it is trivial. Assume that $\|X_{t,j}\|_{op} \le 1$ holds. Then, $\|A_{t,j}\|_{op} = \|X_{t,j}X_{t,j}^\top\|_{op} = \|X_{t,j}\|_{op}^2 \le 1$. Hence, the largest singular value of $A_{t,j}$ is in $[0,1]$, which implies all eigenvalues of symmetric $A_{t,j}$ are in $[0,1]$, i.e., $\max_l \lambda_l = \lambda_{\max} \le 1$.

$$\|X_{t,j+1}\|_{op}^2 = \|X_{t,j+1}X_{t,j+1}^\top\|_{op} = \|A_{t,j+1}\|_{op} = \max_\ell \left(\lambda_\ell[p(\lambda_\ell)]^2\right)$$

Since $\tau(\lambda) = \lambda[p(\lambda)]^2$ is non-decreasing on $[0,1]$ with $\tau(1) \le 1$ (Proposition 1), we have

$$\|X_{t,j+1}\|_{op}^2 = \max_\ell \left(\tau(\lambda_l)\right) = \tau(\max_\ell \lambda_l) = \tau(\lambda_{\max}) \le \tau(1) \le 1$$

By induction, we get $\|X_{t,j}\|_{op} \le 1$ for all $j \ge 0$. The claim also implies $\|A_{t,j}\|_{op} \le 1$ for all $j \ge 0$.

Now, recall that the orthogonality residual at step $j$ is defined as

$$\delta_{t,j} = \left\|\Pi_t - X_{t,j}X_{t,j}^\top\right\|_{op} = \left\|\Pi_t - A_{t,j}\right\|_{op}$$

Let $\lambda_{\min}^+$ be the minimum eigenvalue of $A_{t,j}$, i.e., $\min_l \lambda_l = \lambda_{\min}^+ \le 1$. Then, by the claim, we have

$$\delta_{t,j} = \|\Pi_t - A_{t,j}\|_{op} = \max_l |1 - \lambda_l| = 1 - \lambda_{\min}^+ \tag{28}$$

Now, we consider the next step residual $\delta_{t,j+1}$,

$$\delta_{t,j+1} = \|\Pi_t - A_{t,j+1}\|_{op} = \max_l \left|1 - \lambda_\ell[p(\lambda_\ell)]^2\right| = \max_l \left(1 - \lambda_\ell[p(\lambda_\ell)]^2\right)$$

where the last equality is due to $\|A_{t,j+1}\|_{op} \le 1$. Since $\tau(\lambda) = \lambda[p(\lambda)]^2$ is non-decreasing on $[0,1]$ with $\tau(1) \le 1$, we have

$$\begin{aligned}\delta_{t,j+1} &= 1 - \lambda_{\min}^+[p(\lambda_{\min}^+)]^2 \\ &= 1 - (1 - \delta_{t,j})[p(1 - \delta_{t,j})]^2\end{aligned}$$

where the last equality is due to Eq.(28). Therefore, we conclude that the orthogonality residual is updated by NEWTON−SCHULZ as

$$\delta_{t,j+1} = \phi(\delta_{t,j})$$

where $\phi(u) = 1 - (1 - u)[p(1 - u)]^2$ ▢

**Lemma 3 (Residual Decay by NEWTON−SCHULZ Polynomial).** For NEWTON−SCHULZ polynomial $p_\kappa$, $\phi(u) \le u^{\kappa+1}$ on $[0,1]$ where $\phi$ is a function defined in Lemma 2. Hence, for every $t$ and all $j \ge 0$,

$$\delta_{t,j+1} \le \delta_{t,j}^{\kappa+1}, \qquad \delta_{t,q} \le \delta_{t,0}^{(\kappa+1)^q}.$$

*Proof.* The NEWTON−SCHULZ polynomial $p(\lambda)$ of degree $\kappa$ is defined as the truncation of the Taylor series of $g(\lambda) = \lambda^{-1/2}$ expanded around $\lambda = 1$. This means that the first $\kappa$ derivatives of $p(\lambda)$ and $g(\lambda)$ are identical at $\lambda = 1$:

$$p^{(s)}(1) = g^{(s)}(1) \qquad \text{for } s = 0, 1, \ldots, \kappa$$

Let's consider the variable substitution $\lambda = 1 - u$. The function becomes $f(u) = g(1 - u) = (1 - u)^{-1/2}$. The polynomial $p(1 - u)$ is then the Taylor expansion of $f(u)$ around $u = 0$ up to the term $u^\kappa$:

$$p(1 - u) = \sum_{s=0}^{\kappa} \frac{f^{(s)}(0)}{s!} u^s = \sum_{s=0}^{\kappa} c_s u^s \tag{29}$$

The derivatives of $f(u) = (1 - u)^{-1/2}$ at $u = 0$ are $f^{(s)}(0) = \frac{(2s)!}{s!4^s}$, so that the coefficients are $c_s = \frac{(2s)!}{(s!)^2 4^s}$. Moreover, since $p(1 - u)$ and $f(u)$ share the same first $\kappa$ derivatives at $u = 0$, we get

$$f^{(s)}(0) = (-1)^s p^{(s)}(1), \qquad \text{for } s = 1, \ldots, \kappa \tag{30}$$

Consider the function $\tau(\lambda) := \lambda[p(\lambda)]^2$.

**Claim:** $\tau(\lambda)$ **is non-decreasing on** $[0,1]$ **with** $\tau(1) \leq 1$.

$\tau(1) = 1 \cdot [p(1)]^2 = [g(1)]^2 = 1$ holds. Now, the derivative of $\tau(\lambda)$ is

$$\tau'(\lambda) = [p(\lambda)]^2 + 2\lambda p(\lambda)p'(\lambda) = p(\lambda)\left(p(\lambda) + 2\lambda p'(\lambda)\right)$$

With $\lambda = 1 - u$, we have $p'(\lambda) = -p'(1-u)$. Then, we obtain

$$\tau'(\lambda) = p(1-u) \underbrace{\left(p(1-u) - 2(1-u)p'(1-u)\right)}_{:=S(u)} \tag{31}$$

Since all coefficients of $p(1-u)$ are positive, i.e., $c_s = \frac{(2s)!}{(s!)^2 4^s} > 0$, we have

$$p(1-u) > 0 \text{ for } u \in [0,1]$$

Now, it suffices to prove $S(u) \geq 0$ for $u \in [0,1]$. From Eq.(29), we can compute $S(u)$:

$$
\begin{aligned}
S(u) &= p(1-u) - 2(1-u)p'(1-u) \\
&= \sum_{s=0}^{\kappa} c_s u^s - 2(1-u) \sum_{s=1}^{\kappa} s c_s u^{s-1} \\
&= (c_0 - 2c_1) + \sum_{s=1}^{\kappa-1} \left((2s+1)c_s - 2(s+1)c_{s+1}\right) u^s + (2\kappa+1)c_\kappa u^\kappa
\end{aligned}
\tag{32}
$$

Note that the ratio for the coefficients is,

$$\frac{c_{s+1}}{c_s} = \frac{\frac{(2s+2)!}{((s+1)!)^2 4^{s+1}}}{\frac{(2s)!}{(s!)^2 4^s}} = \frac{2s+1}{2(s+1)}$$

Therefore,

$$(2s+1)c_s - 2(s+1)c_{s+1} = (2s+1)c_s - 2(s+1)\left(\frac{2s+1}{2(s+1)}c_s\right) = 0,$$

and also $c_0 - 2c_1 = 1 - 2 \cdot \frac{1}{2} = 0$. Hence, all coefficients up to degree $\kappa - 1$ cancel in Eq.(32), leaving the exact factorization

$$S(u) = (2\kappa+1)c_\kappa u^\kappa, \qquad c_\kappa = \frac{(2\kappa)!}{4^\kappa(\kappa!)^2} > 0. \tag{33}$$

Putting Eq.(33) into Eq.(31), which is $\tau'(\lambda) = p(1-u)S(u)$, gives

$$\tau'(\lambda) = p(1-u)(2\kappa+1)c_\kappa u^\kappa, \qquad \text{with } u = 1 - \lambda \in [0,1].$$

Since $p(1-u) > 0$, $c_\kappa > 0$, and $u^\kappa \geq 0$ on $[0,1]$, we have $\tau'(\lambda) \geq 0$ for $\lambda \in [0,1]$. Moreover, $\tau'(\lambda) > 0$ for $\lambda \in [0,1)$ (because then $u > 0$), and $\tau'(1) = 0$. Therefore, $\tau$ is non-decreasing on $[0,1]$ (in fact, it is strictly increasing on $[0,1)$), as claimed.

Now, we can apply Lemma 2, since $\tau(\lambda)$ is non-decreasing on $[0,1]$ with $\tau(1) \leq 1$. From Lemma 2, we know the orthogonality residual is updated according to the rule:

$$\delta_{t,j+1} = \phi(\delta_{t,j}) \tag{34}$$

where the function $\phi(u)$ is defined as:

$$\phi(u) = 1 - (1-u)[p(1-u)]^2 \tag{35}$$

**Claim:** $\phi^{(s)}(0) = 0$ **for all** $s = 1, \ldots, \kappa$. The function $\phi(u)$ can be rewritten using $f(u) = (1-u)^{-1/2}$ and $p(1-u)$:

$$\phi(u) = 1 - \frac{1}{f(u)^2}[p(1-u)]^2 \tag{36}$$

Let's define a remainder function $R_\kappa(u)$, which represents the difference between $f(u)$ and its Taylor approximation $p(1 - u)$:

$$R_\kappa(u) = f(u) - p(1 - u) \tag{37}$$

Since $p(1 - u)$ matches the derivatives of $f(u)$ up to order $\kappa$ at $u = 0$, which implies Eq.(30), the remainder function and its first $\kappa$ derivatives are all zero at $u = 0$:

$$R_\kappa^{(s)}(0) = f^{(s)}(0) - p^{(s)}(1) \cdot (-1)^s = 0, \qquad \text{for } s = 1, \ldots, \kappa \tag{38}$$

Substituting $p(1 - u) = f(u) - R_\kappa(u)$ from Eq.(37) to Eq.(36),

$$\begin{aligned}
\phi(u) &= 1 - \frac{[f(u) - R_\kappa(u)]^2}{f(u)^2} \\
&= 1 - \frac{f(u)^2 - 2f(u)R_\kappa(u) + R_\kappa(u)^2}{f(u)^2} \\
&= 1 - \left(1 - \frac{2R(u)}{f(u)} + \frac{R_\kappa(u)^2}{f(u)^2}\right) \\
&= \frac{2R_\kappa(u)}{f(u)} - \frac{R_\kappa(u)^2}{f(u)^2} = R_\kappa(u) \cdot \left(\frac{2f(u) - R_\kappa(u)}{f(u)^2}\right)
\end{aligned}$$

Let's define the term in the brackets as a new function, $H(u)$:

$$H(u) = \frac{2f(u) - R_\kappa(u)}{f(u)^2}$$

So, we have a simple product form: $\phi(u) = R_\kappa(u)H(u)$. Note that $H(u)$ is well-behaved around $u = 0$ since $f(0) = 1$ and $R_\kappa(0) = 0$, making the denominator non-zero. To find the derivatives of $\phi(u)$, we use the Leibniz rule (the generalized product rule) for the $s$-th derivative of a product:

$$\phi^{(s)}(u) = \frac{d^s}{du^s}[R_\kappa(u)H(u)] = \sum_{j=0}^{s} \binom{s}{j} R_\kappa^{(j)}(u) H^{(s-j)}(u)$$

Now, we evaluate this $s$-th derivative at $u = 0$:

$$\phi^{(s)}(0) = \sum_{j=0}^{s} \binom{s}{j} R_\kappa^{(j)}(0) H^{(s-j)}(0)$$

Let's consider any integer $s$ in the range $0 \leq s \leq \kappa$. In the summation above, the index $j$ runs from 0 to $s$. For every term in this sum, the condition $j \leq s \leq \kappa$ holds. From Eq.(38), we know that $R_\kappa^{(j)}(0) = 0$ for all $j \leq \kappa$. Therefore, every single term in the summation contains a factor of $R_\kappa^{(j)}(0)$ which is equal to zero.

$$\phi^{(s)}(0) = \sum_{j=0}^{s} \binom{s}{j} (0) \cdot H^{(s-j)}(0) = 0$$

This result holds for all $s = 1, \ldots, \kappa$. Thus, we have proven that the first $\kappa$ derivatives of $\phi(u)$ at $u = 0$ are all zero:

$$\phi(0) = \phi'(u) = \cdots = \phi^{(\kappa)}(0) = 0$$

This implies that the Taylor series for $\phi(u)$ starts with a term of order $u^{\kappa+1}$:

$$\phi(u) = \frac{\phi^{(\kappa+1)}(0)}{(\kappa + 1)!} u^{\kappa+1} + \mathcal{O}(u^{\kappa+2})$$

**Find the leading term of $\phi(u)$.**

Recall that we can express $p(1 - u)$ using the Taylor remainder theorem:

$$p(1 - u) = f(u) - R_\kappa(u) \tag{39}$$

where $R_\kappa(u)$ is the remainder term, which is of order $\mathcal{O}(u^{\kappa+1})$. Substituting Eq.(39) into the expression for $\phi(u)$ in Eq.(35):

$$\begin{aligned}
\phi(u) &= 1 - (1-u)[f(u) - R_\kappa(u)]^2 \\
&= 1 - (1-u)[f(u)^2 - 2f(u)R_\kappa(u) + R_\kappa(u)^2] \\
&= 1 - (1-u)\left[\frac{1}{1-u} - \frac{2R_\kappa(u)}{\sqrt{1-u}} + R_\kappa(u)^2\right] \\
&= 1 - \left[1 - 2\sqrt{1-u}R_\kappa(u) + (1-u)R_\kappa(u)^2\right] \\
&= 2\sqrt{1-u}R_\kappa(u) - (1-u)R_\kappa(u)^2
\end{aligned}$$

The remainder $R_\kappa(u)$ is given by $R_\kappa(u) = \frac{f^{(\kappa+1)}(c)}{(\kappa+1)!}u^{\kappa+1}$ for some $c \in (0, u)$. As $u \to 0$, the leading term of $\phi(u)$ is determined by $2\sqrt{1-u} \cdot \frac{f^{(\kappa+1)}(0)}{(\kappa+1)!}u^{\kappa+1}$. The derivatives of $f(u) = (1-u)^{-1/2}$ at $u = 0$ are $f^{(s)}(0) = \frac{(2s)!}{s!4^s}$. Thus, the leading coefficient of the Taylor series for $\phi(u)$ is:

$$C_\kappa = \frac{2f^{(\kappa+1)}(0)}{(\kappa+1)!} = \frac{2}{(\kappa+1)!}\frac{(2(\kappa+1))!}{(\kappa+1)!4^{\kappa+1}} = \frac{2}{4^{\kappa+1}}\binom{2\kappa+2}{\kappa+1}$$

For any $\kappa \geq 1$, this coefficient is less than 1. For example, for $\kappa = 1$, $C_1 = \frac{2}{16}\binom{4}{2} = \frac{12}{16} = \frac{3}{4} < 1$. In general, using Stirling's approximation $\binom{2n}{n} \leq \frac{4^n}{\sqrt{\pi n}}$, we have:

$$C_\kappa = \frac{2}{4^{\kappa+1}}\binom{2\kappa+2}{\kappa+1} \leq \frac{2}{4^{\kappa+1}}\frac{4^{\kappa+1}}{\sqrt{\pi(\kappa+1)}} = \frac{2}{\sqrt{\pi(\kappa+1)}} < 1 \qquad \text{for } \kappa \geq 1.$$

Since $\phi(u) = C_\kappa u^{\kappa+1} + \mathcal{O}(u^{\kappa+2})$ and the leading coefficient $C_\kappa$ is less than 1, there exists a sufficiently small $\rho_\kappa \in (0, 1)$ such that $\phi(u) \leq u^{\kappa+1}$ for all $u \in [0, \rho_\kappa]$. Specifically, you can choose any $\theta \in (C_\kappa, 1)$ and take $\rho_\kappa$ so that $|\phi(u) - C_\kappa u^{\kappa+1}| \leq (\theta - C_\kappa)u^{\kappa+1}$ on $[0, \rho_\kappa]$. Then

$$\phi(u) \leq \theta u^{\kappa+1} \leq u^{\kappa+1} \tag{40}$$

which makes the "exists $\rho_\kappa$" completely explicit.

Combining the recursion Eq.(34) and Eq.(40), we have

$$\delta_{t,j+1} = \phi(\delta_{t,j}) \leq (\delta_{t,j})^{\kappa+1}$$

After $q$ steps, we obtain the final result by repeated application:

$$\delta_{t,q} \leq (\delta_{t,q-1})^{\kappa+1} \leq ((\delta_{t,q-2})^{\kappa+1})^{\kappa+1} \leq \cdots (\delta_{t,0})^{(\kappa+1)^q}$$

Finally, we conclude:

$$\delta_{t,q} \leq \delta_{t,0}^{(\kappa+1)^q}$$

This establishes that the orthogonality residual decays with an order of $\kappa + 1$ at each step of the NEWTON–SCHULZ if NEWTON–SCHULZ polynomial is of degree $\kappa$. $\qquad\square$

**Corollary 5** (Final constant factor bound). *Let $\delta_0 := \sup_t \delta_{t,0} < 1$ (Remark 1). Then for all $t$ and $q \geq 1$,*

$$\delta_{t,q} \leq \delta_0^{(\kappa+1)^q}, \qquad \varepsilon_q \leq 1 - \sqrt{1 - \delta_0^{(\kappa+1)^q}}, \qquad \chi_q = (1 - \varepsilon_q)^{-1} \leq \left[1 - \delta_0^{(\kappa+1)^q}\right]^{-1/2}.$$

*Proof.* Combine Lemma 3 with Lemma 1. $\qquad\square$

**Corollary 6** (Case when $\kappa \in \{1, 2\}$). *For the polynomials $p_1(\lambda) = \frac{3}{2} - \frac{1}{2}\lambda$ and $p_2(\lambda) = \frac{15}{8} - \frac{5}{4}\lambda + \frac{3}{8}\lambda^2$, the residual recursion specializes to $\delta_{t,j+1} \leq \delta_{t,j}^2$ and $\delta_{t,j+1} \leq \delta_{t,j}^3$ for all $\delta_{t,j} \in [0, 1]$. These are concrete instances of the general residual map in Lemma 2.*

*Proof.* Direct substitution into $\phi(u) = 1 - (1-u)[p_\kappa(1-u)]^2$ gives $\delta_{j+1} = \frac{\delta_j^2(3+\delta_j)}{4} \leq \delta_j^2$ since $3 + \delta \leq 4$ on $[0, 1]$ for $\kappa = 1$. Also, $\delta_{j+1} = \frac{\delta_j^3(40+15\delta_j+9\delta_j^2)}{64} \leq \delta_j^3$ since $40 + 15\delta + 9\delta^2 \leq 64$ on $[0, 1]$ for $\kappa = 2$ (check at $\delta = 1$). $\qquad\square$

## D.1 DETAILED PROOFS FOR CASE WHEN $\kappa \in \{1, 2\}$

**Residual Decay by 1st-order NEWTON–SCHULZ Polynomial.** Define the polynomial $p(\lambda)$ for NEWTON–SCHULZ steps as

$$p(\lambda) = \frac{3}{2} - \frac{1}{2}\lambda$$

Then for any fixed $t$ and any $j \geq 0$, the residual $\delta_{t,j}$ satisfies

$$\delta_{t,j+1} \leq (\delta_{t,j})^2$$

Consequently, if $\delta_{t,0} \leq \rho < 1$ is sufficiently small for all $t$, then $\delta_{t,q} \leq \rho^{2^q}$ for all $t$.

*Proof.* Deriving the function $\tau(\lambda)$ in Lemma 2,

$$\tau(\lambda) = \lambda[p(\lambda)]^2 = \lambda\left(\frac{3}{2} - \frac{1}{2}\lambda\right)^2 = \frac{9}{4}\lambda - \frac{3}{2}\lambda^2 + \frac{1}{4}\lambda^3$$

Then the derivative of $\tau(\lambda)$ is

$$\tau'(\lambda) = \frac{9}{4} - 3\lambda + \frac{3}{4}\lambda^2 = \frac{3}{4}(\lambda - 1)(\lambda - 3)$$

Since $\tau'(\lambda) \geq 0$ for $\lambda \in [0, 1]$ and $\tau(1) = 1$, we can apply Lemma 2. The orthogonality residual $\delta_{t,j}$ is updated by one NEWTON–SCHULZ step as

$$\delta_{t,j+1} = \phi(\delta_{t,j}) = 1 - (1 - \delta_{t,j})[p(1 - \delta_{t,j})]^2$$
$$= 1 - (1 - \delta_{t,j})\left(\frac{3}{2} - \frac{1}{2}(1 - \delta_{t,j})\right)^2 = \frac{\delta_{t,j}^2(3 + \delta_{t,j})}{4} \leq \delta_{t,j}^2$$

By induction, after $q$ steps, we get the relationship, $\delta_{t,q} \leq (\delta_{t,0})^{2^q}$. Given the assumption that $\delta_{t,0} \leq \rho < 1$ for all $t$, the conclusion follows directly:

$$\delta_{t,q} \leq \rho^{2^q}$$

This shows that the orthogonality residual decreases quadratically with each iteration, which is a very rapid rate of convergence. $\qquad\square$

**Residual Decay by 2nd-order NEWTON–SCHULZ Polynomial.** Define the polynomial $p(\lambda)$ for NEWTON–SCHULZ steps as

$$p(\lambda) = \frac{15}{8} - \frac{5}{4}\lambda + \frac{3}{8}\lambda^2$$

Then for any fixed $t$ and any $j \geq 0$, the residual $\delta_{t,j}$ satisfies

$$\delta_{t,j+1} \leq (\delta_{t,j})^3$$

Consequently, if $\delta_{t,0} \leq \rho < 1$ is sufficiently small for all $t$, then $\delta_{t,q} \leq \rho^{3^q}$ for all $t$.

*Proof.* Deriving the function $\tau(\lambda)$ in Lemma 2,

$$\tau(\lambda) = \lambda[p(\lambda)]^2 = \lambda\left(\frac{15}{8} - \frac{5}{4}\lambda + \frac{3}{8}\lambda^2\right)^2$$

Then the derivative of $\tau(\lambda)$ is

$$\tau'(\lambda) = \left(\frac{15}{8} - \frac{5}{4}\lambda + \frac{3}{8}\lambda^2\right)^2 + 2\lambda\left(\frac{15}{8} - \frac{5}{4}\lambda + \frac{3}{8}\lambda^2\right)$$
$$= \left(\frac{15}{8} - \frac{5}{4}\lambda + \frac{3}{8}\lambda^2\right)\left(\frac{15}{8} + \frac{3}{4}\lambda + \frac{3}{8}\lambda^2\right) = \frac{1}{64}\left(20 + (5 - 3\lambda)^2\right)\left(4 + (1 + \lambda)^2\right)$$

Since $\tau'(\lambda) \geq 0$ for $\lambda \in [0,1]$ and $\tau(1) = 1$, we can apply Lemma 2. The orthogonality residual $\delta_{t,j}$ is updated by one NEWTON–SCHULZ step as

$$
\begin{aligned}
\delta_{t,j+1} = \phi(\delta_{t,j}) &= 1 - (1 - \delta_{t,j})[p(1 - \delta_{t,j})]^2 \\
&= 1 - (1 - \delta_{t,j})\left(\frac{15}{8} - \frac{5}{4}(1 - \delta_{t,j}) + \frac{3}{8}(1 - \delta_{t,j})^2\right)^2 \\
&= 1 - (1 - \delta_{t,j})\left(1 + \frac{1}{2}\delta_{t,j} + \frac{3}{8}\delta_{t,j}^2\right)^2 \\
&= 1 - (1 - \delta_{t,j})\left(1 + \delta_{t,j} + \delta_{t,j}^2 + \frac{3}{8}\delta_{t,j}^3 + \frac{9}{64}\delta_{t,j}^4\right) \\
&= \frac{\delta_{t,j}^3(40 + 15\delta_{t,j} + 9\delta_{t,j}^2)}{64} \leq \delta_{t,j}^3
\end{aligned}
$$

By induction, after $q$ steps, we get the relationship, $\delta_{t,q} \leq (\delta_{t,0})^{3^q}$. Given the assumption that $\delta_{t,0} \leq \rho < 1$ for all $t$, the conclusion follows directly:

$$
\delta_{t,q} \leq \rho^{3^q}
$$

This shows that the orthogonality residual decreases cubically with each iteration, which is a very rapid rate of convergence. $\qquad\square$

# E  WALL-CLOCK VIA COMPUTATIONAL COMPLEXITY.

At each iteration $t$, MUON performs an orthogonalization of the momentum matrix $M_t$ via either SVD or NEWTON–SCHULZ (NS). We write $\Phi_{\text{gemm}}$ for the effective GEMM throughput (FLOP/s), and $\Phi_{\text{svd}}$ for the effective throughput of the SVD routine. In practice, $\Phi_{\text{gemm}} \gg \Phi_{\text{svd}}$ due to far higher hardware utilization of GEMM.

**Per-iteration Orthogonalization FLOPs.**  For a single layer index by $\ell$ with $m \le n$ (for $m > n$, apply to $M_t^\top$ and transpose-trick):

- **MUON with SVD.** A thin SVD of $M_t \in \mathbb{R}^{m \times n}$ and extracting the polar factor $U_t V_t^\top$ costs (Golub & Reinsch, 1971):

$$\text{FLOPs}_{\text{svd}}^{(\ell)}(m, n) = \Theta(4m^2 n + 8m^3)$$

  Wall-clock time per layer is $t_{\text{svd}}^{(\ell)} = \text{FLOPs}_{\text{svd}}^{(\ell)}(m, n)/\Phi_{\text{svd}}$.

- **MUON with NS ($q$-steps, $\kappa$-degree).** NEWTON–SCHULZ follows Horner's rule when recursively updating the scaled momentum matrix using the NEWTON–SCHULZ polynomial. NEWTON–SCHULZ forms $A = XX^\top \in \mathbb{R}^{m \times m}$ and applies the degree-$\kappa$ polynomial to $X$ via Horner's rule. Each NS step needs one $m \times n$ by $n \times m$ GEMM to build $A$ and $\kappa$ multiplies $AY$ (each $m \times m$ by $m \times n$). Hence,

$$\text{FLOPs}_{\text{ns}}^{(\ell)}(m, n; q, \kappa) = \Theta(2q(\kappa + 1)m^2 n)$$

  Wall-clock time per layer is $t_{\text{ns}}^{(\ell)} = \text{FLOPs}_{\text{ns}}^{(\ell)}(m, n; q, \kappa)/\Phi_{\text{gemm}}$.

**Lemma 11.** *For a layer with $m \le n$, the wall-clock time ratio between* MUON *with* SVD *and* MUON *with* NEWTON–SCHULZ *($q$-steps, $\kappa$-degree) is,*

$$\frac{t_{svd}^{(\ell)}}{t_{ns}^{(\ell)}} = \frac{\Theta(4m^2 n + 8m^3)}{\Theta(2q(\kappa + 1)m^2 n)} \cdot \frac{\Phi_{gemm}}{\Phi_{svd}} = \Theta\left(\frac{2 + 4\frac{m}{n}}{q(\kappa + 1)}\right) \cdot \underbrace{\frac{\Phi_{gemm}}{\Phi_{svd}}}_{\text{efficiency ratio} \gg 1}$$

**Discussion of Lemma 11.**  With the practical setting $q \in \{2, 3\}$ and $\kappa \in \{1, 2\}$, the algebraic factor $\frac{2 + 4(m/n)}{q(\kappa + 1)}$ is $\mathcal{O}(0.3 \sim 1)$, so the wall-clock speedup is essentially the GEMM/SVD efficiency ratio. On modern GPUs, $\Phi_{\text{gemm}}/\Phi_{\text{svd}}$ is often $4 \sim 10$, so NS typically yields a multi-$\times$ speedup per iteration over SVD, matching empirical observations in Figure 1.

**Practical Interpretation.**  NEWTON–SCHULZ scales linearly in $q$ (accuracy knob) and uses only GEMMs, which map efficiently to GPUs. Exact SVD pays an additional $m^3$ term and typically incurs larger constants. Hence for small $q$ and modest $\kappa$, NEWTON–SCHULZ is substantially cheaper per update than a full SVD while achieving near-exact orthogonalization in practice.

# F NUMERICAL EXPERIMENTS DETAIL

## F.1 EXPERIMENTAL SETTING

**Task and metric.** All experiments are conducted on CIFAR-10 (50k train / 10k test) with the standard channel-wise normalization mean $= (0.4914, 0.4822, 0.4465)$ and std $= (0.2470, 0.2435, 0.2616)$.

We report cross-entropy train loss and test loss. Plots show the $(mean \pm 1 \cdot std)$ over 5 independent runs, and we also report the wall-clock time (in seconds) accumulated from the beginning of training.

**Model.** We use a compact convolutional network (CIFARNET, approximately $2M$ parameters): a fixed whitening $2 \times 2$ convolution (weights frozen, bias trainable), followed by three ConvGroups (each: $3 \times 3$ conv $\rightarrow$ MaxPool2d(2) $\rightarrow$ GELU $\rightarrow$ $3 \times 3$ conv $\rightarrow$ GELU) with widths $(64, 256, 256)$ and a linear head. BatchNorm layers keep affine weights frozen (biases are trainable). For stability, the first portion of each convolution is Dirac-initialized, and the linear head is variance-normalized.

**Hyperparameters and schedules.** Unless stated otherwise, we train for 50 epochs with a batch size of $B = 512$ on the GPU (the CPU fallback uses $B = 256$). Each run uses a different seed.

- **Main optimizer (one of MUON ($q$), MUON (SVD), or SGD-M)** on convolutional filters: learning rate $\eta = 0.0860632$, momentum $\beta = 0.730778$.
- **Auxiliary SGD** (whitening bias, BatchNorm biases, linear head): learning rates $\eta_{\text{other}} = 1.4949 \times 10^{-3}$ and $\eta_{\text{head}} = 1.72446$ with momentum $0.989703$ (Nesterov).
- **Label smoothing** $0.2$, and **gradient clipping** $\| \cdot \|_2 \leq 1.0$.
- **Schedule:** The same warm-up + cosine schedule is applied to *all* optimizers: a 5% linear warm-up of total steps followed by cosine decay to $0$. Schedulers step once per update.

**Evaluation.** At the end of each epoch, we evaluate the test loss using the same normalization (no augmentation). We record (i) epoch-wise training loss, (ii) test loss, and (iii) the cumulative wall-clock time. Results are aggregated across 5 runs. Figures report $(mean \pm 1 \cdot std)$ and include both epoch-aligned and time-aligned views to disentangle statistical and systemic effects.

## F.2 NEWTON–SCHULZ STEPS ($q$) ABLATIONS.

**Optimizers under comparison.** We compare:

1. **MUON $q$-step**: SGD with momentum followed by an orthogonalization of the momentum matrix using $q$ NEWTON–SCHULZ steps (Algorithm 1); $q \in \{0, 1, 2, 3\}$. The case $q = 0$ is a normalization-only ablation (no orthogonalization).
2. **MUON (SVD)**: MUON with an exact polar step $UV^\top$ via SVD.
3. **SGD-M**: SGD with momentum baseline with identical schedules.

All methods share identical schedules and auxiliary updates (Appendix F.1).

**Orthogonalization details.** For MUON with $q$-NEWTON–SCHULZ steps, we use NEWTON–SCHULZ polynomial, $p_\kappa$

$$p(\lambda) = a + b\lambda + c\lambda^2, \qquad (a, b, c) = (15/8, -5/4, 3/8),$$

applied as $X \leftarrow aX + (bA + cA^2)X$ with $A = XX^\top$ and $X = M/\|M\|_F$; if the matrix is *tall,* we employ the transpose trick. The SVD variant normalizes $M$ by its Frobenius norm and returns $UV^\top$. In all MUON variants, each weight tensor is re-normalized once per step to the Frobenius norm $\sqrt{\text{out\_channels}}$ to stabilize scales.

**Ablations on $q$.** Our main comparison varies the number of NEWTON–SCHULZ steps $q \in \{1, 2, 3\}$ while holding the polynomial fixed to $p_\kappa$, and contrasts these with MUON with SVD and SGD-M. All other components (architecture, schedules, augmentation) are kept identical across methods to ensure a fair comparison.

## G  ADDITIONAL NUMERICAL EXPERIMENTS

**Optimizers compared:**

- SGD with Momentum
- Muon (Newton–Schulz) with 1, 2, 3 steps per update
- Muon with exact SVD (polar factor)

**Dataset and Model:**

- MNIST (60K) / MLP (0.5M)
- CIFAR-10 (50K) / CifarNet (2M) : Main text
- CIFAR-100 (50K) / ResNet-18 (11.2M)
- Tiny-ImageNet (100K) / WideResNet-28-10 (36.6M)
- FineWeb (10M tokens from sample-10BT) / NanoGPT (124.2M) & GPT-2 (1.3B)
    - block_size = 1024
    - num_blocks = (10M - 1) // 1024 = 9,765 sequences (for causal LM)
    - n_layer: 12 & 24 (transformer blocks)
    - n_head: 12 & 16
    - n_embd: 768 & 2048
    - Vocab size: tokenizer.vocab_size from GPT2TokenizerFast (50257)
    - Token embedding: nn.Embedding(vocab_size, 768)
    - Positional embedding: nn.Embedding(block_size, 768)
    - Each of the 12 transformer blocks
        * LayerNorm $\rightarrow$ multi-head causal self-attention (QKV + output projection) $\rightarrow$ residual
        * LayerNorm $\rightarrow$ 4$\times$-wide MLP (3072 hidden) $\rightarrow$ residual
    - Final LayerNorm
    - Total n_params: 124.2M & 1313.63M (1.31B)

**Hyper-parameters**

- **MLP on MNIST:**
    - model: $784 \rightarrow 512 \rightarrow 256 \rightarrow 10$
    - learning rate: 0.08; momentum: 0.7
    - 256 batch size; 50 epochs, 5 runs
- **ResNet-18 on CIFAR-100:**
    - model: torchvision.models.resnet18
    - learning rate: 0.08; momentum: 0.7
    - 512 batch size; 50 epochs, 5 runs
- **WideResNet-28-10 on Tiny-ImageNet:**
    - model: WideResNet(depth=28, widen_factor=10, num_classes=200, drop_rate=0.0)
    - learning rate: 0.08; momentum: 0.7
    - 128 batch size; 30 epochs, 3 runs
- **NanoGPT & GPT-1.3B on FineWeb:**
    - model: NanoGPT & GPT-1.3B
    - learning rate: 0.02; momentum: 0.95; batch size: 8
    - 10M training tokens, 1M validation tokens
    - 8 RTX-3090 GPUs
    - max steps: 6000

## G.1 MLP ON MNIST

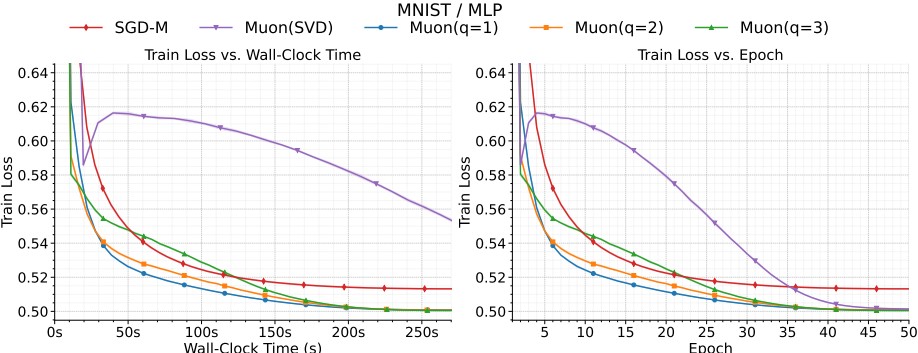

Figure 2: Train losses of MLP on MNIST across wall-clock time and epochs

Table 2: Wall-clock time training performance of MLP (0.5M) on MNIST dataset

| | Wall-clock time train loss | | | | |
|---|---|---|---|---|---|
| Optimizer | 50 sec | 100 sec | 150 sec | 200 sec | 250 sec |
| **SGD-M** | 0.550 | 0.525 | 0.517 | 0.514 | 0.513 |
| **Muon with SVD** | 0.616 | 0.612 | 0.600 | 0.583 | 0.565 |
| **Muon ($q$=1)** | 0.526 | 0.514 | 0.506 | 0.502 | 0.501 |
| **Muon ($q$=2)** | 0.532 | 0.518 | 0.509 | 0.503 | 0.501 |
| **Muon ($q$=3)** | 0.548 | 0.529 | 0.511 | 0.503 | 0.501 |

## G.2 CIFARNET ON CIFAR-10

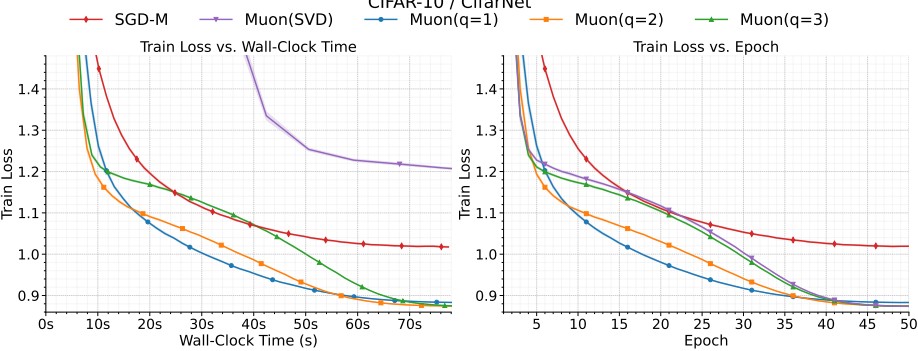

Figure 3: Train losses of CifarNet on CIFAR-10 across wall-clock time and epochs

Table 3: Wall-clock time training performance of CifarNet (2M) on CIFAR-10

| | Wall-clock time train loss | | | | |
|---|---|---|---|---|---|
| Optimizer | 10 sec | 20 sec | 30 sec | 40 sec | 50 sec | 60 sec |
| **SGD-M** | 1.366 | 1.157 | 1.084 | 1.045 | 1.021 | 1.010 |
| **Muon with SVD** | 2.211 | 1.334 | 1.227 | 1.207 | 1.193 | 1.182 |
| **Muon ($q$=1)** | 1.130 | 1.014 | 0.945 | 0.905 | 0.888 | 0.884 |
| **Muon ($q$=2)** | 1.129 | 1.064 | 1.001 | 0.932 | 0.888 | 0.876 |
| **Muon ($q$=3)** | 1.191 | 1.145 | 1.079 | 0.990 | 0.905 | 0.876 |

## G.3 RESNET-18 ON CIFAR-100

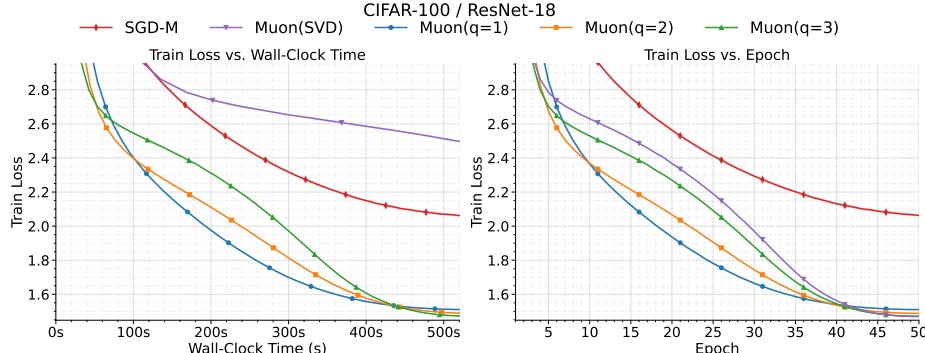

Figure 4: Train losses of ResNet-18 on CIFAR-100 across wall-clock time and epochs

Table 4: Wall-clock time training performance of ResNet-18 (11.2M) on CIFAR-100

| | **Wall-clock time train loss** | | | | |
|---|---|---|---|---|---|
| **Optimizer** | **100 sec** | **200 sec** | **300 sec** | **400 sec** | **500 sec** |
| **SGD-M** | 3.096 | 2.612 | 2.324 | 2.157 | 2.072 |
| **Muon with SVD** | 3.202 | 2.767 | 2.668 | 2.594 | 2.522 |
| **Muon ($q$=1)** | 2.427 | 1.986 | 1.716 | 1.563 | 1.514 |
| **Muon ($q$=2)** | 2.407 | 2.127 | 1.826 | 1.584 | 1.495 |
| **Muon ($q$=3)** | 2.554 | 2.329 | 1.985 | 1.624 | 1.481 |

## G.4 WIDERESNET-28-10 ON TINY-IMAGENET

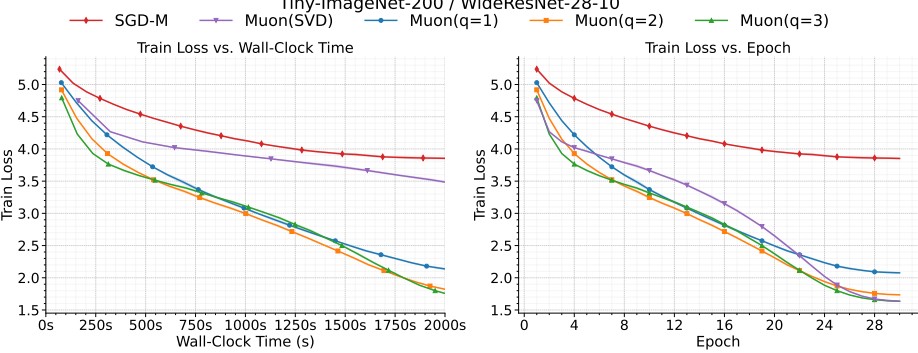

Figure 5: Train losses of WideResNet-28-10 on Tiny-ImageNet across wall-clock time and epochs

Table 5: Wall-clock time training performance of WideResNet-28-10 (36.6M) on Tiny-ImageNet

| | **Wall-clock time train loss** | | | | |
|---|---|---|---|---|---|
| **Optimizer** | **400 sec** | **800 sec** | **1200 sec** | **1600 sec** | **2000 sec** |
| **SGD-M** | 4.694 | 4.299 | 4.045 | 3.914 | 3.857 |
| **Muon with SVD** | 4.267 | 4.020 | 3.846 | 3.734 | 3.520 |
| **Muon ($q$=1)** | 4.035 | 3.369 | 2.903 | 2.495 | 2.144 |
| **Muon ($q$=2)** | 3.766 | 3.246 | 2.815 | 2.311 | 1.856 |
| **Muon ($q$=3)** | 3.666 | 3.317 | 2.929 | 2.372 | 1.800 |

## G.5 NanoGPT on FineWeb

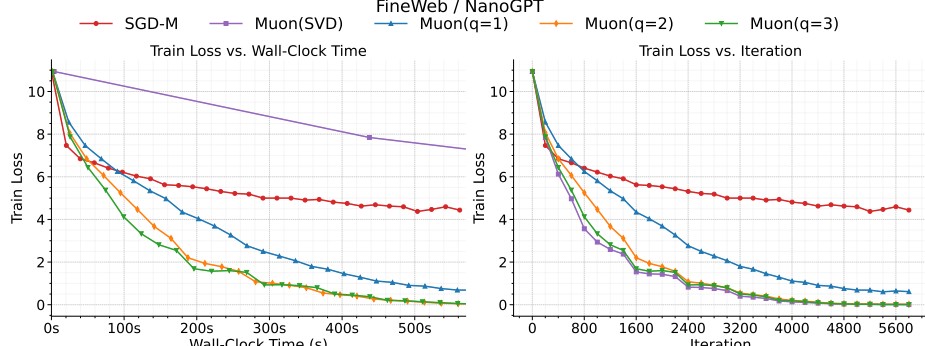

Figure 6: Train losses of NanoGPT on FineWeb across wall-clock time and epochs

Table 6: Wall-clock time training performance of NanoGPT (124M) on FineWeb

| Optimizer | Wall-clock time train loss | | | | | |
|---|---|---|---|---|---|---|
| | 100 sec | 200 sec | 300 sec | 400 sec | 500 sec | 600 sec |
| SGD-M | 10.938 | 7.656 | 6.000 | 5.375 | 4.938 | 4.594 |
| Muon with SVD | 10.250 | 9.188 | 8.625 | 8.062 | 7.594 | 7.125 |
| Muon ($q$=1) | 5.969 | 4.094 | 2.469 | 1.367 | 0.879 | 0.637 |
| Muon ($q$=2) | 5.000 | 2.344 | 1.047 | 0.395 | 0.157 | 0.030 |
| Muon ($q$=3) | 3.984 | 2.125 | 1.180 | 0.523 | 0.142 | 0.026 |

## G.6 GPT-2 based model (1.3B) on FineWeb

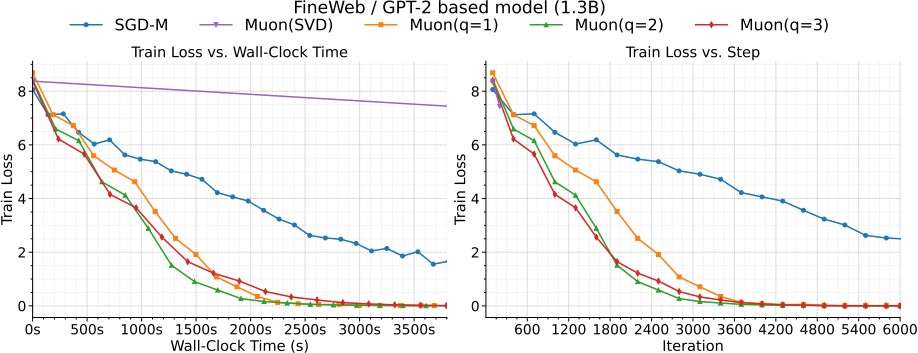

Figure 7: Train losses of GPT-2 based model (1.3B) on FineWeb across wall-clock time and epochs

Table 7: Wall-clock time training performance of GPT-2 based model (1.3B) on FineWeb

| Optimizer | Wall-clock time train loss | | | | | |
|---|---|---|---|---|---|---|
| | 600 sec | 1200 sec | 1800 sec | 2400 sec | 3000 sec | 3600 sec |
| SGD-M | 6.4062 | 5.3750 | 4.2812 | 3.2969 | 2.4219 | 2.0156 |
| Muon with SVD | 8.2280 | 8.0810 | 7.9340 | 7.7870 | 7.6400 | 7.4930 |
| Muon ($q$=1) | 6.0625 | 3.5156 | 1.0781 | 0.1436 | 0.1338 | 0.0109 |
| Muon ($q$=2) | 5.1562 | 2.8906 | 0.5898 | 0.1133 | 0.0349 | 0.0113 |
| Muon ($q$=3) | 5.6562 | 3.2031 | 1.2188 | 0.4609 | 0.1201 | 0.0286 |

# H ADDITIONAL ABLATION EXPERIMENTS

## H.1 NEWTON–SCHULZ–POLYNOMIAL DEGREE-$\kappa$ ABLATIONS.

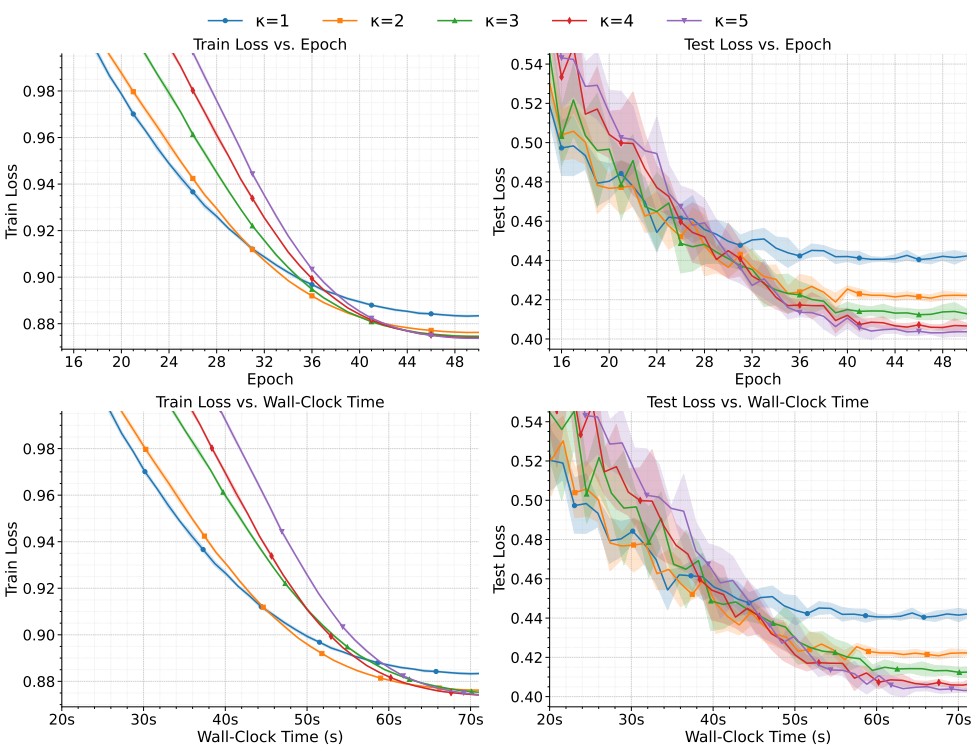

Figure 8: **Degree $\kappa$ sweep.** NEWTON–SCHULZ polynomial degree $\kappa \in \{1, \ldots, 5\}$ at fixed $q = 3$. Larger $\kappa$ improves train/test loss but increases time per step.

Beyond the step-sweep in Fig. 1, we perform a controlled ablation that varies the *degree $\kappa$* of the NEWTON–SCHULZ polynomial while fixing the number of NEWTON–SCHULZ steps to $q = 3$ for all variants. Concretely, we instantiate a family of MUON with degree-$\kappa$ polynomial optimizers whose orthogonalization step applies, per layer, the update

$$X \leftarrow p_\kappa(XX^\top)X, \quad X = \frac{M}{\|M\|_F}$$

where $M$ is the momentum matrix, and $p_\kappa(\lambda) = \sum_{m=0}^{\kappa} c_m \lambda^m$ is the degree-$\kappa$ NEWTON–SCHULZ polynomial that matches the derivatives of $\lambda^{-1/2}$ at $\lambda = 1$ up to order $\kappa$. The coefficients are generated analytically from the Taylor series of $\lambda^{-1/2}$ at 1, ensuring the residual recursion $\delta_{j+1} \leq (\delta_j)^{\kappa+1}$ (Lemma 3). Tall matrices are handled using the transpose trick.

**Ablation on degree $\kappa$.** Fixing $q = 3$, increasing the degree $\kappa \in \{1, \ldots, 5\}$ improves optimization (loss drops faster at a fixed epoch) but lengthens each step, yielding a clear accuracy–time trade-off (see Fig. 8). (This mirrors the theory that the residual contracts as $\delta_{j+1} \leq \delta_j^{\kappa+1}$ (Lemma 3), while computation scales with polynomial evaluations.)

**Compared optimizers.** We evaluate $\kappa \in \{1, 2, 3, 4, 5\}$ under a fixed iteration budget $q = 3$. This ablation isolates the effect of the polynomial degree $\kappa$ at a fixed step $q$, directly testing the theory-driven prediction that larger $\kappa$ accelerates orthogonality residual decay (see Table 1).

## H.2 RANK–DEPENDENCE.

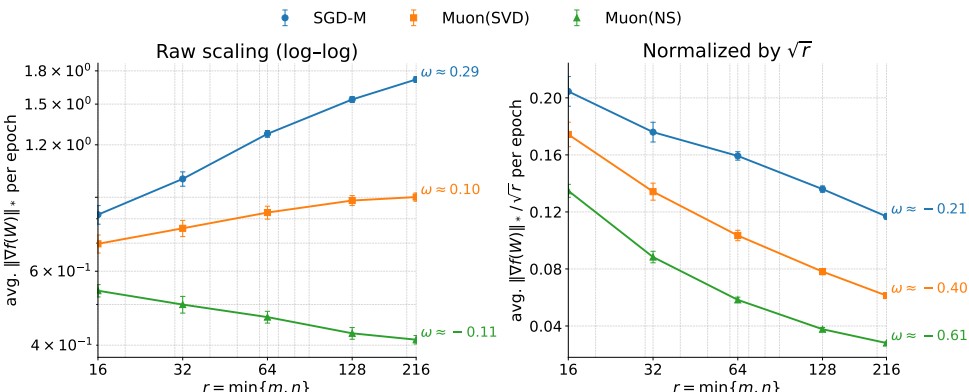

Figure 9: **Rank dependence.** Epoch-averaged gradient norm vs. rank $r$ (left: raw log–log; right: normalized by $\sqrt{r}$). MUON variants are nearly $r$-invariant; SGD-M scales up with rank.

Table 8: **log–log slopes** ($\omega$) from Fig. 9.

| Method | $\omega$ (**raw**) | $\omega$ (**normalized by** $\sqrt{r}$ ) |
|---|---|---|
| SGD-M | 0.292 | $-0.208$ |
| MUON (SVD) | 0.102 | $-0.398$ |
| MUON (NS) | $-0.106$ | $-0.606$ |

**Rank dependence.** We vary the monitored layer's effective rank $r \in \{16, 32, 64, 128, 216\}$ and plot the epoch-averaged $\|\nabla f(W)\|_*$ on a log–log scale. SGD-M shows a positive slope $\approx 0.3$ (grows with $r$), whereas MUON and its variants are nearly flat. After normalizing by $\sqrt{r}$, the two MUON variants follow the predicted $r^{-1/2}$ trend, while SGD-M remains non-flat (see Fig. 9).

**Goal and prediction.** We test how the nuclear–norm of the per–step gradient on a monitored layer scales with the layer's dimension parameter $r := \min\{m, n\}$ for a weight $W \in \mathbb{R}^{m \times n}$. The theory predicts that orthogonalizing the momentum suppresses the $r$-growth which occurs in either MUON with NEWTON–SCHULZ or MUON with SVD, while SGD-M has $\sqrt{r}$ dependence.

**Controlling $r$.** We vary the out-channels of the first convolution in the first `ConvGroup` and keep all other widths fixed. For this layer, the weight tensor has shape $[\text{out}, \text{in}, 3, 3]$ with $\text{in} = 24$ (from the fixed "whitening" $2 \times 2$ conv). Flattening by rows yields a matrix of shape $m \times n$ with $m = \text{out}$ and $n = \text{in} \cdot 3 \cdot 3 = 24 \cdot 9 = 216$, so $r = \min\{\text{out}, 216\}$. We sweep $\text{out} \in \{16, 32, 64, 128, 216\}$, hence $r \in \{16, 32, 64, 128, 216\}$.

**Optimizers under comparison.** We compare:

- **SGD-M** (baseline),
- **MUON** (SVD) (exact polar $UV^\top$), and
- **MUON** (NS) with $q = 3$ Newton–Schulz steps.

All methods share identical schedules and auxiliary updates (Appendix F.1).

**Results.** Across $r \in \{16, 32, 64, 128, 216\}$, SGD-M exhibits a positive log–log slope (about $0.3$), while both MUON variants are nearly flat. After dividing by $\sqrt{r}$, the MUON curves show a slope of about $-0.5$ (MUON with SVD: -0.4 and MUON with NEWTON–SCHULZ: -0.61) , as predicted, whereas SGD-M becomes almost flat (slope with $-0.21$).

## H.3 BATCH SIZE $B$ ABLATIONS.

We sweep $B \in \{64, 128, 256, 512, 1024\}$ with MUON ($q = 3$) under identical schedules and report epoch–aligned and time–aligned views (Fig. 10). The schedule is step–based, so larger $B$ implies fewer total steps over $E$ epochs. We report both epoch–aligned and wall–clock-aligned curves.

From a systems perspective, increasing $B$ improves throughput up to a regime of diminishing returns. From an optimization perspective, a larger $B$ reduces gradient noise but also decreases the frequency of orthogonalization steps per epoch ($N_{\text{proj}} = N_{\text{train}}/B$).

In our runs, the best time–to-accuracy is achieved for a large batch size $B = 1024$, while a small $B$ suffers from noise, taking a greater amount of time.

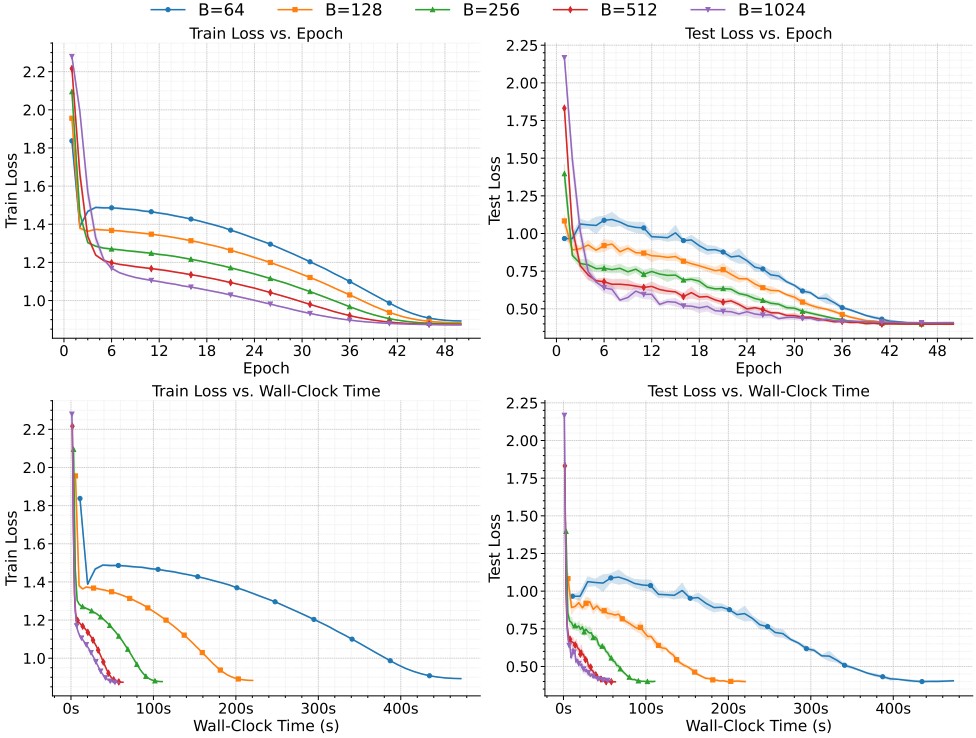

Figure 10: **Batch size.** Train/test loss vs. epoch and wall-clock for $B \in \{64, 128, 256, 512, 1024\}$.

## H.4 DEGREE-2 NS POLYNOMIAL VS. AD-HOC DEGREE-2 NS POLYNOMIAL

Figure 11: **NS polynomial vs. Ad-hoc polynomial.** Train/test loss vs. epoch and wall-clock.

**Analysis of $p_{\text{ad-hoc}}$**

Let $p_{\text{ad-hoc}}(\lambda) = 3.4445 - 4.7750\,\lambda + 2.0315\,\lambda^2$ and $\tau(\lambda) = \lambda p(\lambda)^2$.

The statement that $\tau_{\text{ad-hoc}}(\lambda)$ is monotone non-decreasing on $[0,1]$ is false, and the underlying condition that $p_{\text{ad-hoc}}(\lambda) \in [0,1]$ is also not met.

On the interval $[0,1]$, the range of $p(\lambda)$ is $[0.701, 3.4445]$. Since this range is not contained within $[0,1]$, the premise $p(\lambda) \in [0,1]$ is false.

A function is monotone non-decreasing if its derivative is greater than or equal to zero over the entire interval. The derivative of $\tau(\lambda)$ is:

$$\tau'_{\text{ad-hoc}}(\lambda) = \frac{d}{d\lambda}\tau_{\text{ad-hoc}}(\lambda) = 20.635\lambda^4 - 77.5824\lambda^3 + 110.3556\lambda^2 - 65.781\lambda + 11.8641$$

To check for monotonicity, we can evaluate the derivative at the endpoints of the interval $[0,1]$:

- At $\lambda = 0$: $\tau'_{\text{ad-hoc}}(0) = 11.8641$
- At $\lambda = 1$: $\tau'_{\text{ad-hoc}}(1) = 20.635 - 77.5824 + 110.3556 - 65.781 + 11.8641 = -0.5087$

Since $\tau'_{\text{ad-hoc}}(0) > 0$ and $\tau'_{\text{ad-hoc}}(1) < 0$, the derivative changes from positive to negative within the interval. This means the function $\tau_{\text{ad-hoc}}(\lambda)$ increases for a portion of the interval and then decreases.

Therefore, the claim that $\tau_{\text{ad-hoc}}(\lambda)$ is monotone non-decreasing on $[0,1]$ is false. The function has a local maximum at approximately $\lambda \approx 0.308$.

Hence, the monotonicity premise of Lemma 2 fails for $p_{\text{ad-hoc}}$, even though $\tau(1) = p(1)^2 = 0.701^2 < 1$ holds. This explains why our guarantees apply to $p_\kappa$ but not to the ad-hoc quadratic, which remains an empirical heuristic.

