# OpenReview forum: "Convergence of Muon with Newton-Schulz"
_ICLR.cc/2026/Conference — ICLR 2026 Poster_

### Official Review · Reviewer_V23p · 2025-10-29

**Soundness:** 4
**Presentation:** 4
**Contribution:** 4
**Rating:** 8
**Confidence:** 4

**Summary:**

This paper studies the MUON optimizer, which is designed for matrix-structured parameters. In practice, MUON replaces expensive SVD computations with a few Newton-Schulz (NS) steps to orthogonalize its momentum. Previous analyses only covered the idealized SVD version. This paper gives the first convergence proof for the practical version that actually uses NS.

The main result shows that MUON with NS converges to a stationary point at the same rate as the SVD version, up to a small constant that converges doubly exponentially fast to 1 as the number of NS steps increases. The paper also shows that MUON improves rank dependence compared to SGD with momentum. Experiments on CIFAR-10 confirm the theoretical findings.

**Strengths:**

The paper is well written and well organized. The intuition behind the proofs is explained in a way that makes the technical results easy to follow.

This work provides the first convergence proof for MUON when the orthogonalization step is done with Newton–Schulz iterations. This directly corresponds to how MUON is implemented and used in practice, rather than analyzing an idealized or simplified variant. This is an important contribution because most previous works relied on the unrealistic assumption of an exact SVD-based orthogonalization, which is not used in real training.

The theoretical assumptions are reasonable and consistent with standard conditions in stochastic optimization. They are stated clearly and make sense for modern deep learning models. It is also appreciated that the authors do not assume extra smoothness or artificial conditions beyond what is standard for first-order methods.

The main theorems are precise, and the proofs are well-structured and clear. The paper shows in detail how the Newton-Schulz approximation error decreases and how it influences the convergence rate. The results are intuitive and easy to interpret.

Finally, the comparisons between MUON with NS, MUON with SVD, and SGD with momentum are well designed. They are based on the same assumptions, metrics, and convergence definitions, which makes the differences in behavior easy to understand. This unified view helps highlight the benefit of MUON’s matrix-aware design.

**Weaknesses:**

The paper is already strong in both quality and presentation, but there are still a few limitations.

The numerical experiments are pretty limited (which is fine given how popular Muon is), but it would be better to have larger-scale experiments when comparing methods. The results are shown only on CIFAR-10 using a small CNN. While this setup is enough to confirm the theory, it does not demonstrate the behavior of MUON in more realistic large-scale scenarios. Testing the optimizer on larger architectures, such as Transformers or vision models with high-rank layers, would make the empirical section much more convincing and highlight the practical impact of the theoretical results.

Although the theoretical assumptions are standard and well-justified, their validity in large deep networks can be questioned. In practice, smoothness and bounded variance often fail to hold exactly. For example, Transformer gradients can have heavy-tailed distributions, which makes the variance assumption only approximately true.

**Questions:**

See weaknesses

---

> ### Author Response · Authors · 2025-11-22
>
> We sincerely appreciate the time and effort you have invested in reviewing our paper, and we are grateful for your positive assessment of our theoretical contributions, including recognizing that this is the first convergence analysis of Muon with a finite number of Newton–Schulz steps.
>
> While our main contributions are theoretical, we agree that stronger empirical support improves the paper. In response to your comments, we have substantially expanded the experimental section beyond the CifarNet/CIFAR-10 setup in the main text. We now report results on a broader range of architectures and datasets, and explicitly indicate the number of trainable parameters for each model:
> - MLP (0.5M parameters) / MNIST dataset
> - CifarNet (2M) / CIFAR-10  dataset
> - ResNet-18 (11.2M) / CIFAR-100  dataset
> - WideResNet-28-10 (36.6M) / Tiny-ImageNet dataset
> - NanoGPT (124M, Transformer) / FineWeb dataset
> - GPT-2 (1.3B, Transformer) / FineWeb dataset
>
> ---
> The corresponding train losses over wall-clock time are summarized below.
>
> - **Table 1: Train loss on MNIST using MLP (0.5M)**
>
> | Optimizer       | 50 sec | 100 sec  | 150 sec  | 200 sec  | 250 sec  |
> | :--------------- | :----: | :----: | :----: | :----: | :----: |
> | SGD-M            | 0.550                | 0.525 | 0.517 | 0.514 | 0.513 |
> | Muon with SVD    | 0.616                | 0.612 | 0.600 | 0.583 | 0.565 |
> | Muon ($q$=1)     | 0.526                | 0.514 | 0.506 | 0.502 | 0.501 |
> | Muon ($q$=2)     | 0.532                | 0.518 | 0.509 | 0.503 | 0.501 |
> | Muon ($q$=3)     | 0.548                | 0.529 | 0.511 | 0.503 | 0.501 |
>
>
>
> - **Table 2: Train loss on CIFAR-10 using CifarNet (2M)**
>
> | Optimizer        | 10 sec | 20 sec | 30 sec | 40 sec | 50 sec | 60 sec |
> | :--------------- | :----: | :----: | :----: | :----: | :----: | :----: |
> | SGD-M            | 1.366  | 1.157  | 1.084  | 1.045  | 1.021  | 1.010  |
> | Muon with SVD    | 2.211  | 1.334  | 1.227  | 1.207  | 1.193  | 1.182  |
> | Muon ($q$=1)     | 1.130  | 1.014  | 0.945  | 0.905  | 0.888  | 0.884  |
> | Muon ($q$=2)     | 1.129  | 1.064  | 1.001  | 0.932  | 0.888  | 0.876  |
> | Muon ($q$=3)     | 1.191  | 1.145  | 1.079  | 0.990  | 0.905  | 0.876  |
>
>
> - **Table 3: Train loss on CIFAR-100 using ResNet-18 (11.2M)**
>
> | Optimizer        | 100 sec | 200 sec | 300 sec | 400 sec | 500 sec |
> | :--------------- | :-----: | :-----: | :-----: | :-----: | :-----: |
> | SGD-M            | 3.096   | 2.612   | 2.324   | 2.157   | 2.072   |
> | Muon with SVD    | 3.202   | 2.767   | 2.668   | 2.594   | 2.522   |
> | Muon ($q$=1)     | 2.427   | 1.986   | 1.716   | 1.563   | 1.514   |
> | Muon ($q$=2)     | 2.407   | 2.127   | 1.826   | 1.584   | 1.495   |
> | Muon ($q$=3)     | 2.554   | 2.329   | 1.985   | 1.624   | 1.481   |
>
>
> - **Table 4: Train loss on Tiny-ImageNet using WideResNet-28-10 (36.6M)**
>
> | Optimizer        | 400 sec | 800 sec | 1200 sec | 1600 sec | 2000 sec |
> | :--------------- | :-----: | :-----: | :------: | :------: | :------: |
> | SGD-M            | 4.694   | 4.299   | 4.045    | 3.914    | 3.857    |
> | Muon with SVD    | 4.267   | 4.020   | 3.846    | 3.734    | 3.520    |
> | Muon ($q$=1)     | 4.035   | 3.369   | 2.903    | 2.495    | 2.144    |
> | Muon ($q$=2)     | 3.766   | 3.246   | 2.815    | 2.311    | 1.856    |
> | Muon ($q$=3)     | 3.666   | 3.317   | 2.929    | 2.372    | 1.800    |
>
>
> - **Table 5: Train loss on FineWeb using NanoGPT (124M)**
>
> | Optimizer        | 100 sec | 200 sec | 300 sec | 400 sec | 500 sec | 600 sec |
> | :--------------- | :-----: | :-----: | :-----: | :-----: | :-----: | :-----: |
> | SGD-M            | 10.938  | 7.656   | 6.000   | 5.375   | 4.938   | 4.594   |
> | Muon with SVD    | 10.250  | 9.188   | 8.625   | 8.062   | 7.594   | 7.125   |
> | Muon ($q$=1)     | 5.969   | 4.094   | 2.469   | 1.367   | 0.879   | 0.637   |
> | Muon ($q$=2)     | 5.000   | 2.344   | 1.047   | 0.395   | 0.157   | 0.030   |
> | Muon ($q$=3)     | 3.984   | 2.125   | 1.180   | 0.523   | 0.142   | 0.026   |
>
>
> - **Table 6: Train loss on FineWeb using GPT-2 based model (1.3B)**
>
> | Optimizer        | 600 sec | 1200 sec | 1800 sec | 2400 sec | 3000 sec | 3600 sec |
> | :--------------- | :-----: | :------: | :------: | :------: | :------: | :------: |
> | SGD-M            | 6.4062  | 5.3750   | 4.2812   | 3.2969   | 2.4219   | 2.0156   |
> | Muon with SVD    | 8.2280  | 8.0810   | 7.9340   | 7.7870   | 7.6400   | 7.4930   |
> | Muon ($q$=1)     | 6.0625  | 3.5156   | 1.0781   | 0.1436   | 0.1338   | 0.0109   |
> | Muon ($q$=2)     | 5.1562  | 2.8906   | 0.5898   | 0.1133   | 0.0349   | 0.0113   |
> | Muon ($q$=3)     | 5.6562  | 3.2031   | 1.2188   | 0.4609   | 0.1201   | 0.0286   |
>
> These experiments now cover MLPs, CNNs of varying depth and width, and Transformer-based language models with up to 1.3B parameters, which directly aligns with the kind of large-scale settings you mentioned in your review. The parameter counts span from $5 \times 10^5$ to $1.3 \times 10^9$, better reflecting practical large-scale training regimes.

---

> > ### Author Response · Authors · 2025-11-22
> >
> > We also appreciate your comment that standard assumptions such as smoothness and bounded variance may only hold approximately in very deep networks, especially in the presence of heavy-tailed gradient noise (e.g., in Transformers). Our analysis follows the standard conditions employed in convergence studies of many stochastic first-order methods for nonconvex objectives, and we view our results as a first step under these widely used assumptions, providing a theoretical foundation for the practical use of Muon with finite NS steps.
> >
> > At the same time, the new large-scale experiments with NanoGPT and the 1.3B-parameter GPT-2–based model suggest that Muon with Newton–Schulz remains stable and effective (possibly even in regimes where these assumptions are  approximate). We hope that these additions help address your concerns and further strengthen the connection between our theory and the practical usage of Muon in modern deep learning systems.

---

### Official Review · Reviewer_KDNU · 2025-10-31

**Soundness:** 4
**Presentation:** 3
**Contribution:** 2
**Rating:** 4
**Confidence:** 4

**Summary:**

The goal of this work is to provide a formal analysis and proof of the convergence of the recently introduced and potentially useful MUON optimization algorithm using momentum orthogonalization with a few newton-steps. This work fills the gap between the existing theory on MUON with an exact SVD-based polar factor. My main concern is that MUON is not widely used, with mostly a few arXiv papers referenced for its use.

Using standard analytic assumptions, tools and techniques, the authors provide complete convergence rates (dependence on the standard terms) for SGD with momentum, MUON with SVD, and MUON. Diverse numerical experiments on CIFAR-10 with modest sized CNN predominantly support their theory. Their results explain several contrasts between MUON w/SVD,MUON, and SGD w/momentum, and closes the theory-practice gap.

**Strengths:**

This paper is very clear and generally very well written.
The numerical results are extensive, but primarily in the appendix. I suggest moving several of the key plots from the appendix to the main text.

**Weaknesses:**

I did find many redundant statements of the main results throughout the paper (often with exact wording…); while I generally agree that repetition like this can enhance clarity of interpretation and impact, I found this manuscript to be excessive in that regard. The numerical results are extensive, but primarily in the appendix. I suggest moving several of the key plots from the appendix to the main text. Also, from what I saw, there is little additional information provided by the presentation of both Loss and Accuracy, so I suggest the authors pick one for the main text and move the other quantification to the Appendix. A bit more explanation of the details of the results presented in Figure 1 w.r.t. Learning dynamics across values of ‘q’ would be beneficial. A few sentences on mathematical definition and interpretation of the different norms used (Section 3.1) would be helpful for a broader audience. The definition of ‘r’on line 151 is very far removed from its first use on line 280; suggest moving. Remove variables from the abstract.

My

**Questions:**

Typos:
L53: ‘much’ -> ‘large’
L80: remove comma after ‘Shultz’
L84: ‘closer’-> ‘close’
L116: include ‘in “the” MUON…”
L293: ‘Decaying’ -> Decay

---

> ### Author Response · Authors · 2025-11-22
>
> We sincerely thank you for your careful review and constructive comments. We appreciate that you found the paper clearly written and the theoretical analysis sound, and we are grateful for your detailed suggestions on how to improve the presentation and empirical support.
>
> ### **Practical relevance and adoption of Muon**
>
> Regarding the reviewer’s comment that their main concern is “Muon is not widely used,” we respectfully disagree and would like to clarify the current situation. While Muon is indeed a relatively recent optimizer (first made public in December 2024), it has already attracted substantial attention in both the optimization and foundation-model communities. In less than a year, the original article (released as a blog post) has been cited by many follow-up papers and variants (see, e.g., the citation list of the original Muon article at <https://scholar.google.com/scholar?cites=3421207913306544575>; more recent works, including ours, are not yet counted), and the accompanying open-source repositories have quickly accumulated thousands of stars on GitHub. We believe this provides strong evidence that practitioners and researchers are actively exploring and adopting Muon in real large-scale training pipelines. It is, in fact, quite rare for a newly proposed optimization method to have such visible impact within such a short time period.
>
> Many of the citing works are currently available as arXiv preprints rather than fully archival publications, but this is typical for fast-moving areas such as large-scale optimization and foundation models. The fact that these are very recent works—including the papers we cite—does not weaken the practical relevance of Muon; if anything, it underscores that Muon is being actively used, extended, and studied by the community.
>
> Our goal in this work is precisely to provide a **rigorous convergence analysis for the practical Muon implementation based on Newton–Schulz steps**, rather than the idealized SVD-based variant (which is not used in practice). We view this as an important step toward building trust in, and further encouraging broader adoption of, Muon. In addition to prior empirical validations of Muon, we now include more extensive experiments in the revised manuscript—including large-scale models—demonstrating that Muon with a small fixed $q$ is both efficient and stable in realistic training scenarios. We hope these additions help address your concerns about the practical relevance of the algorithm we analyze.
>
>
> ### **Revisions to improve clarity and organization**
>
> We also carefully revised the paper in response to your helpful comments on presentation:
>
> - **Reducing repetition and focusing the narrative.**
>   We went through the manuscript and removed repeated expressions of the main results, while preserving a clear and coherent summary of the contributions.
>
> - **Clarifying Figure 1 and train/test distinction.**
> For Figure 1, we now explicitly state in the caption and the main text that the left column shows train loss and the right column shows test loss (previously test accuracy). This clarification, together with the updated labeling, addresses the potential confusion the reviewer mentioned and makes the distinction between training and test evaluation behavior clearer, thereby improving the readability of the comparison between methods.
>
> - **Clarifying norms and notation.**
>   To make the paper more accessible to a broader audience, we now provide mathematical definitions and interpretations of the different norms used. These are collected in Appendix A.1, and Section 3.1 explicitly refers the reader to this appendix at the first appearance of these norms.
>
> - **Restating the definition of $r$.**
>   To avoid the long distance between the definition and the first use of $r$, we now restate the definition directly in Theorem 1, so that readers do not need to search back in the text.
>
> - **Typographical corrections.**
>   We have corrected the typos that the reviewer pointed out, and we used this opportunity to perform an additional proofreading pass over the entire manuscript.
>
> Thank you again for these detailed comments. They have helped us improve the readability, organization, and empirical support of the paper, and we hope that the revisions address your concerns.

---

> > ### Comment · Reviewer_KDNU · 2025-11-26
> >
> > I thank the authors for these comments and changes. My review stands.

---

> ### Author Response · Authors · 2025-11-26
>
> We thank Reviewer KDNU again for the time invested in reviewing our work. However, with all due respect, **we would like to clearly explain why we believe the main remaining concern (“Muon is not widely used”) should not be grounds for leaning towards rejecting this paper, which we and other reviewers also see as well-prepared and presenting solid, technically significant results**.
>
> First, even on factual grounds, we respectfully disagree with the reviewer’s main concern that Muon is “not widely used.” Muon is already making a significant impact in the optimization and large-scale training communities, and many large-scale practitioners are actively experimenting with Muon in real training pipelines. This level of activity is unusually high for a newly proposed optimizer. Consequently, closing the gap between theory and practice, as we do in this work, is both timely and tremendously important.
>
> More importantly, even if one adopts a conservative view of how “widely” Muon is currently deployed in production, we do not see how this concern, by itself, can reasonably justify rejecting this paper. Our main contribution is theoretical: we close a clear and acknowledged gap between the idealized SVD-based Muon and the practical Newton–Schulz–based Muon that is used in implementations. Specifically, we (i) provide the first convergence guarantees for the original (practical NS-based) Muon, (ii) show that it matches the SVD-polar convergence rate up to a constant that decays doubly exponentially in the number of NS steps, and (iii) establish an improved rank dependence over SGD with momentum. These are substantial and technically nontrivial results about a matrix-aware optimizer that has already garnered significant interest. As the other reviewers emphasized, this is precisely the sort of work that builds foundational understanding and can help de-risk and further encourage adoption of such methods.
>
> In addition, we have significantly expanded the empirical section in the revised manuscript, including experiments on large-scale models (NanoGPT with 124M parameters and a GPT-2–based model with 1.3B parameters). These results demonstrate that the practical NS-based Muon analyzed in our theory is stable, efficient, and competitive in realistic training settings. We believe this directly addresses any residual doubts about the practical relevance of the optimizer and the applicability of our analysis.
>
> For these reasons, **we respectfully and sincerely request a re-evaluation of our work that fairly reflects the significance and validity of our contributions**. If there are any aspects of the paper that could be improved, we would be more than happy to incorporate further revisions. However, we believe that a decision leaning toward rejection, based primarily on the concerns stated in the original review, would not do justice to the strength, rigor, and significance of the results presented in this work.

---

### Official Review · Reviewer_WRnz · 2025-11-01

**Soundness:** 3
**Presentation:** 3
**Contribution:** 4
**Rating:** 8
**Confidence:** 3

**Summary:**

The paper investigates the convergence properties of the MUON algorithm, which demonstrates its superior properties against SGD or ADAM in terms of convergence speed. Other than existing works with an exact SVD orthogonalization of the momentum in its native matrix form, the paper studies the MUON with a finite steps of NEWTON–SCHULZ. The main results show that the MUON algorithm with a fixed finite steps of NEWTON–SCHULZ converges a stationary point with the same rate as the exact SVD-orthogonalization, up to a constant factor. The work further shows that this constant factor decays doubly exponentially in the fixed finite step of NEWTON–SCHULZ $q$ and improves with the degree of a polynomial used in NEWTON–SCHULZ $κ$.

**Strengths:**

The paper addresses a very demanding question about the convergence speed of the recently proposed MUON algorithm with a finite step of NEWTON–SCHULZ. As an approximate version of the MUON with the exact SVD orthogonalization of the momentum update, the presented analysis provides a solid theoretical understanding of the convergence behavior of the MUON in connection to its exact SVD counterpart, and ultimately guarantees the properties of deploying the MUON algorithm with a finite step of NEWTON–SCHULZ to large-scale training of DNNs, such as LLMs.

**Weaknesses:**

The core message conveyed in the paper was to show that the MUON algorithm with a finite step of NEWTON–SCHULZ converges similarly to some stationary point as its exact SVD counterpart in terms of convergence rate. However, one crucial aspect of training DNNs is the quality of stationary points. In other words, the discrepancy between stationary points generated by the MUONs with finite NEWTON–SCHULZ and the exact SVD can be large, thus is worth investigating.

**Questions:**

For a given setting of training a DNN, how close are the stationary points generated by the MUONs with finite NEWTON–SCHULZ and the exact SVD? If they are observed to be only satisfying the $\epsilon$ stationary point, are the qualities of these stationary points similar, in terms of generalizability to unseen data?

---

> ### Author Response · Authors · 2025-11-22
>
> We sincerely thank you for your thoughtful and positive review, and for recognizing both the novelty and importance of providing a convergence analysis of Muon with a finite number of Newton–Schulz (NS) steps. We especially appreciate your comments that our results help justify the use of Muon with finite NS steps in large-scale training scenarios.
>
> ### **On the quality of stationary points**
>  Thank you for the insightful question. In our setting, both Muon with SVD and Muon with finite NS steps optimize the same objective, and our theory shows that, when measured in epochs, the NS variant converges to a stationary point at essentially the same rate as the SVD-polar idealization, up to a constant factor that rapidly approaches 1 as $q$ increases theoretically and also empirically.
> While our current analysis is not about explicitly bound the distance between the stationary points of the two algorithms (which we believe is beyond the scope of this paper, although interesting), the doubly-exponential decay of the NS approximation error suggests that, for modest $q$, the orthogonalization directions – and hence the optimization trajectories – are very close, so one should expect comparable optimization and generalization quality in practice.
>
> To address your question more concretely, we examine empirical behavior in terms of both training loss and test performance (e.g., Figure 1). When we plot convergence per epoch, Muon with NS (e.g., $q=3$)  and Muon with exact SVD exhibit very similar training curves and test performances. This strongly suggests that both variants converge to stationary points of similar quality, in the sense that they eventually achieve very similar objective values and generalization performance on unseen data.
> More importantly, when we instead plot progress as a function of wall-clock time, Muon with NS reaches this regime much faster due to its substantially cheaper orthogonalization step, achieving stationary points of comparable quality with significantly improved time-to-solution.
>
> ### **Additional experiments**
>
> While your question was primarily about the quality of the attained solutions, we also took this opportunity to broaden our numerical validation and compare the training behavior of Muon more systematically. In the revised manuscript, we report results for a range of architectures and datasets, including MLPs on MNIST, ResNet-18 on CIFAR-100, WideResNet-28-10 on Tiny-ImageNet, and Transformer-based language models (NanoGPT with 124M parameters and a GPT-2–based model with 1.3B parameters) on FineWeb. Across this spectrum—from small image benchmarks to billion-parameter language models—we consistently observe that Muon with NS outperforms Muon with SVD in terms of wall-clock efficiency; that is, Muon with NS attains stationary points of similar quality substantially faster in wall-clock time.

---

### Official Review · Reviewer_Wnwj · 2025-11-01

**Soundness:** 3
**Presentation:** 3
**Contribution:** 3
**Rating:** 6
**Confidence:** 2

**Summary:**

In this paper, the convergence of Muon is analysed, for both the original and derivative versions. It is proved that Muon with Newton-Schulz converges with the same rate as SVD-polar idealization, and it can remove the square-root-of-rank loss compared to vector-based optimizers. The results explain the performance gain of Muon compared to vector-based optimizers, and are validated with experiments on the CIFAR-10 dataset and a CNN model.

**Strengths:**

This is one of the first convergence analysis result for Muon with finite steps of Newton-Schulz, which is a big step forward to narrow the gap between theory and practice.

**Weaknesses:**

The experimental part is relatively too simple. Experiments with larger and various datasets, larger number of parameters in the model, various model types such as MLP and transformer, more epochs, and more specific analysis how the results validate the theoretical analysis, will be nice.

**Questions:**

None.

---

> ### Author Response · Authors · 2025-11-22
>
> We sincerely appreciate the time and effort you have invested in reviewing our paper, and we are grateful for your positive assessment of our theoretical contributions, including recognizing that this is the first convergence analysis of Muon with a finite number of Newton–Schulz steps.
>
> While our main contributions are theoretical, we agree that stronger empirical support improves the paper. In response to your comments, we have substantially expanded the experimental section beyond the CifarNet/CIFAR-10 setup in the main text. We now report results on a broader range of architectures and datasets, and explicitly indicate the number of trainable parameters for each model:
> - MLP (0.5M parameters) / MNIST dataset
> - CifarNet (2M) / CIFAR-10  dataset
> - ResNet-18 (11.2M) / CIFAR-100  dataset
> - WideResNet-28-10 (36.6M) / Tiny-ImageNet dataset
> - NanoGPT (124M, Transformer) / FineWeb dataset
> - GPT-2 (1.3B, Transformer) / FineWeb dataset
>
> ---
> The corresponding train losses over wall-clock time are summarized below.
>
> - **Table 1: Train loss on MNIST using MLP (0.5M)**
>
> | Optimizer       | 50 sec | 100 sec  | 150 sec  | 200 sec  | 250 sec  |
> | :--------------- | :----: | :----: | :----: | :----: | :----: |
> | SGD-M            | 0.550                | 0.525 | 0.517 | 0.514 | 0.513 |
> | Muon with SVD    | 0.616                | 0.612 | 0.600 | 0.583 | 0.565 |
> | Muon ($q$=1)     | 0.526                | 0.514 | 0.506 | 0.502 | 0.501 |
> | Muon ($q$=2)     | 0.532                | 0.518 | 0.509 | 0.503 | 0.501 |
> | Muon ($q$=3)     | 0.548                | 0.529 | 0.511 | 0.503 | 0.501 |
>
>
>
> - **Table 2: Train loss on CIFAR-10 using CifarNet (2M)**
>
> | Optimizer        | 10 sec | 20 sec | 30 sec | 40 sec | 50 sec | 60 sec |
> | :--------------- | :----: | :----: | :----: | :----: | :----: | :----: |
> | SGD-M            | 1.366  | 1.157  | 1.084  | 1.045  | 1.021  | 1.010  |
> | Muon with SVD    | 2.211  | 1.334  | 1.227  | 1.207  | 1.193  | 1.182  |
> | Muon ($q$=1)     | 1.130  | 1.014  | 0.945  | 0.905  | 0.888  | 0.884  |
> | Muon ($q$=2)     | 1.129  | 1.064  | 1.001  | 0.932  | 0.888  | 0.876  |
> | Muon ($q$=3)     | 1.191  | 1.145  | 1.079  | 0.990  | 0.905  | 0.876  |
>
>
> - **Table 3: Train loss on CIFAR-100 using ResNet-18 (11.2M)**
>
> | Optimizer        | 100 sec | 200 sec | 300 sec | 400 sec | 500 sec |
> | :--------------- | :-----: | :-----: | :-----: | :-----: | :-----: |
> | SGD-M            | 3.096   | 2.612   | 2.324   | 2.157   | 2.072   |
> | Muon with SVD    | 3.202   | 2.767   | 2.668   | 2.594   | 2.522   |
> | Muon ($q$=1)     | 2.427   | 1.986   | 1.716   | 1.563   | 1.514   |
> | Muon ($q$=2)     | 2.407   | 2.127   | 1.826   | 1.584   | 1.495   |
> | Muon ($q$=3)     | 2.554   | 2.329   | 1.985   | 1.624   | 1.481   |
>
>
> - **Table 4: Train loss on Tiny-ImageNet using WideResNet-28-10 (36.6M)**
>
> | Optimizer        | 400 sec | 800 sec | 1200 sec | 1600 sec | 2000 sec |
> | :--------------- | :-----: | :-----: | :------: | :------: | :------: |
> | SGD-M            | 4.694   | 4.299   | 4.045    | 3.914    | 3.857    |
> | Muon with SVD    | 4.267   | 4.020   | 3.846    | 3.734    | 3.520    |
> | Muon ($q$=1)     | 4.035   | 3.369   | 2.903    | 2.495    | 2.144    |
> | Muon ($q$=2)     | 3.766   | 3.246   | 2.815    | 2.311    | 1.856    |
> | Muon ($q$=3)     | 3.666   | 3.317   | 2.929    | 2.372    | 1.800    |
>
>
> - **Table 5: Train loss on FineWeb using NanoGPT (124M)**
>
> | Optimizer        | 100 sec | 200 sec | 300 sec | 400 sec | 500 sec | 600 sec |
> | :--------------- | :-----: | :-----: | :-----: | :-----: | :-----: | :-----: |
> | SGD-M            | 10.938  | 7.656   | 6.000   | 5.375   | 4.938   | 4.594   |
> | Muon with SVD    | 10.250  | 9.188   | 8.625   | 8.062   | 7.594   | 7.125   |
> | Muon ($q$=1)     | 5.969   | 4.094   | 2.469   | 1.367   | 0.879   | 0.637   |
> | Muon ($q$=2)     | 5.000   | 2.344   | 1.047   | 0.395   | 0.157   | 0.030   |
> | Muon ($q$=3)     | 3.984   | 2.125   | 1.180   | 0.523   | 0.142   | 0.026   |
>
>
> - **Table 6: Train loss on FineWeb using GPT-2 based model (1.3B)**
>
> | Optimizer        | 600 sec | 1200 sec | 1800 sec | 2400 sec | 3000 sec | 3600 sec |
> | :--------------- | :-----: | :------: | :------: | :------: | :------: | :------: |
> | SGD-M            | 6.4062  | 5.3750   | 4.2812   | 3.2969   | 2.4219   | 2.0156   |
> | Muon with SVD    | 8.2280  | 8.0810   | 7.9340   | 7.7870   | 7.6400   | 7.4930   |
> | Muon ($q$=1)     | 6.0625  | 3.5156   | 1.0781   | 0.1436   | 0.1338   | 0.0109   |
> | Muon ($q$=2)     | 5.1562  | 2.8906   | 0.5898   | 0.1133   | 0.0349   | 0.0113   |
> | Muon ($q$=3)     | 5.6562  | 3.2031   | 1.2188   | 0.4609   | 0.1201   | 0.0286   |
>
> These experiments now cover MLPs, CNNs of varying depth and width, and Transformer-based language models with up to 1.3B parameters, which directly aligns with the kind of large-scale settings you mentioned in your review. The parameter counts span from $5 \times 10^5$ to $1.3 \times 10^9$, better reflecting practical large-scale training regimes.

---

> > ### Author Response · Authors · 2025-11-22
> >
> > Beyond adding more experimental results, we also analyze how the empirical behavior mirrors our theory. Across all settings, Muon with a small number of Newton–Schulz steps achieves essentially the same final training loss as Muon with exact SVD, consistent with our theoretical guarantee that the convergence rate differs only by a constant factor that rapidly approaches 1 as $q$ increases. Moreover, the performance gap between $q = 1$ and $q = 2,3$ shrinks quickly in all models, in line with the doubly-exponential decay of the Newton–Schulz approximation error established in our analysis.
> >
> > We have incorporated these new experiments and clarifications into a revised version of the manuscript.

---

### Meta-Review · Area_Chair_RVWn · 2026-01-06

**Summary:**

The reviews on this paper were almost universally positive, and it is a clear accept. It addresses an interesting theoretical question about an increasingly important method. Reviewers found the paper well motivated, easy to follow, and theoretically sound.

**Reviewer Concerns:**

The main reviewer concern (Reviewer KDNU), was that Muon is not widely adopted enough to be of interest -- which I disagree with.

Another concern of a few reviewers was that the experimental section did not include large enough scale experiments. However, the authors address this to some degree in the rebuttal, and in any case, I feel that this is out of scope of this paper, which is primarily a theoretical contribution. The presented experiments aim to confirm the theory, not to demonstrate the effectiveness of Muon (which is being done by other more empirically focused work).

**Reviewer Scores:**

All but one score is positive and would likely not have increased. The only negative reviewer (Reviewer KDNU) did not seem to be inclined to raise their score, although I do feel that the authors gave a reasonable rebuttal to their main concern (Muon not being popular enough).

---

### Decision · Program_Chairs · 2026-01-26

Accept (Poster)